# Linear Contextual Bandits With Interference

**Yang Xu** [1]   **Wenbin Lu** [1]   **Rui Song** [1]

## Abstract

Interference, a key concept in causal inference, extends the reward modeling process by accounting for the impact of one unit's actions on the rewards of others. In contextual bandit (CB) settings where multiple units are present in the same round, interference can significantly affect the estimation of expected rewards for different arms, thereby influencing the decision-making process. Although some prior work has explored multi-agent and adversarial bandits in interference-aware settings, how to model interference in CB remains significantly underexplored. In this paper, we introduce a systematic framework to address interference in Linear CB (LinCB), bridging the gap between causal inference and online decision-making. We propose a series of algorithms that explicitly quantify the interference effect in the reward modeling process and provide comprehensive theoretical guarantees, including sublinear regret bounds, finite sample upper bounds, and asymptotic properties. The effectiveness of our approach is demonstrated through simulations and synthetic data generated based on MovieLens data.

## 1. Introduction

Decision making widely exists in various real-world settings, such as advertising (Zhou et al., 2023), clinical trials (Tsiatis et al., 2008), and transportation (Wang et al., 2024). The core of decision-making problems lies in selecting the optimal *action* from a set of possible options to maximize an *outcome* or *reward* that is valuable to the decision-maker.

To simplify decision-making involving multiple *individuals* (or *agents*/*units*), it is often assumed that an individual's outcome depends only on the action applied to them and is unaffected by actions taken on others. This assumption, part of the Stable unit Treatment Value Assumption (SUTVA),

is common in causal inference literature. However, in many real-world situations, this assumption is frequently violated due to *interference*, where an individual's outcome is influenced by the actions of others (Rosenbaum, 2007).

During the COVID-19 pandemic, local governments have sought optimal personalized quarantine policies over extended periods. Here, each local government acts as an agent capable of making decisions over time. The observed reward, such as the number of infected individuals, reflects the community's health status following the application of a specific policy. Over time, governments adjust their policies based on the latest health status to mitigate virus spread. In this case, one community's health outcomes can be influenced by the quarantine policies of neighboring communities due to population movement and the nature of virus spread, illustrating the presence of interference.

Another example is in advertising markets, where advertisers manage multiple ad lines targeting overlapping audiences through different channels. The overlap in target consumers and the similarity between ad lines can cause the Return on Investment (ROI) of one ad to be influenced by the impression status of another. For instance, if one ad targets students interested in Nike shoes and another in Adidas shoes, the ROI of the Nike ad might be affected by whether customers have already seen the Adidas ad due to the shared audience and similar products.

In classical causal inference literature, the issue of interference has been extensively explored in both experimental design (Aronow & Samii, 2017; Viviano, 2020; Leung, 2022a;b; Viviano, 2024) and observational studies (Su et al., 2019; Bargagli-Stoffi et al., 2020; Forastiere et al., 2021; Qu et al., 2021). However, interference within online sequential decision-making frameworks, such as bandits, remains relatively underexplored.

When multiple agents, each capable of sequential decision-making, are involved, the problem is often known as the multi-agent multi-armed bandits (MAMAB). However, existing work in MAMAB faces limitations in incorporating agent-specific information that can enhance personalized decision-making. For example, in the COVID-19 spread scenario, communities (treated as *agents*) are not static; there may be population flows, new communities joining, and existing communities dissolving, each with unique charac-

[1]Department of Statistics, North Carolina State University, USA. Correspondence to: Rui Song <songray@gmail.com>.

*Proceedings of the 42nd International Conference on Machine Learning*, Vancouver, Canada. PMLR 267, 2025. Copyright 2025 by the author(s).

teristics. Similarly, in the advertising market, ad lines (also treated as *agents*) evolve quickly across campaigns, each with distinct characteristics such as size, content, and target audiences. Therefore, it is essential to consider a bandit framework under interference that accounts for agents or individuals evolving over time. This problem, often incorporating agent-specific information, is typically addressed within the framework of contextual bandits (CB).

Addressing interference in the CB framework presents two primary challenges. **First**, the *structure-wise* challenge: since the actions of multiple units influence each other's reward modeling, the resulting action space becomes high-dimensional without a precise understanding of the interference structure (Agarwal et al., 2024). Estimating heterogeneous treatment effects under interference in single-stage settings is already a complex task (Viviano, 2020; Leung, 2022b), and extending this to sequential decision-making scenarios like bandits (where the balance between exploration and exploitation is crucial) further compounds the challenge. **Second**, the *theory-wise* challenge: classical statistical inference tools, such as the Central Limit Theorem (CLT), become inapplicable as each action and corresponding reward may depend on all previously collected data. This presents a dual challenge: dependencies arise both across rounds due to online updates and within rounds due to interference between units. These dependencies violate the standard assumption of sample independence, complicating the application of traditional inference techniques.

While some recent works have addressed the issue of interference in MAMAB (Verstraeten et al., 2020; Bargiacchi et al., 2018; Dubey et al., 2020; Agarwal et al., 2024) and adversarial bandits (Jia et al., 2024), to the best of our knowledge, no existing work addresses interference within the contextual bandits framework. To bridge this gap, we introduce a comprehensive framework for linear contextual bandits in interference-aware scenarios, providing theoretical guarantees via both statistical inference and regret analysis. Our contributions are as follows.

**First**, we are the very first work to address the interference issue in contextual bandits with multiple units involved in each round, bridging the gap between SUTVA violations in causal inference and online decision making.

**Second**, we propose a systematic framework that extends the classical LinCB to interference-aware scenarios, offering comprehensive theoretical guarantees, including finite-sample upper bounds and sublinear regret. Remarkably, the regret bound achieves the minimax optimal rate in terms of the number of observed samples in the classical contextual bandits literature, even in the presence of interference.

**Third**, we are also the first work to establish the asymptotic properties of regression coefficients and the optimal value

function in an online setting with interference. We introduce a probability of exploration and a small clipping rate to ensure both estimation consistency and asymptotic properties. This clipping rate also balances the trade-off between statistical efficiency and regret minimization. The performance of our estimator is validated through simulation studies and synthetic data based on MovieLens.

The structure of this paper is as follows. Section 2 reviews recent work on interference and its integration into bandit frameworks. Section 3 introduces our framework for reward modeling and action selection, covering both offline optimization and online learning. In Section 4, we establish the theoretical foundations of the proposed estimators and the optimal value function, along with a regret analysis. Sections 5 and 6 demonstrate the effectiveness of our proposed algorithm through a simulation study and a synthetic dataset based on MovieLens. Finally, we provide a brief summary and discuss potential extensions in Section 7.

## 2. Related Work

**Interference in Single Stage.** In single-stage setting, existing literature varies significantly in defining interference, often assuming different structures for simplifying reward modeling. There are some work focuses on using partial interference and exposure mapping to quantify interference (Sobel, 2006; Qu et al., 2021; Hudgens & Halloran, 2008; Forastiere et al., 2021; Aronow & Samii, 2017; Bargagli-Stoffi et al., 2020). While this approach typically requires fewer assumptions during the reward modeling stage, it often relies on additional requirements, such as knowing the form of the exposure mapping function (Manski, 2013; Aronow & Samii, 2017; Bargagli-Stoffi et al., 2020) or assuming i.i.d. clusters (Qu et al., 2021). These assumptions can be overly restrictive in later stages and may not clearly explain direct and spillover effects.

On the contrary, another body of work, such as Su et al. (2019), considers the reward as a linear function of neighbors' covariates and treatments, which makes more interpretable assumptions in interference modeling and allows for comprehensive study in theory. This formulation is similar to Cliff & Ord (1981) and Getis (2009), where an autoregressive model is used to capture spatial interference with similar parametric modeling assumptions. In our work, we consider linear CB as the first step to handle interference while maintaining interpretability, and study both theories and algorithms under this framework.

**Cooperative Multi-Agent Bandits.** Multi-agent bandits typically assume a fixed set of $N$ agents making decisions over time. Some existing works, such as Martínez-Rubio et al. (2019) and Landgren et al. (2016), focus on information sharing between agents in a distributed system. While

sharing historical data can enhance the reward learning process among agents, these studies still assume that each agent's reward depends solely on its own actions, excluding the possibility of interference.

Another line of research, including Verstraeten et al. (2020), Bargiacchi et al. (2018), Dubey et al. (2020), Agarwal et al. (2024), and Jia et al. (2024), considers more general reward models where the actions of other agents can affect an individual agent's reward. Specifically, Bargiacchi et al. (2018) and Verstraeten et al. (2020) extended the Upper Confidence Bound (UCB) algorithm and Thompson Sampling (TS) from the classical MAB setting to multi-agent scenarios. Dubey et al. (2020) introduced a kernelized UCB algorithm, where interference is mediated through network contexts. Although Jia et al. (2024) also considered interference in bandits, their focus was on adversarial bandits with homogeneous actions, which is not as flexible as our approach where heterogeneous actions are allowed for units within the same round.

However, all of the aforementioned literature only considers an MAB setting with a fixed and finite number of agents, which is different from our setting where agents or units can vary over time with evolving contextual information.

## 3. Problem Formulation

In each round $t \in \{1, \ldots, T\}$, we assume there are $N_t$ units with contextual information $\boldsymbol{X}_{ti} \in \mathbb{R}^d$ in a network making sequential decisions simultaneously. At each time step $t$, unit $i \in \{1, \ldots, N_t\}$ chooses an action $A_{ti} \in \mathcal{A}$ and collects a reward $R_{ti}$. Define $\bar{N}_t = \sum_{s=1}^t N_t$ as the total number of units up to round $t$. Due to interference, the potential outcome of unit $i$ at round $t$ is defined as $R_{ti}(\boldsymbol{a}_t)$, where $\boldsymbol{a}_t = (a_{t1}, \ldots, a_{tN_t})^T$ is the action assignment vector for all units at round $t$.

To quantify the interference level between any two units in the same round, we suppose there exists a (normalized) adjacency matrix $\boldsymbol{W}_t = \{W_{t,ij}\}_{1 \le i,j \le N_t} \in \mathbb{R}^{N_t \times N_t}$ at round $t$, such that $W_{t,ij}$ denotes the causal interference strength from unit $i$ to unit $j$. Note that in our setup, $\boldsymbol{W}_t$ is not required to be symmetric, i.e., the causal interference strength from unit $i$ to unit $j$ can differ from that of $j$ to $i$. By default, we assume $W_{t,ij} \in [-1, 1]$, and $W_{t,ii} = 1$ for any $t$ and $1 \le i, j \le N_t$.

Defining an interference matrix $\boldsymbol{W}_t$ is both intuitive and flexible enough to model various real-world scenarios. For instance, in the special case where $\boldsymbol{W}_t$ is symmetric and takes values from $\{0, 1\}$, it can represent a neighborhood structure or friend network, where $W_{t,ij} = 1$ indicates that units $i$ and $j$ are connected, and $W_{t,ij} = 0$ means they are not. Since this information is derived from societal interactions (for example, during Covid-19, it was often known

which communities were connected by their geographical locations), we first consider the case where $\boldsymbol{W}_t$ is known[1]. We assume the reward of unit $i$ at round $t$ follows

$$R_{ti} = \sum_{j=1}^{N_t} W_{t,ij} \cdot f(\boldsymbol{X}_{tj}, A_{tj}) + \eta_{ti}, \qquad (1)$$

where the reward $R_{ti}$ is a linear combination of some payoff function $f$, and $\eta_{ti} \sim \mathcal{N}(0, \sigma^2)$ is a noise term satisfying $\eta_{ti} \perp (\boldsymbol{X}_t, \boldsymbol{W}_t) | \boldsymbol{A}_t$[2]. Throughout this paper, we assume that $|\mathbb{E}[R_{ti}]| \le U$, i.e., the expected reward of each unit $i$ at round $t$ can be uniformly bounded by a large constant $U$.

Assuming the reward generation process follows Equation (1) is both intuitive and possesses a very beneficial property. With some simple algebra, we can show that

$$
\begin{aligned}
\sum_{i=1}^{N_t} \mathbb{E}[R_{ti}] &= \sum_{i=1}^{N_t} \sum_{j=1}^{N_t} W_{t,ij} \cdot f(\boldsymbol{X}_{tj}, A_{tj}) \\
&= \sum_{j=1}^{N_t} \sum_{i=1}^{N_t} W_{t,ji} f(\boldsymbol{X}_{ti}, A_{ti}) = \sum_{i=1}^{N_t} \omega_{ti} f(\boldsymbol{X}_{ti}, A_{ti}),
\end{aligned}
\qquad (2)
$$

where we define $\omega_{ti} := \sum_{j=1}^{N_t} W_{t,ji}$ as the *interference weight* of unit $i$ at round $t$. The last term further indicates that the optimal action depends solely on the covariates of each individual unit, with interference influencing the direction of optimality through the sign of the weight $\omega_{ti}$. In the specific case where $\boldsymbol{W}_t \in \{0, 1\}^{N_t \times N_t}$, $\omega_{ti}$ can be interpreted as the "number of friends" or connections for unit $i$ at round $t$. This measure is straightforward to obtain; for example, in a COVID-19 context, it could represent the number of geographically adjacent communities. Since the optimal action that maximizes $\sum_{i=1}^{N_t} \mathbb{E}[R_{ti}]$ is determined by $\boldsymbol{X}_{ti}$ and $\omega_{ti}$, we do not need to account for the covariate information and actions of all units to achieve the globally optimal action. This simplifies the decision-making process and makes it more practical for real-world applications.

### 3.1. Offline Optimization

Let's consider a linear payoff function for $f$ to establish the entire framework and theory behind. Suppose there exists a coefficient vector $\boldsymbol{\beta}_a \in \mathbb{R}^d$ for each action $a \in \mathcal{A}$ to quantify the effect of each covariate in $\boldsymbol{X}_{ti}$. Similar to classical linear CB, we assume

$$f(\boldsymbol{X}_{tj}, a_{tj}) = \sum_{a \in \mathcal{A}} \boldsymbol{X}_{tj}^\top \boldsymbol{\beta}_a \cdot \boldsymbol{1}\{a_{tj} = a\}. \qquad (3)$$

By incorporating Equation (2), the optimal action maximizing cumulative reward for each round-unit pair $(t, i)$

---

[1]Extensions to unknown $\boldsymbol{W}_t$ will be discussed in Section 7.

[2]Here we assume the noise term $\eta_{ti}$ is conditionally independent of the contextual information and the interference matrix $\boldsymbol{W}_t$, given the action taken at round $t$. This is a more relaxed assumption compared to i.i.d. random noise.

(without considering exploration) is given by

$$A_{ti} = \arg\max_{a \in \mathcal{A}} \omega_{ti} \boldsymbol{X}_{ti}^\top \boldsymbol{\beta}_a, \qquad (4)$$

where the sign of $\omega_{ti}$ determines if the interference weight causes a flip in action that maximizes the cumulative reward.

To simplify notation, we introduce the following vector representations. Denote $\boldsymbol{\beta} = (\boldsymbol{\beta}_1^\top, \ldots, \boldsymbol{\beta}_K^\top)^\top \in \mathbb{R}^{dK}$, $\boldsymbol{A}_t = (A_{t1}, \ldots, A_{tN_t})^\top \in \mathbb{R}^{N_t}$, $\boldsymbol{R}_t = (R_{t1}, \ldots, R_{tN_t})^\top$ as the vector of rewards collected at round $t$, $\boldsymbol{X}_t = (\boldsymbol{X}_{t1}, \ldots, \boldsymbol{X}_{tN_t})^\top \in \mathbb{R}^{N_t \times d}$ as the covariate information matrix for all units at round $t$, and $\boldsymbol{W}_{ti} = \text{diag}(\{W_{t,ij}\}_{1 \leq j \leq N_t})$ as an $N_t \times N_t$ diagonal matrix. Furthermore, define $\mathbf{I}_{\boldsymbol{A}_t=a} := \text{diag}(\mathbf{1}\{\boldsymbol{A}_t = a\})$ as an $N_t \times N_t$ diagonal matrix where the $i$th diagonal element is the indicator of $\mathbf{1}\{A_{ti} = a\}$. We then define a $dK$-dimensional transformed covariate vector for each $(t,i)$ as:

$$\widetilde{\boldsymbol{X}}_{ti} = (\mathbf{1}_{N_t}^\top \boldsymbol{W}_{ti} \mathbf{I}_{\boldsymbol{A}_t=1} \boldsymbol{X}_t, \ldots, \mathbf{1}_{N_t}^\top \boldsymbol{W}_{ti} \mathbf{I}_{\boldsymbol{A}_t=K} \boldsymbol{X}_t)^\top. \qquad (5)$$

With some straightforward algebra, the expected reward can be expressed linearly as $\mathbb{E}[R_{ti}] = \widetilde{\boldsymbol{X}}_{ti}^\top \boldsymbol{\beta}$.

Denote $\widetilde{\boldsymbol{X}}_t = (\widetilde{\boldsymbol{X}}_{t1}, \ldots, \widetilde{\boldsymbol{X}}_{tN_t})^\top \in \mathbb{R}^{N_t \times dK}$ as the transformed covariate information matrix at round $t$. Furthermore, define $\widetilde{\boldsymbol{X}}_{1:t} = (\widetilde{\boldsymbol{X}}_1^\top, \ldots, \widetilde{\boldsymbol{X}}_t^\top)^\top \in \mathbb{R}^{\bar{N}_t \times dK}$, and $\boldsymbol{R}_{1:t} = (\boldsymbol{R}_1^\top, \ldots, \boldsymbol{R}_t^\top)^\top \in \mathbb{R}^{\bar{N}_t}$. Without exploration, the offline ordinary least square (OLS) estimator is given by

$$\widehat{\boldsymbol{\beta}}_t^* = (\widetilde{\boldsymbol{X}}_{1:t}^\top \widetilde{\boldsymbol{X}}_{1:t})^{-1} \widetilde{\boldsymbol{X}}_{1:t}^\top \boldsymbol{R}_{1:t} \in \mathbb{R}^{dK}. \qquad (6)$$

In an offline optimization setting, one can replace the true value of $\boldsymbol{\beta}$ in Equation (4) with $\widehat{\boldsymbol{\beta}}_t^*$ to obtain an estimate of the optimal individualized treatment rule.

### 3.2. Online Algorithms

In the context of online bandits with interference, we extend three algorithms from classical contextual bandits to account for the presence of interference: Linear Epsilon-Greedy With Interference (LinEGWI), Linear Upper Confidence Bound With Interference (LinUCBWI), and Linear Thompson Sampling With Interference (LinTSWI). These algorithms, summarized in Algorithm 1, differ primarily in Line 13 based on their respective exploration strategies.

**LinEGWI**: For EG-based exploration, we select action

$$a_{ti} = (1 - Z_{ti}) \cdot \arg\max_a \omega_{ti} \boldsymbol{X}_{ti}^\top \widehat{\boldsymbol{\beta}}_{ti,a} + Z_{ti} \cdot \text{DU}(1,K), \qquad (7)$$

where $Z_{ti} \sim \text{Ber}(\epsilon_{ti})$, and $\text{DU}(1,K)$ denotes the discrete uniform distribution s.t. $\mathbb{P}(A = a) = \frac{1}{K}$ for any $a \in [K]$.

**LinUCBWI**: Define $\widetilde{\Sigma}_t := (\widetilde{\boldsymbol{X}}_{1:t}^\top \widetilde{\boldsymbol{X}}_{1:t})^{-1}$. The UCB un-

---

**Algorithm 1** Linear Contextual Bandits with Interference

**Input**: Number of units $N_t$; Burning period $T_0$; Interference structure $\{\boldsymbol{W}_t\}_{1 \leq t \leq T}$; Clipping rate $p_t > O(\bar{N}_t^{-1/2})$.

1: **for** Time $t = 1, \cdots, T_0$ **do**
2:    $a_{ti} \sim \text{DU}(1,K), 1 \leq i \leq N_t$;
3: **end for**
4: $A \leftarrow \widetilde{\boldsymbol{X}}_{1:T_0}^\top \widetilde{\boldsymbol{X}}_{1:T_0}, b \leftarrow \boldsymbol{X}_{1:T_0}^\top \boldsymbol{R}_{1:T_0}$;
5: **for** Time $t = T_0 + 1, \cdots, T$ **do**
6:    Observe $N_t$ units with features $\{\boldsymbol{X}_{ti}\}_{1 \leq i \leq N_t}$
7:    Update $\widehat{\boldsymbol{\beta}}_{t-1} \leftarrow A^{-1}b$
8:    **for** unit $i = 1, 2, \cdots, N_t$ **do**
9:       Estimate the optimal arm

$$\widehat{\pi}_{ti} = \arg\max_{a \in \mathcal{A}} \omega_{ti} \boldsymbol{X}_{ti}^\top \widehat{\boldsymbol{\beta}}_{t-1,a};$$

10:       **if** $\lambda_{\min}\{\frac{1}{\bar{N}_{t-1}} \sum_{s=1}^{t-1} \sum_{i=1}^{N_t} \widetilde{\boldsymbol{X}}_{si} \widetilde{\boldsymbol{X}}_{si}^\top\} < p_{t-1} \cdot \lambda_{\min}\{\frac{1}{\bar{N}_{t-1}} \sum_{s=1}^{t-1} \sum_{i=1}^{N_t} \boldsymbol{X}_{si} \boldsymbol{X}_{si}^\top\}$ **then**
11:         Choose $A_{ti} \sim \text{DU}(1,K), 1 \leq i \leq N_t$;
12:       **else**
13:         Choose arm $a_{ti}$ by Equation (7), (9), or (10);
14:       **end if**
15:       Receive reward $R_{ti}$;
16:    **end for**
17:    Update $\widetilde{\boldsymbol{X}}_{ti}$ by Equation (5), $\forall i \in \{1, \ldots, N_t\}$
18:    Update $A \leftarrow A + \widetilde{\boldsymbol{X}}_t^\top \widetilde{\boldsymbol{X}}_t, b \leftarrow b + \widetilde{\boldsymbol{X}}_t^\top \boldsymbol{R}_t$
19: **end for**

---

der $A_{ti} = a \in \{1, \ldots, K\}$ can be derived as

$$\text{UCB}_{ti,a} \leftarrow \omega_{ti} \boldsymbol{X}_{ti}^\top \widehat{\boldsymbol{\beta}}_{t-1,a} + \alpha |\omega_{ti}| \cdot \sqrt{\boldsymbol{X}_{ti}^\top (\widetilde{\Sigma}_{t-1})_a^{-1} \boldsymbol{X}_{ti}}, \qquad (8)$$

where $\alpha$ is a hyperparameter that controls the exploration-exploitation tradeoff, and $(\widetilde{\Sigma}_{t-1})_a^{-1}$ denotes the $d \times d$ block-diagonal submatrix of $(\widetilde{\Sigma}_{t-1})^{-1}$ corresponding to the variance of $\widehat{\boldsymbol{\beta}}_{t-1,a}$. Thus, LinUCBWI selects the arm $a_{ti}$ by

$$a_{ti} = \arg\max_a \text{UCB}_{ti,a}. \qquad (9)$$

**LinTSWI**: Define $v$ as a hyper-parameter denoting the level of exploration. For unit $i$ at round $t$, LinTSWI will sample $\widetilde{\boldsymbol{\beta}}_{ti} \sim \mathcal{N}(\boldsymbol{\beta}_{t,\text{post}}, v^2 \boldsymbol{\Sigma}_{t,\text{post}}^{-1})$ and then choose arm $a_{ti}$ s.t.

$$a_{ti} = \arg\max_a \omega_{ti} \boldsymbol{X}_{ti}^\top \widetilde{\boldsymbol{\beta}}_{ti,a}. \qquad (10)$$

For space limit, we save the expression for posteriors, as well as other derivation details, to Appendix C.

There are two main differences between Algorithm 1 and classical LinCB algorithms. First, due to the presence of interference, $\boldsymbol{\beta}$ is estimated using the transformed covariate

information $\widetilde{\boldsymbol{X}}_{ti}$. This transformation depends on not only the covariates, but also the interference matrix and actions involving all units in round $t$. Second, we incorporate an additional clipping step in Line 10 to ensure that the probability of exploration does not decay faster than $O(\bar{N}_t^{-1/2})$ (see Assumption A.2 in Appendix A). This clipping step is crucial for maintaining sufficient exploration, which is necessary for estimation consistency and valid inference of $\boldsymbol{\beta}$, as will be detailed in the theory section. Note that when $\boldsymbol{W}_t \equiv I$ for all $t$, our method downgrades to the classical linear CB algorithms, aside from adding a step for clipping to ensure valid statistical inference.

**Remark.** The computational complexity of LinCBWI is approximately $O(d^2 K^2 \bar{N}_T)$. Specifically, for each unit $i$ in round $t$, the primary computational costs arise from matrix inversion (Line 7 of Algorithm 1) and the smallest eigenvalue computation (Line 10 of Algorithm 1), both requiring $O(d^3 K^3)$ in standard implementations. Using optimization techniques such as iterative eigenvalue decomposition and the Sherman-Morrison update can reduce this to $O(d^2 K^2)$. Multiplying by the total number of units $\bar{N}_T$ gives the overall complexity of Algorithm 1. In classical LinCB, the complexity is $O(d^2 K \bar{N}_T)$. LinCBWI introduces an additional factor of $K$ due to interference considerations, requiring a joint update of $\boldsymbol{\beta}_a$ for all $a \in \mathcal{A}$. A detailed comparison of computational time is provided in Section 5.3, where we show that although LinCBWI incurs slightly higher runtime, it remains computationally efficient in practice.

## 4. Theory

In this section, we provide comprehensive theoretical guarantees for the Linear CB under interference. Due to space limit, we move all assumptions to Appendix A.

### 4.1. Tail bound of the online OLS estimator

**Theorem 4.1.** *(Tail Bound of the Online OLS Estimator) Suppose Assumptions A.1-A.2 hold. In either LinUCBWI, LinTSWI or LinEGWI, for any $h > 0$, we have*

$$\mathbb{P}\left(\|\widehat{\boldsymbol{\beta}}_t - \boldsymbol{\beta}\|_1 > h\right) \leq dK \exp\left\{-\frac{h^2 \bar{N}_t p_t^2 \lambda^2}{2d^3 K^3 \sigma^2 L_w^2 L_x^2}\right\},$$

*where $L_w$ and $L_x$ are some constants for boundedness (see Assumption A.1), and $p_t$ controls the clipping rate in Algorithm 1.*

**Remark.** Given that $d$, $\sigma$, $L_w$ and $L_x$ are positive constants, the tail bound for the online OLS estimator simplifies to $\mathbb{P}\left(\|\widehat{\boldsymbol{\beta}}_t - \boldsymbol{\beta}\|_1 > h\right) \lesssim \exp(-h^2 \bar{N}_t p_{t-1}^2)$. As detailed in Assumption A.2, $p_t$ is a non-increasing sequence that controls the level of exploration in Line 10 of Algorithm 1. As long as $\bar{N}_t p_t^2 \to \infty$, $\widehat{\boldsymbol{\beta}}_t$ will converge in probability to $\boldsymbol{\beta}$.

Therefore, in Algorithm 1, we set the clipping rate at round $t$ to $p_t > O(\bar{N}_t^{-1/2})$ to ensure sufficient exploration and thus the convergence of the online OLS estimator.

### 4.2. The probability of exploration

Define $\kappa_{ti}(\omega_{ti}, \boldsymbol{X}_{ti}) = \mathbb{P}(a_{ti} \neq \widehat{\pi}_{t-1}(\boldsymbol{X}_{ti}))$, where the probability operator $\mathbb{P}$ is taken with respect to $a_{ti}$ and all historical data collected before round $t$. The term $\kappa_{ti}(\omega_{ti}, \boldsymbol{X}_{ti})$ represents the probability of exploration for unit $i$ at round $t$. We define the limit of $\kappa_{ti}(\omega_{ti}, \boldsymbol{X}_{ti})$ as $\kappa_\infty(\omega, \boldsymbol{x}) = \lim_{\bar{N}_t \to \infty} \mathbb{P}(a_{ti} \neq \pi^*(\boldsymbol{x}))$. Since $\kappa_{ti}$ is nonnegative by definition, it follows immediately from the Sandwich Theorem that $\kappa_\infty$ exists for both UCB and TS. For EG, $\kappa_\infty(\omega, \boldsymbol{X}) = \lim_{\bar{N}_t \to \infty} \kappa_{ti}(\omega, \boldsymbol{X}) = \lim_{\bar{N}_t \to \infty} \epsilon_{ti}/K$. In the following theorem, we establish the exploration upper bounds for LinUCBWI, LinTSWI, and LinEGWI, which are crucial for understanding the consistency conditions of the online OLS estimator and the necessity of clipping.

**Theorem 4.2.** *Suppose Assumptions A.1-A.3 hold. In either LinUCBWI, LinTSWI or LinEGWI, for any $0 < \xi < |\zeta_{ti}|/2$ with $\zeta_{ti} = \omega_{ti} \boldsymbol{X}_{ti}^\top (\boldsymbol{\beta}_1 - \boldsymbol{\beta}_0)$, we have*

*(1) In UCB, there exists a constant $C > 0$, such that*

$$\kappa_{ti}(\omega_{ti}, \boldsymbol{X}_{ti}) \leq CK^2 \left(\frac{2\alpha L_w L_x}{\sqrt{\bar{N}_{t-1} p_{t-1} \lambda}} + \xi\right)^\gamma$$
$$+ dK^3 \exp\left\{-\frac{\xi^2 \bar{N}_{t-1} p_{t-1}^2 \lambda^2}{8d^3 K^3 \sigma^2 L_w^4 L_x^4}\right\}.$$

*(2) In TS, we have*

$$\kappa_{ti}(\omega_{ti}, \boldsymbol{X}_{ti}) \leq K^2 \exp\left(-\frac{\bar{N}_{t-1} p_{t-1} \lambda (|\zeta_{ti}| - \xi)^2}{8v^2 L_w^2 L_x^2}\right)$$
$$+ 2dK^3 \exp\left\{-\frac{\xi^2 \bar{N}_{t-1} p_{t-1}^2 \lambda^2}{8d^3 K^3 \sigma^2 L_w^4 L_x^4}\right\}.$$

*(3) In EG, we have $\kappa_{ti}(\omega_{ti}, \boldsymbol{X}_{ti}) = \epsilon_{ti}(K-1)/K$.*

**Remark.** This theorem extends the results from Ye et al. (2023) to scenarios with interference and $K > 2$. As $\bar{N}_t p_t \to \infty$, the exploration probability in both UCB and TS will converge to 0 when $\bar{N}_t \to \infty$. Specifically, in UCB, the exploration upper bound consists of two components: the first term arises from the margin condition, and the second term from the tail bound of $\widehat{\boldsymbol{\beta}}_t$. When $\bar{N}_t$ is large, the second term, which decays at a rate of $O(\exp\{-\bar{N}_{t-1} p_{t-1}^2\})$, will be dominated by the first term, which decays at a rate of $O((\bar{N}_{t-1} p_{t-1})^{-\gamma/2})$ if we set $\xi = O((\bar{N}_{t-1} p_{t-1}^2)^{-1/2})$. In TS, the upper bound is dominated by the second term, which converges to 0 at a rate of $O(\exp\{-\bar{N}_{t-1} p_{t-1}^2\})$.

$L_w$ serves as an upper bound that controls the overall level of interference in Assumption A.1.c. A larger $L_w$ would increase the upper bound of exploration for both UCB and TS.

This is reasonable, as a higher level of interference typically results in a more relaxed upper bound on the probability of exploration. However, $L_w$ has no effect on EG where the probability of exploration is often pre-specified.

### 4.3. Statistical inference on the online OLS estimator $\widehat{\beta}_t$

Accurate estimation and inference of $\beta$ are crucial for effective decision-making. A consistent and stable estimate of $\beta_a$ plays a key role in the algorithm update (Line 13 of Algorithm 1), and valid inference provides deeper insights into feature importance/selection. Therefore, this section focuses on establishing the asymptotic properties of the online OLS estimator for $\beta$.

**Theorem 4.3.** *Suppose Assumptions A.1-A.3 hold, and $\bar{N}_t p_t \to \infty$ as $\bar{N}_t \to \infty$. We have*

$$\sqrt{\bar{N}_t}(\widehat{\beta}_t - \beta) \xrightarrow{\mathcal{D}} \mathcal{N}(\mathbf{0}_{dK}, \sigma^4 G^{-1}), \qquad (11)$$

*where, for simplicity, the expression of $G$ is specified in Equation (53) of Appendix H.*

**Remark.** Theorem 4.3 establishes the asymptotic normality of the online OLS estimator, providing an explicit form for its asymptotic variance. This result holds for the EG, UCB, and TS algorithms used for exploration. Despite the presence of interference, the asymptotic normality of the estimator only requires the total number of units $\bar{N}_t$ to approach infinity. In other words, bidirectional asymptotic normality is achieved as long as either $t \to \infty$ or the number of units at some stage $N_t \to \infty$.

The asymptotic normality of the online OLS estimator under interference requires careful treatment of dependencies between units both across time (due to the nature of online decision-making) and within each round (due to the existence of interference). To address this, we construct a specialized martingale difference sequence (as shown in the proof in Appendix H) that effectively accounts for the $\bar{N}_t$ units collected up to time $t$. This decomposition, together with the linear structure of interference, allows us to circumvent the independence requirement of classical CLT, which is central to establishing the asymptotic properties discussed in this and the following subsection.

### 4.4. Statistical inference on the optimal value $\widehat{V}^{\pi^*}$

In real applications, researchers may want to understand how closely the value function approaches its optimal level under the optimal policy $\pi^*$, at the end of a round $t$, as well as the magnitude of the potential impact of the optimal policy. This further motivate us to conduct estimation and inference on the optimal value function in online learning.

Suppose that the contextual information $X_{ti} \sim \mathcal{P}_{\mathcal{X}}$ and the interference weight $\omega_{ti} \sim \mathcal{W}$. Define the oracle policy

as $\pi^*(X_{ti}) = \arg\max_a \omega_{ti} X_{ti}^\top \beta_a$, and the optimal value function as $V^{\pi^*}$, which represents the expected reward under the oracle policy $\pi^*$. Specifically,

$$V^{\pi^*} = \mathbb{E}\left[\sum_{a \in \mathcal{A}} \mathbf{1}\{\pi^*(X) = a\}\omega\beta_a^\top X\right], \qquad (12)$$

where the expectation is taken w.r.t. $X$ and $\omega$.

Define $\widehat{\mu}_{t-1}^{(t,i)}(X_t, A_t) = \widetilde{X}_{ti}\widehat{\beta}_{t-1}$ as the expected reward that unit $i$ can obtain given the covariate information and actions of all units at round $t$. We propose a doubly robust (DR) estimator for $V^{\pi^*}$, where

$$\widehat{V}_t^{\mathrm{DR}} = \frac{1}{\bar{N}_t}\sum_{s=1}^{t}\sum_{i=1}^{N_s}\left[\frac{\mathbf{1}\{a_{si} = \widehat{\pi}_{s-1}(X_{si})\}}{1 - \widehat{\kappa}_{s-1}(\omega_{ti}, X_{si})} \cdot \left\{r_{si}\right.\right.$$
$$\left.\left. - \widehat{\mu}_{s-1}^{(i,s)}(X_s, \widehat{\pi}_{s-1}(X_s))\right\} + \widehat{\mu}_{s-1}^{(i,s)}(X_s, \widehat{\pi}_{s-1}(X_s))\right].$$

As a combination of inverse probability weighting (IPW) and the direct method (DM), our proposed doubly robust (DR) estimator offers double protection for consistency with $V^{\pi^*}$ under model misspecification in $\widehat{\kappa}_{t-1}$ or $\widehat{\mu}_{t-1}^{(i,t)}$. For brevity, some derivation details are moved to Appendix D.

**Theorem 4.4.** *Suppose Assumptions A.1-A.4 hold. We have*

$$\sqrt{\bar{N}_t}(\widehat{V}_t^{DR} - V^{\pi^*}) \xrightarrow{\mathcal{D}} \mathcal{N}(\mathbf{0}_{dK}, \sigma_V^2), \qquad (13)$$

*where $\sigma_V^2$ is given by*

$$\sigma_V^2 = \mathbb{E}\left[\frac{\sigma^2}{1 - \kappa_\infty(\omega, x)}\right] + Var\left[\sum_{a \in \mathcal{A}} \omega x^\top \beta_a \cdot \mathbf{1}\{\pi^*(x) = a\}\right]. \qquad (14)$$

**Remark.** The asymptotic variance of the optimal value function comprises two components. The first term arises from the IPW estimator and accounts for the variance of the random noise $\eta_{ti}$. The second term originates from the DM estimator and captures the variance due to uncertainty in the context $x$ and the interference weight $\omega$. Notably, our theorem extends the results of Ye et al. (2023) by establishing the asymptotic properties of the estimated optimal value function under interference. In the special case where $\omega \equiv 1$ in Equation (14), our results reduce to theirs.

### 4.5. Regret Bound

Now we establish the regret bound for Algorithm 1. Define the regret at the end of round $T$ as

$$R_T = \sum_{t=1}^{T}\sum_{i=1}^{N_t}\mathbb{E}\left[\omega_{ti}X_{ti}^\top\beta_{\pi^*(X_{ti})} - \omega_{ti}X_{ti}^\top\beta_{a_{ti}}\right].$$

**Theorem 4.5.** *For LinEGWI, LinUCBWI and LinTSWI in Algorithm 1, the general regret bound under interference*

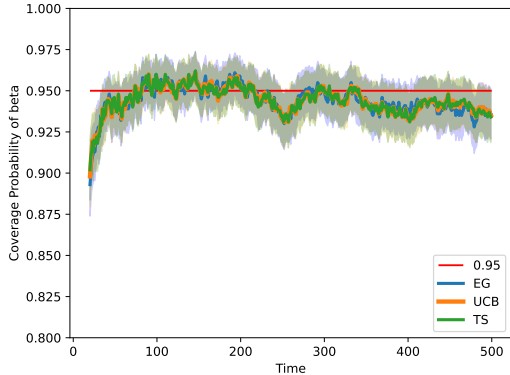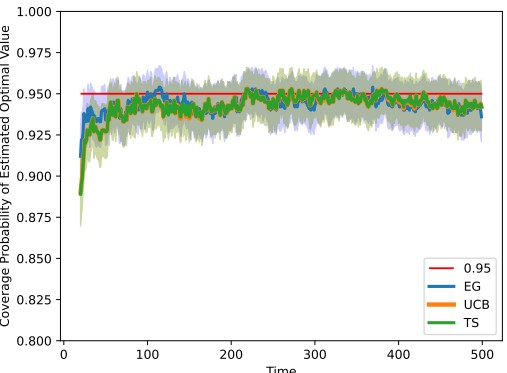

*Figure 1.* The coverage plot of $\boldsymbol{\beta}$ (a. left) and $V^{\pi^*}$ (b. right)

*can be derived as*

$$R_T = \sum_{t=1}^{T} \sum_{i=1}^{N_t} \mathbb{E}[R_{ti}^* - R_{ti}] = O(\bar{N}_T^{1/2} \log \bar{N}_T),$$

*which is sublinear in $\bar{N}_T$.*

**Remark.** The regret upper bound, even in the presence of interference, remains proportional to the square root of the total number of units (up to a logarithmic term). This aligns with the best achievable regret in literature without interference. This upper bound can be decomposed into two components: (1) the regret due to estimation accuracy (exploitation), denoted by $R_T^{(1)}$, and (2) the regret due to exploration, denoted by $R_T^{(2)}$. For the EG, UCB, and TS algorithms, the regret from exploitation is proven to be $o(\bar{N}_T^{-1/2})$ and is thus negligible. However, the regret due to exploration varies across algorithms. Specifically, in UCB and TS, $R_T^{(2)}$ also depends on the interference level $L_w$, which increases as $L_w$ becomes larger. In contrast, for EG, the probability of exploration is user-specified and independent of the interference weight $\omega$. As a result, $R_T^{(2)}$ is $O(\bar{N}_T^{1/2} \log \bar{N}_T)$ by setting $\epsilon_{ti}$ properly. For detailed expressions of the upper bounds for each algorithm and the order for hyper-parameters, please refer to Appendix J.

Before concluding this section, we summarize the main findings of Sections 4.3-4.5. Sections 4.3-4.4 and 4.5 are connected through a "clipping rate", $p_t$. Ideally, setting $p_t$ to 0 minimizes regret, but may undermine the consistency of $\boldsymbol{\beta}$ due to insufficient sampling of certain arms. On the other hand, a larger $p_t$ would lead to over-exploration and increase overall regret. This trade-off between Sections 4.3-4.4 and 4.5 is a novel perspective, motivating us to examine both aspects to balance statistical efficiency with regret minimization.

## 5. Simulation

### 5.1. Coverage Probability

To demonstrate the asymptotic normality of $\boldsymbol{\beta}$ and $V^{\pi^*}$ in Theorem 4.3-4.4, we estimate the asymptotic variance and verify whether the true value of $\boldsymbol{\beta}$ and $V^{\pi^*}$ falls within the estimated confidence interval with a high probability of coverage under $B = 1000$ times of replicates. By Equation (11) and (13), $\boldsymbol{\beta}$ falls into the confidence region if and only if $\frac{\bar{N}_t}{\sigma^4}(\widehat{\boldsymbol{\beta}} - \boldsymbol{\beta})^\top G(\widehat{\boldsymbol{\beta}} - \boldsymbol{\beta}) \leq \chi_\alpha^2(dK)$, where $df = dK$ is the degree of freedom of the chi-square distribution. Similarly, $V^{\pi^*}$ falls into the confidence interval if and only if $\sqrt{\bar{N}_t}|\widehat{V}_t^{\mathrm{DR}} - V^{\pi^*}| \leq z_{\alpha/2}\sigma_V$. Additional details of the simulation setup are summarized in Appendix B.1.

Coverage probabilities of the OLS estimator $\widehat{\boldsymbol{\beta}}$ and optimal value function $V^{\pi^*}$ under three exploration algorithms (LinEGWI, LinUCBWI, and LinTSWI) are shown in Figure 1. As we can see, the coverage consistently hovers around 95%, with the estimated confidence band almost always covering the red line. This result supports the validity of the statistical inference presented in Theorems 4.3 and 4.4.

### 5.2. Comparison with Baseline Approaches

First, we compare our proposed method with the classical linear CB algorithms to illustrate the importance of taking interference into consideration. The results are shown in Figure 2 based on $B = 100$ times of replication.

As shown, our approaches (LinEGWI, LinUCBWI, and LinTSWI) achieve significantly lower average regrets at a faster rate than classical LinCB algorithms, showcasing the effectiveness of Linear Contextual Bandits With Interference (LinCBWI) in addressing interference. Without interference (Figure 6 in Appendix B.4), our algorithm reduces to the classical LinCB method and delivers similar

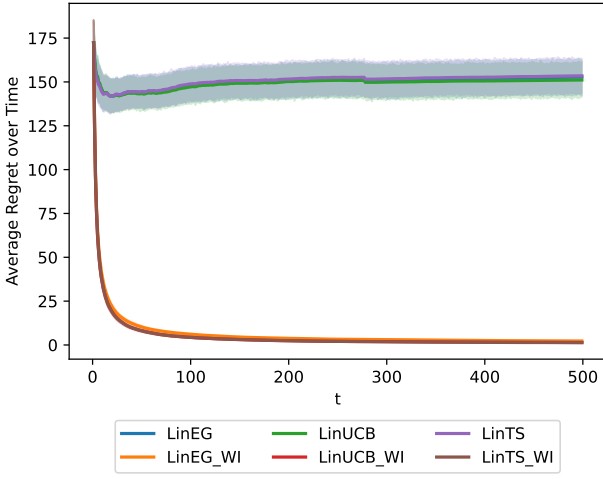

*Figure 2.* Comparison of average regret in the presence of interference with $K = 3$ arms

results. Further details can be found in Appendix B.2.

## 5.3. Computational Time of LinCBWI

At the end of Section 3.2, we briefly discussed the theoretical computational complexity of running Algorithm 1. To empirically validate this analysis, we compare the runtime of LinCBWI to that of classical LinCB under the same setting as in Section 5.2. The results are presented in Figure 3. As expected, LinCBWI incurs slightly higher computational time due to its joint updates, but the overall runtime remains within a few seconds. Therefore, computational complexity is unlikely to pose a practical bottleneck.

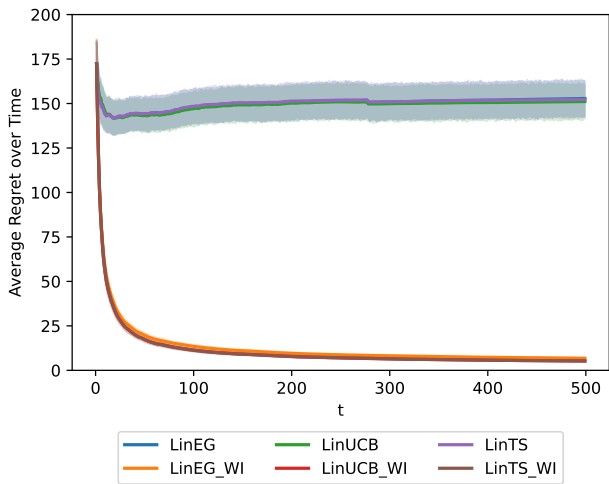

*Figure 3.* Computational Time Comparison between our proposed algorithm (LinCBWI) and baselines (LinCB)

## 5.4. Sensitivity Analysis When $W_t$ Misspecified

In this subsection, we conduct a sensitivity analysis to evaluate the impact of misspecifying the interference matrix. The simulation setup follows that of Section 5.2, except we manually introduce a perturbed matrix for LinCBWI: $\breve{W}_t = W_t + \Xi_t$, where $\Xi_t = \{\xi_{ij}\}_{1 \leq i,j \leq N_t}$ with $\xi_{ij} \sim \text{Unif}(-b, b)$. Any resulting values outside the range $[-1, 1]$ are clipped. Intuitively, a larger $b$ indicates a higher degree of misspecification. We consider $b \in \{0, 0.5, 1, 2\}$ to assess LinCBWI's robustness.

Due to space constraints, full results are deferred to Appendix E; the main paper presents results for $b = 1$ as a representative case. Notably, $b = 1$ corresponds to a substantial level of misspecification, given that $W_t$ is normalized to lie within $[-1, 1]$.

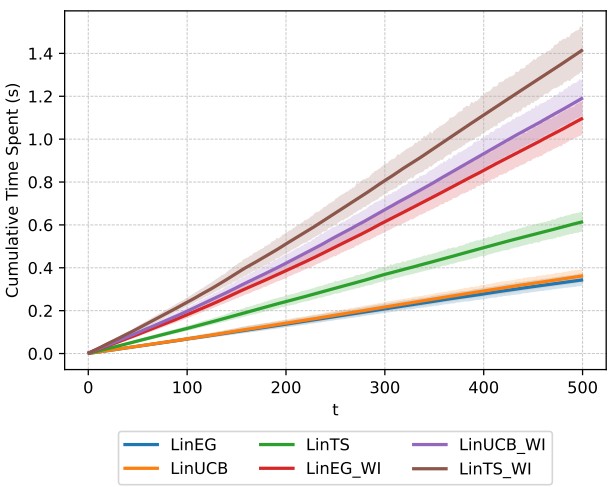

*Figure 4.* Average regret comparison when $b = 1$

Figure 4 shows that LinCBWI remains robust, achieving near-zero average regret despite the severe perturbation. This suggests that incorporating interference structures, even when estimated imprecisely, enhances decision-making performance and provides deeper modeling flexibility. The robustness of LinCBWI alleviates practical concerns about unknown or misspecified interference.

## 6. Synthetic Data based on MovieLens

The MovieLens 1M dataset (Harper & Konstan, 2015) contains over 1 million movie ratings from 6k users, aiding movie recommendations based on historical ratings. At each round $t$, user $i$ with context $X_{ti}$ is recommended a movie genre ($A_{ti}$) and provides a rating ($R_{ti}$). Here, we define $A = 1$ for "Comedy" and $A = 0$ for "Drama". There are two types of interference affect reward modeling for $R_{ti}$,

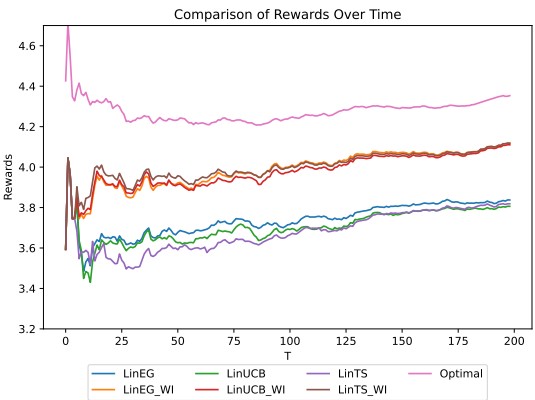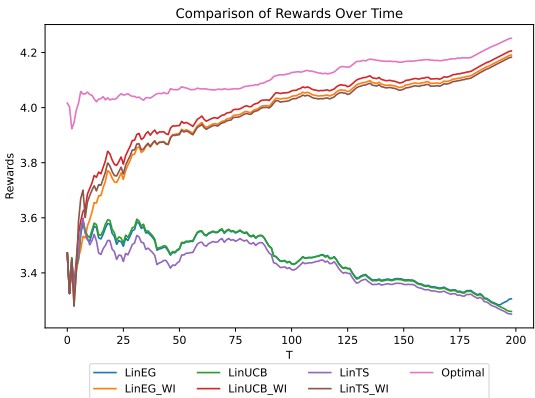

*Figure 5.* Average rating comparison under reward generating model I (RGP1, left) and II (RGP2, right)

which is overlooked in classical bandits:

(1) Users often rate multiple movies within a short time (a round), indicating that one recommendation made to a user may influence their ratings for others in the same round.

(2) Contextual connections (e.g., occupation, ZIP code, age) between users in the same round can cause one recommendation made to a user to affect the ratings of other users.

We consider two different pseudo-true reward generating processes (RGPs), RGP1 and RGP2, to further evaluate the performance of our algorithms. Specifically, RGP1 assumes only the presence of interference, while RGP2 additionally assumes that the true reward model satisfies linearity. Detailed setup descriptions are summarized in Appendix B.3.

The comparison results for both RGPs are shown in Figure 5. In both figures, our algorithms consistently outperform classical contextual bandit approaches. Notably, in the case of RGP1, there is a gap of average reward between our algorithms and the oracle model, which disappears in RGP2. This gap likely arises because the true reward model may not be linear. Despite so, the effectiveness of our approach can be validated through both scenarios as it consistently outperforms baselines due to its ability to account for the potential interference structure. All supplementary code is available at our Github repository.

## 7. Summary and Future Directions

In this paper, we propose a series of algorithms to address interference in LinCB, supported by thorough analyses of regret, upper bounds, and the asymptotic properties of the online estimators. Before concluding, we outline two practical extensions of this work:

Our algorithm can naturally extend to cases where the interference matrix $\boldsymbol{W}_t$ is unknown. Inspired by low-

rank factorization (Shi et al., 2019; Agarwal et al., 2020), we can assume that $\boldsymbol{W}_t$ exhibits a learnable low-rank structure informed by agents' contextual information, i.e., $\boldsymbol{W}_t = \boldsymbol{X}_t \Phi \boldsymbol{X}_t^\top$, where $\Phi \in \mathbb{R}^{d \times d}$ represents a universal weighting matrix. This matrix can be learned alongside the bandit algorithm to generalize the interference structure. Extending the algorithm in this way requires jointly estimating $\Phi$ and $\boldsymbol{\beta}$, which can be formulated as the optimization problem $\arg\min_{\boldsymbol{\beta},\Phi} \sum_{t=1}^{T} \left( \mathbf{1}^\top \boldsymbol{R}_t - \mathbf{1}^\top \widetilde{\boldsymbol{X}}_t \boldsymbol{\beta} \right)^2 + \lambda \|\Phi\|_1$, where $\lambda$ controls the sparsity of $\Phi$. While promising, this approach requires rigorous justification to ensure identifiability, as well as further computational and theoretical development. Given the scope of this extension, it presents an intriguing direction for future research.

It is also possible to generalize the reward model in Equations (2) and (3). One plausible extension involves replacing the linear assumption with a neural network, as in neural contextual bandits (Zhou et al., 2020; Zhang et al., 2020). While this enhances modeling flexibility and practical applicability, it may introduce extra complexity for statistical inference in theoretical analysis.

Another promising direction is to investigate the regret lower bound in the presence of interference. In this paper, we have shown that, despite interference, extensions of classical algorithms such as EG, UCB, and TS maintain the standard optimal regret rate proportional to the square root of the sample size. This is the best achievable regret bound within these exploration algorithms. Parallel research efforts, though at the cost of reduced generalizability, have explored methods to decrease exploration probabilities and achieve smaller regret bounds, as discussed in Goldenshluger & Zeevi (2013), Han et al. (2020), and He et al. (2022). Extending such approaches to settings with interference remains an open and exciting direction for future research.

## Acknowledgments

This research was conducted under the support of the National Science Foundation through Grant DMS-2113637.

## Impact Statement

This paper is the first to address the interference problem in the contextual bandit problems, with broad applicability to real-world domains such as recommendation systems, personalization, clinical applications, and more. We believe that this work, along with the potential extensions discussed in the summary, will inspire future research that advances decision-making across various machine learning applications.

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

# A. Assumptions

**Assumption A.1.** (Boundedness)

a. Define $\Sigma := \mathbb{E}\left[\boldsymbol{x}\boldsymbol{x}^\top\right]$ as the covariance of contextual information. There exists a constant $\lambda > 0$, such that $\lambda_{\min}(\Sigma) > \lambda$.

b. $\forall \boldsymbol{X}_{ti} \in \mathcal{X}$, there exists a constant $L_x$, such that $\|\boldsymbol{X}_{ti}\|_2 \leq L_x$ for any $t \in [T]$ and $i \in [N_t]$.

c. $\forall \boldsymbol{W}_t \in \mathcal{W}$, there exists a constant $L_w$, such that $\sum_j |W_{t,ij}| \leq L_w$ and $\sum_j |W_{t,ji}| \leq L_w$ for any $t \in [T]$ and $i \in [N_t]$.

**Assumption A.2.** (Clipping) For any action $a$ and round $t$, there exists a positive and non-increasing sequence $p_t$, such that $\lambda_{\min}\left\{\frac{1}{\bar{N}_{t-1}} \sum_{s=1}^{t-1} \sum_{i=1}^{N_s} \widetilde{\boldsymbol{X}}_{si} \widetilde{\boldsymbol{X}}_{si}^\top\right\} > p_{t-1} \cdot \lambda_{\min}(\Sigma)$.

**Assumption A.3.** (Margin Condition) For any $\epsilon > 0$, there exists a positive constant $\gamma > 0$, such that for any two different arms $a$ and $a'$, we have $\mathbb{P}(0 < |f(\boldsymbol{X}, a) - f(\boldsymbol{X}, a')| < \epsilon) = O(\epsilon^\gamma)$.

Assumption A.1 includes several bounded conditions. Assumption A.1.a ensures that there is no strong collinearity between different features, which is necessary for a stable OLS estimator. This condition is commonly assumed in bandit-related inference papers (Zhang et al., 2020; Chen et al., 2021; Ye et al., 2023). Assumptions A.1.b and A.1.c ensure that the contextual information and the interference level for each individual unit are bounded.

Assumption A.2 is a technical requirement that guarantees the bandit algorithm explores all actions sufficiently at a rate of $p_t$, enabling consistent estimation of the OLS estimator. This exploration procedure is widely assumed in bandits inference literature (Deshpande et al., 2018; Hadad et al., 2021; Ye et al., 2023). Notice that Assumption A.2 is actively enforced in our algorithm. Specifically, Line 10 of Algorithm 1 ensures that if an arm is explored insufficiently (determined based on and a smallest eigenvalue comparison), the algorithm enforces additional exploration, preventing extreme imbalance and ensuring statistical consistency. In our simulation studies, we simply set $p_t \equiv 0.01$ and observed that Line 10 is rarely triggered. Therefore, this assumption is unlikely to pose significant practical concerns.

Assumption A.3, known as the margin condition, is a standard assumption in the contextual bandits literature (Audibert & Tsybakov, 2007; Luedtke & Van Der Laan, 2016). It ensures that the expected rewards of different arms are sufficiently separated, thereby making the bandit learning problem well-posed and meaningful.

**Assumption A.4.** (Rate Double Robustness) Define the $L_2$ norm as $\|z_t\|_{2,N_T} = \sqrt{\frac{1}{\bar{N}_T} \sum_{t=1}^{T} \sum_{i=1}^{N_t} z_t^2}$. We assume that the convergence rate of propensity score model $\|\hat{\kappa}_{t-1}(\omega_{ti}, \boldsymbol{X}_{ti}) - \kappa_{t-1}(\omega_{ti}, \boldsymbol{X}_{ti})\|_{2,N_T} = O_p(\bar{N}_T^{-\alpha_1})$, the convergence of outcome regression model $\|\widehat{\mu}_{t-1}^{(t,i)}(\boldsymbol{X}_t, \widehat{\pi}_{t-1}(\boldsymbol{X}_t)) - \mu^{(t,i)}(\boldsymbol{X}_t, \widehat{\pi}_{t-1}(\boldsymbol{X}_t))\|_{2,N_T} = O_p(\bar{N}_T^{-\alpha_2})$, with $\alpha_1 + \alpha_2 > 1/2$.

Assumption A.4 requires that the convergence rates of the conditional mean function and the estimated probability of exploration satisfy certain conditions. This is a standard assumption in causal inference literature, as noted in Luedtke & Van Der Laan (2016); Kennedy (2022). In our setting, this assumption is almost always satisfied, given that $\|\widehat{\mu}_{t-1}^{(t,i)}(\boldsymbol{X}_t, \widehat{\pi}_{t-1}(\boldsymbol{X}_t)) - \mu^{(t,i)}(\boldsymbol{X}_t, \widehat{\pi}_{t-1}(\boldsymbol{X}_t))\|_{2,N_T} = O_p(\bar{N}_T^{-1/2})$ follows directly from Theorem 4.3. Therefore, it suffices to ensure that $\|\hat{\kappa}_{t-1}(\omega_{ti}, \boldsymbol{X}_{ti}) - \kappa_{t-1}(\omega_{ti}, \boldsymbol{X}_{ti})\|_{2,N_T} = o_p(1)$ for Assumption A.3 to hold. This can be easily achieved by using a sample-based exploration estimand. In practice, as $\hat{\kappa}_{t-1}(\omega_{ti}, \boldsymbol{X}_{ti})$ tends to be small as $t$ increases, we set $\hat{\kappa}_{t-1}(\omega_{ti}, \boldsymbol{X}_{ti}) = \sum_{s \leq t-1, i \in [N_s]} \mathbf{1}\{A_{si} \neq \widehat{\pi}(\boldsymbol{X}_{si})\}/\bar{N}_{t-1}$, which proves to be sufficient in simulation and real data analysis.

# B. Simulation Setup and a Supplementary Plot Without Interference

## B.1. Simulation Setup in Section 5.1

The simulation setup of testing coverage probability is as follows. In the estimation of $\boldsymbol{\beta}$, the entire process is replicated for $B = 1000$ times to calculate the empirical coverage. For each replication, we assume there are a total of $T = 500$ rounds, and we randomly sample the true $\boldsymbol{\beta}$ from $\boldsymbol{\beta}_0 = (2, -3, 1)^\top$ and $\boldsymbol{\beta}_1 = (1, 1, 3)^\top$.

In the estimation of $\boldsymbol{\beta}$, we assume $N_t \sim \text{Poisson}(5)$ units are interacting with the environment. $\boldsymbol{X}_{ti} = (X_{ti,1}, \ldots, X_{ti,3}) \in \mathbb{R}^3$ denotes the feature information of unit $i$ at round $t$, where $X_{ti,1} \equiv 1$, $X_{ti,2} \sim \mathcal{N}(4,1)$, $X_{ti,3} \sim \text{Unif}(0,3)$, and all of the samples are i.i.d. over $(t,i)$. At each round, we generate the interference matrix $\boldsymbol{W}_t \in \mathbb{R}^{N_t \times N_t}$ as follows. Suppose the

diagonal elements $W_{ii} = 1$. For each $i > j$, we generate $W_{ij} \sim \alpha \cdot \text{Unif}(-0.6, -0.3) + (1 - \alpha) \cdot \text{Unif}(0.1, 0.4)$, where $\alpha \sim \text{DU}(1, K)$.

In the estimation of $V^{\pi*}$, for a more balanced variance composition in Equation (14), we set up the data generating process for $\boldsymbol{W}_t$ and $\boldsymbol{X}_{ti}$ as follows. For each $i \neq j$, we generate $W_{t,ij} \sim \alpha \cdot \text{Unif}(-0.2, -0.1) + (1 - \alpha) \cdot \text{Unif}(0.05, 0.2)$, where $\alpha \sim \text{DU}(1, K)$. For contextual information, we generate $\boldsymbol{X}_{ti} = (X_{ti,1}, \ldots, X_{ti,3}) \in \mathbb{R}^3$ for unit $i$ at round $t$, where $X_{ti,1} \equiv 0.2$, $X_{ti,2} \sim \mathcal{N}(0.8, 0.04)$, $X_{ti,3} \sim \text{Unif}(0, 0.6)$, and all of the samples are i.i.d. over $(t, i)$.

To establish the asymptotic normality of the optimal value function, we need to posit another mild assumption regarding the convergence rates of the two estimation models (propensity score and outcome regression models) presented in the main paper.

## B.2. Simulation Setup in Section 5.2

In reward comparison, we set $K = 3$ arms and a total of $T = 500$ rounds. For each round $t$, a total of $N_t \sim \text{Poisson}(\lambda)$ units will interact with the environment. We generate the interference matrix $\boldsymbol{W}_t \in \mathbb{R}^{N_t \times N_t}$ as follows: Suppose the diagonal elements $W_{ii} = 1$. For each $i > j$, we generate $W_{ij} \sim \alpha \cdot \text{Unif}(-0.9, -0.6) + (1 - \alpha) \cdot \text{Unif}(0.1, 0.4)$, where $\alpha \sim \text{DU}(1, K)$. The lower triangular elements are set equivalent to the upper triangular.

Define $\boldsymbol{X}_{ti} = (X_{ti,1}, \ldots, X_{ti,d}) \in \mathbb{R}^d$ as the feature information of unit $i$ at round $t$. Here, we let $d = 5$, where the first column is intercept $1$, $(X_{ti,2}, X_{ti,3}) \sim MVN(\boldsymbol{\mu}, \Sigma)$, and $(X_{ti,4}, X_{ti,5})$ follows some uniform distribution.

Following the reward-generating process described in Equation (2) with a linear payoff function (or equivalently, Equation (15)), we uniformly sample $\boldsymbol{\beta}_0 \sim \text{Unif}(-5, 1)$, $\boldsymbol{\beta}_1 \sim \text{Unif}(-1, 5)$, $\boldsymbol{\beta}_2 \sim \text{Unif}(-3, 3)$, and replicate this process for $S = 100$ times to test the robustness of different approaches w.r.t. the change of environment. All experiments were conducted on a local computer with 16 GB of memory.

## B.3. Real Data Setup in Section 6

Based on the timestamps of each rating and the relative user density, we divided the dataset into $T = 200$ rounds. For each round-unit pair $(t, i)$, $\boldsymbol{X}_{ti} \in \mathbb{R}^d$ is a $d = 7$ dimensional vector that includes an intercept term, age, gender, and 4 dummy variables representing the top 4 most popular occupation types. We construct an interference matrix $\boldsymbol{W}_t$ based on the contextual information of users in the same round using normalized Jaccard similarity. Note that if a user provides multiple ratings in the same round (which is highly likely according to our observations), we treat them as "different" users with the same contextual information, thus the corresponding element in $\boldsymbol{W}_t$ is set to 1. We proceed with two different pseudo-true reward generating processes.

I:  For each user $j$, we calculate $\bar{R}_j(a)$ as the average rating of user $j$ under movie type $A = a$. Then the true reward of user $i$ at round $t$ is given by $R_{ti} = \sum_{j=1}^{N_t} W_{t,ij} \bar{R}_j(a)$.

II:  Following Equation (2) with linear payoff functions, we fit a linear regression model to $R_{ti}$, with

$$R_{ti} = \sum_{a \in \mathcal{A}} \sum_{j=1}^{N_t} W_{t,ij} \boldsymbol{X}_{tj}^\top \boldsymbol{\beta}_a \mathbf{1}\{A_{tj} = a\} + \epsilon_{ti}, \tag{15}$$

to estimate $\boldsymbol{\beta}_0$, $\boldsymbol{\beta}_1$, and $\sigma$ as specified above. We then use these estimated values to regenerate $\widetilde{R}_{ti}$, which we assume represents the true reward of user $i$ at round $t$.

## B.4. Results Comparison Without Interference

Using the same simulation setup as in Section 5.2, but with the interference matrix $\boldsymbol{W}_t$ replaced by an identity matrix, we compare the results of the classical linear CB algorithm with our proposed methods. The results, shown in Figure 6, indicate that all methods yield comparable performance over time, with average regrets converging to zero at a rapid rate. This illustrates that our method, LinCBWI, will downgrade to classical LinCB algorithms when interference does not exist.

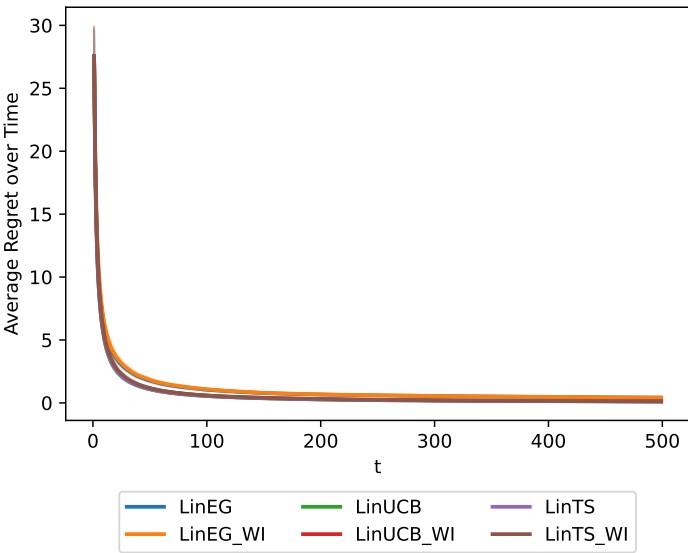

*Figure 6.* Comparison of average regret in the absence of interference

## C. The Derivation of Three Exploration Strategies

In this section, we provide a detailed derivation of the exploration strategies for LinEGWI, LinUCBWI, and LinTSWI, which were briefly introduced in Section 3.2.

### C.1. LinEGWI

First, to generalize the classical EG algorithm in the presence of interference, we explore different arms with probability $\epsilon_{ti}$ and select the estimated optimal arm with probability $1 - \epsilon_{ti}$. That is,

$$a_{ti} = (1 - Z_{ti}) \cdot \arg\max_a \omega_{ti} \boldsymbol{X}_{ti}^\top \widehat{\boldsymbol{\beta}}_{ti,a} + Z_{ti} \cdot \mathrm{DU}(1, K), \tag{16}$$

where $Z_{ti} \sim \mathrm{Ber}(\epsilon_{ti})$, and $\mathrm{DU}(1, K)$ denotes the discrete uniform distribution s.t. $\mathbb{P}(A = a) = \frac{1}{K}$ for any $a \in [K]$.

### C.2. LinUCBWI

We next consider the extension of linear UCB to interference-existing scenarios. The key idea behind UCB is to use the variance of the parameter estimates, specifically the upper confidence bound, to guide exploration. In the presence of interference, this process is equivalent to comparing the UCBs of $\omega_{ti} \boldsymbol{X}_{ti}^\top \widehat{\boldsymbol{\beta}}_a$ and selecting the action that maximizes the value. Define $\widetilde{\Sigma}_t := \left( \widetilde{\boldsymbol{X}}_{1:t}^\top \widetilde{\boldsymbol{X}}_{1:t} \right)^{-1}$. Since $\mathrm{Var}(\widehat{\boldsymbol{\beta}}_t) = \sigma^2 \cdot \widetilde{\Sigma}_t$, the UCB under $A_{ti} = a \in \{1, \ldots, K\}$ can be derived as follows:

$$\mathrm{UCB}_{ti,a} \leftarrow \omega_{ti} \boldsymbol{X}_{ti}^\top \widehat{\boldsymbol{\beta}}_{t-1,a} + \alpha |\omega_{ti}| \cdot \sqrt{\boldsymbol{X}_{ti}^\top (\widetilde{\Sigma}_{t-1})_a^{-1} \boldsymbol{X}_{ti}}, \tag{17}$$

where $\alpha$ is a hyperparameter that controls the exploration-exploitation tradeoff, and $(\widetilde{\Sigma}_{t-1})_a^{-1}$ denotes the $d \times d$ block-diagonal submatrix of $(\widetilde{\Sigma}_{t-1})^{-1}$ corresponding to the variance of $\widehat{\boldsymbol{\beta}}_{t-1,a}$. Thus, LinUCBWI algorithm selects the arm $a_{ti}$ according to

$$a_{ti} = \arg\max_a \mathrm{UCB}_{ti,a}. \tag{18}$$

## C.3. LinTSWI

In linear Thompson sampling, the prior of $\boldsymbol{\beta}$ is often pre-specified. At each round, units with transformed covariate matrix $\widetilde{\boldsymbol{X}}_{ti}$ is used to update the posterior of $\boldsymbol{\beta}$ after collecting the reward $R_{ti}$. Here, we adapt a normal prior for $\boldsymbol{\beta}$, which follows

$$\begin{aligned}
&\text{(Prior)} \quad \boldsymbol{\beta} \sim \pi(\boldsymbol{\beta}) := \mathcal{N}(\boldsymbol{\mu}_0, \boldsymbol{\Sigma}_0) \\
&\text{(Update)} \quad R_{ti} = \widetilde{\boldsymbol{X}}_{ti}^\top \boldsymbol{\beta} + \eta_{ti}, \quad \forall i \in \{1, \ldots, N_t\}
\end{aligned} \tag{19}$$

The posterior distribution of $\boldsymbol{\beta}$ given $\{\boldsymbol{X}_{1:t}, \boldsymbol{A}_{1:t}, \boldsymbol{R}_{1:t}\}$ can be derived as

$$f\big(\boldsymbol{\beta}|\{\boldsymbol{X}_{1:t}, \boldsymbol{A}_{1:t}, \boldsymbol{R}_{1:t}\}\big) \propto f(\boldsymbol{R}_{1:t}|\boldsymbol{\beta}, \widetilde{\boldsymbol{X}}_{1:t}) \cdot \pi(\boldsymbol{\beta}),$$

After simple calculations, the posterior mean and variance for $\boldsymbol{\beta}$ can be obtained by

$$\begin{aligned}
\boldsymbol{\Sigma}_{t,\text{post}}^{-1} &\leftarrow \boldsymbol{\Sigma}_0^{-1} + \sum_{i,t} \widetilde{\boldsymbol{X}}_{ti} \widetilde{\boldsymbol{X}}_{ti}^\top / \sigma^2, \\
\boldsymbol{\beta}_{t,\text{post}} &\leftarrow \boldsymbol{\Sigma}_{t,\text{post}} \Big\{ \boldsymbol{\Sigma}_0^{-1} \boldsymbol{\mu}_0 + \sum_{i,t} R_{ti} \widetilde{\boldsymbol{X}}_{ti} / \sigma^2 \Big\}.
\end{aligned} \tag{20}$$

Suppose $v$ is a hyper-parameter deciding the level of exploration in TS. For unit $i$ at round $t$, LinTSWI will sample $\widetilde{\boldsymbol{\beta}}_{ti} \sim \mathcal{N}(\boldsymbol{\beta}_{t,\text{post}}, v^2 \boldsymbol{\Sigma}_{t,\text{post}}^{-1})$ and then choose arm $a_{ti}$ such that

$$a_{ti} = \arg\max_a \omega_{ti} \boldsymbol{X}_{ti}^\top \widetilde{\boldsymbol{\beta}}_{ti,a}. \tag{21}$$

# D. Optimal Value Function Estimators: A Detailed Introduction

In Section 4.4 of the main paper, we present the final doubly robust (DR) estimator $\widehat{V}_t^{\text{DR}}$ and establish its asymptotic normality in Theorem 4.4. Here, we offer a detailed introduction to three commonly used estimators in causal inference and elaborate on the construction of the DR estimator.

As detailed in the main paper, our goal lies in providing different estimates for $V^{\pi^*}$, where

$$V^{\pi^*} = \mathbb{E}\left[ \pi^*(\boldsymbol{X}) \omega \boldsymbol{\beta}_1^\top \boldsymbol{X} + (1 - \pi^*(\boldsymbol{X})) \omega \boldsymbol{\beta}_0^\top \boldsymbol{X} \right].$$

The first estimator we propose to estimate $V^{\pi^*}$ is the Inverse Probability Weighting (IPW) estimator, also known as the Importance Sampling (IS) estimator in reinforcement learning. The core idea is to use the propensity ratio, $\frac{\mathbf{1}\{a_{ti}=\pi^*(\boldsymbol{X}_{ti})\}}{\mathbb{P}\{a_{ti}=\pi^*(\boldsymbol{X}_{ti})\}}$, to adjust for distribution shifts caused by exploration. However, since the true values of $\pi^*$ and $\mathbb{P}\{a_{ti} = \pi^*(\boldsymbol{X}_{ti})\}$ are unknown, we replace them with their corresponding sample estimates. Therefore,

$$\widehat{V}_t^{\text{IPW}} = \frac{1}{\bar{N}_t} \sum_{s=1}^t \sum_{i=1}^{N_s} \frac{\mathbf{1}\{a_{si} = \widehat{\pi}_{s-1}(\boldsymbol{X}_{si})\}}{1 - \widehat{\kappa}_{s-1}(\omega_{si}, \boldsymbol{X}_{si})} \cdot r_{si},$$

where $\hat{\kappa}_{t-1}(\omega_{ti}, \boldsymbol{X}_{ti}) = \sum_{s \leq t-1, i \in [N_s]} \mathbf{1}\{A_{si} \neq \widehat{\pi}(\boldsymbol{X}_{si})\}/\bar{N}_{t-1}$, and $\widehat{\pi}_{s-1}(\boldsymbol{X}_{si})$ is obtained from Line 9 of Algorithm 1.

The second estimator we propose is the Direct Method (DM). The concept is straightforward: we substitute the unknown true values, such as $\pi^*$ and $\boldsymbol{\beta}$, with their sample estimates in the optimal value function $V^{\pi^*}$ to directly estimate the optimal reward. Thus,

$$\widehat{V}_t^{\text{DM}} = \frac{1}{\bar{N}_t} \sum_{s=1}^t \sum_{i=1}^{N_s} \sum_{a \in \mathcal{A}} \mathbf{1}\{\widehat{\pi}_{s-1}(\boldsymbol{X}_{si}) = a\} \omega_{si} \boldsymbol{X}_{si}^\top \widehat{\boldsymbol{\beta}}_{s-1,a}.$$

Following the same logic used to derive Equation (2), the above estimator can be rewritten as

$$\widehat{V}_t^{\text{DM}} = \frac{1}{\bar{N}_t} \sum_{s=1}^t \sum_{i=1}^{N_s} \widehat{\mu}_{s-1}^{(i,s)}(\boldsymbol{X}_s, \widehat{\pi}_{s-1}(\boldsymbol{X}_s)),$$

where $\widehat{\mu}_{t-1}^{(t,i)}(\boldsymbol{X}_t, \boldsymbol{A}_t) = \widetilde{\boldsymbol{X}}_{ti}\widehat{\boldsymbol{\beta}}_{t-1}$ denotes the expected reward that unit $i$ can obtain given the covariate information and actions of all units at round $t$.

By combining the two estimators mentioned above, we derive the doubly robust (DR) estimator, where

$$\widehat{V}_t^{\mathrm{DR}} = \frac{1}{\bar{N}_t} \sum_{s=1}^{t} \sum_{i=1}^{N_s} \left[ \frac{\mathbf{1}\{a_{si} = \widehat{\pi}_{s-1}(\boldsymbol{X}_{si})\}}{1 - \hat{\kappa}_{s-1}(\omega_{ti}, \boldsymbol{X}_{si})} \cdot \left\{ r_{si} - \widehat{\mu}_{s-1}^{(i,s)}(\boldsymbol{X}_s, \widehat{\pi}_{s-1}(\boldsymbol{X}_s)) \right\} + \widehat{\mu}_{s-1}^{(i,s)}(\boldsymbol{X}_s, \widehat{\pi}_{s-1}(\boldsymbol{X}_s)) \right].$$

In $\widehat{V}_t^{\mathrm{DR}}$, the second term, i.e., $\widehat{\mu}_{s-1}^{(i,s)}(\boldsymbol{X}_s, \widehat{\pi}_{s-1}(\boldsymbol{X}_s))$, corresponds to the direct estimator. The first term involving the propensity ratio is an augmentation term derived from the IPW estimator, which provides additional protection against model misspecifications, thereby ensuring double robustness. Specifically, as long as either the propensity score model $\hat{\kappa}_{t-1}$ or the outcome regression model $\widehat{\mu}_{t-1}^{(t,i)}$ is correctly specified, $\widehat{V}_t^{\mathrm{DR}}$ becomes a consistent estimator of the optimal value function $V^{\pi^*}$. Furthermore, under Assumption A.4 (which is demonstrated to be quite mild in our setting in Appendix A), the DR estimator achieves asymptotic normality as the total number of units, $\bar{N}_t$, approaches infinity. Details are summarized in Theorem 4.4.

## E. Sensitivity Analysis on Misspecified Interference Matrix

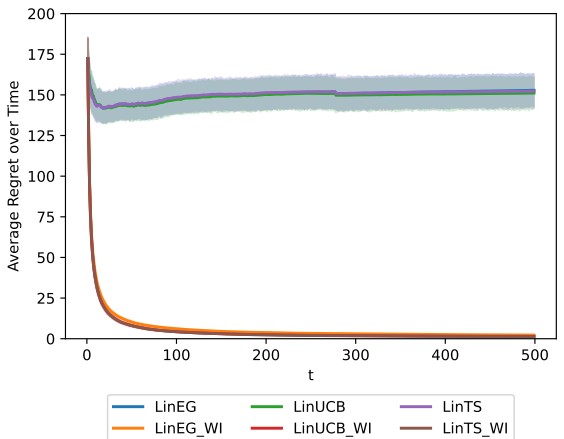
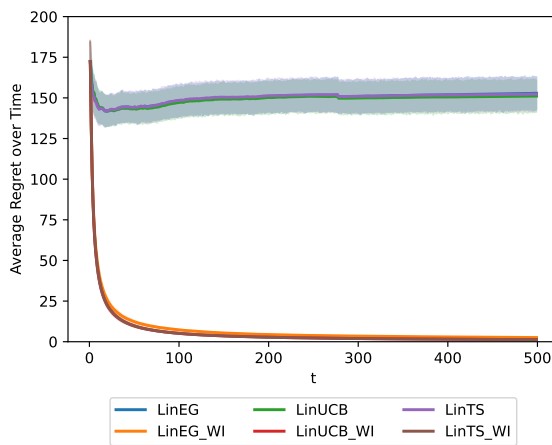

*Figure 7.* Average regret comparison when perturbation parameter $b = 0$ (i.e., no misspecification, left) and $b = 0.5$ (right)

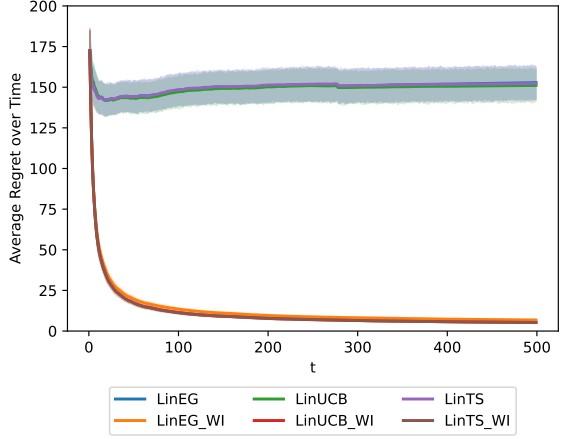
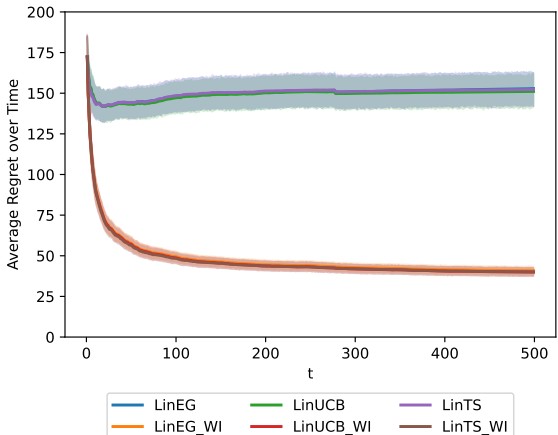

*Figure 8.* Average regret comparison when perturbation parameter $b = 1$ (left) and $b = 2$ (right)

In this section, we provide the full set of sensitivity analysis results omitted from Section 5.4 due to space constraints. As introduced in the main paper, we evaluate the impact of interference misspecification by introducing a perturbed matrix for LinCBWI, defined as $\breve{W}_t = W_t + \Xi_t$. Here, $\Xi_t = \{\xi_{ij}\}_{1 \le i,j \le N_t}$ denotes a perturbation matrix with $\xi_{ij} \sim \text{Unif}(-b, b)$. A larger value of $b$ indicates a more severe level of misspecification. We consider $b \in \{0, 0.5, 1, 2\}$ to assess LinCBWI's robustness to such perturbations.

The results are presented in Figures 7–8. When $b = 0$, i.e., no misspecification, the result exactly matches Figure 2. As $b$ increases, we observe a modest degradation in LinCBWI's performance: the average regret moves further from zero, as seen in the right panel of Figure 8 for $b = 2$. Nonetheless, even under this severe level of perturbation, LinCBWI consistently outperforms classical LinCB. These results suggest that incorporating interference structures, even when estimated with considerable error, improves decision-making performance and provides valuable modeling flexibility in the bandit framework.

Lastly, we consider an extreme scenario in which the interference matrix used by LinCBWI is entirely uninformative. Specifically, we generate $\breve{W}_t \sim \text{Unif}(-1, 1)$, meaning each entry of $\breve{W}_t$ is independently drawn from a uniform distribution and bears no relation to the true interference matrix $W_t$. In this case, LinCBWI performs comparably to classical LinCB and even slightly better, as shown in Figure 9. We attribute this marginal improvement to potential overfitting introduced by the more complex modeling structure of LinCBWI.

Importantly, this finding demonstrates that even when the interference structure is completely misspecified, LinCBWI does not suffer from performance degradation relative to LinCB. Taken together with the results in Figures 7–8, this highlights the robustness and stability of our proposed method in the presence of misspecified or uninformative interference information.

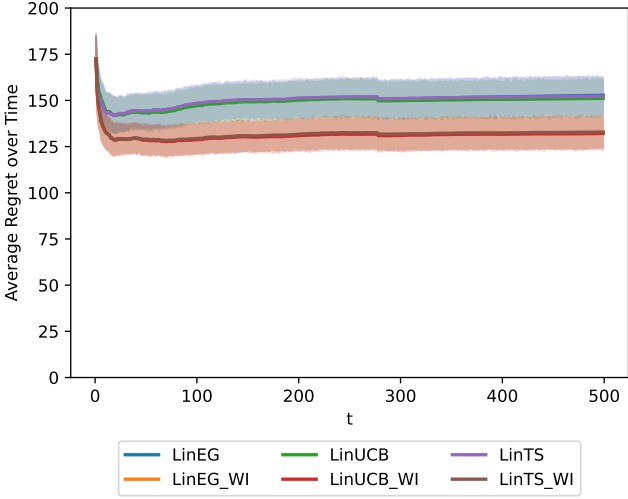

*Figure 9.* Average regret comparison when the interference matrix used in LinCBWI is completely uninformative, i.e., $\breve{W}_t \sim \text{Unif}(-1, 1)$

## F. Proof of Theorem 4.1: the Upper Bound of the Online OLS Estimator

The proof of this theorem was originally presented in Bastani & Bayati (2020). Here, we provide a slightly modified version to fit our specific context with interference. Define $\widetilde{\Sigma}_t = \frac{1}{\bar{N}_t} \sum_{s=1}^t \sum_{i=1}^{N_s} \widetilde{X}_{si} \widetilde{X}_{si}^\top$. According to the definition of $\widehat{\beta}_t$,

$$\|\widehat{\beta}_t - \beta\|_2 = \left\| \widetilde{\Sigma}_t^{-1} \cdot \left\{ \frac{1}{\bar{N}_t} \sum_{s=1}^t \sum_{i=1}^{N_s} \widetilde{X}_{si} \eta_{si} \right\} \right\|_2 \le \left\| \widetilde{\Sigma}_t^{-1} \right\|_2 \cdot \left\| \left\{ \frac{1}{\bar{N}_t} \sum_{s=1}^t \sum_{i=1}^{N_s} \widetilde{X}_{si} \eta_{si} \right\} \right\|_2.$$

Since $\widetilde{\Sigma}_t$ is a symmetric positive semi-definite matrix, we have

$$\left\| \widetilde{\Sigma}_t^{-1} \right\|_2 = \lambda_{\max}\left( \widetilde{\Sigma}_t^{-1} \right) = \left\{ \lambda_{\min}(\widetilde{\Sigma}_t) \right\}^{-1},$$

where the right hand side of the above equation, by Assumption A.1-A.2, is lower bounded by $p_t \lambda$. Therefore,

$$\|\widehat{\boldsymbol{\beta}}_t - \boldsymbol{\beta}\|_2 \leq \left\|\widetilde{\Sigma}_t^{-1}\right\|_2 \cdot \left\|\frac{1}{\bar{N}_t}\sum_{s=1}^{t}\sum_{i=1}^{N_s}\widetilde{\boldsymbol{X}}_{si}\eta_{si}\right\|_2 = \left\{\lambda_{\min}(\widetilde{\Sigma}_t)\right\}^{-1}\cdot\left\|\frac{1}{\bar{N}_t}\sum_{s=1}^{t}\sum_{i=1}^{N_s}\widetilde{\boldsymbol{X}}_{si}\eta_{si}\right\|_2 \leq \frac{1}{\bar{N}_t p_t \lambda}\left\|\sum_{s=1}^{t}\sum_{i=1}^{N_s}\widetilde{\boldsymbol{X}}_{si}\eta_{si}\right\|_2.$$

Define the $l$th element of $\widetilde{\boldsymbol{X}}_{ti}$ as $\widetilde{X}_{ti,l}$, where $l = 1, \ldots, dK$, with $d$ denoting the dimension of covariates and $K$ denoting the number of arms. For any $h > 0$,

$$\begin{aligned}
\mathbb{P}\left(\|\widehat{\boldsymbol{\beta}}_t - \boldsymbol{\beta}\|_2 \leq h\right) &\geq \mathbb{P}\left(\left\|\sum_{s=1}^{t}\sum_{i=1}^{N_s}\widetilde{\boldsymbol{X}}_{si}\eta_{si}\right\|_2 \leq h\bar{N}_t p_t \lambda\right) \\
&\geq \mathbb{P}\left(\left|\sum_{s=1}^{t}\sum_{i=1}^{N_s}\widetilde{X}_{si,1}\eta_{si}\right| \leq \frac{h\bar{N}_t p_t \lambda}{\sqrt{dK}}, \ldots, \left|\sum_{s=1}^{t}\sum_{i=1}^{N_s}\widetilde{X}_{si,dK}\eta_{si}\right| \leq \frac{h\bar{N}_t p_t \lambda}{\sqrt{dK}}\right) \\
&= 1 - \mathbb{P}\left(\bigcup_{l=1}^{dK}\left\{\left|\sum_{s=1}^{t}\sum_{i=1}^{N_s}\widetilde{X}_{si,l}\eta_{si}\right| > \frac{h\bar{N}_t p_t \lambda}{\sqrt{dK}}\right\}\right).
\end{aligned} \tag{22}$$

To proceed with deriving the lower bound of the above equation, we will utilize Lemma 1 from Chen et al. (2021). As this lemma is directly applicable to our context, we will state it here and refer readers to the original paper for the proof.

**Lemma F.1.** *Suppose* $\{\mathcal{F}_q : q = 1, \ldots, \bar{N}_T\}$ *is an increasing filtration of* $\sigma-$*fields. Let* $\{Z_q : q = 1, \ldots, \bar{N}_T\}$ *be a sequence of random variables such that* $Z_q$ *is* $\mathcal{F}_{q-1}-$*measurable and* $|Z_q| \leq L$. *Let* $\eta_q : q = 1, \ldots, \bar{N}_T$ *be independent* $\sigma-$*gaussian, and* $\eta_q \perp \mathcal{F}_{q-1}$ *for all* $q$. *Let* $\mathcal{S} = \{s_1, \ldots, s_{|\mathcal{S}|}\} \subseteq \{1, \ldots, \bar{N}_T\}$ *be an index set where* $|\mathcal{S}|$ *is the number of elements in* $\mathcal{S}$. *Then for any* $h > 0$,

$$\mathbb{P}\left(\sum_{s\in\mathcal{S}} Z_s \eta_s \geq h\right) \leq \exp\left\{-\frac{h^2}{2|\mathcal{S}|\sigma^2 L^2}\right\}. \tag{23}$$

In our context, we flatten the unit $\{t, i\}_{1\leq t\leq T, 1\leq i\leq N_t}$ to a unit queue $Q(t, i) = \sum_{s=1}^{t-1} N_s + i$, such that all of the units are measured in a chronological order. Notice that the specific order of units within a round does not matter, as estimation and parameter updates occur only after all units in the round have been observed. As such, we also use $\widetilde{X}_{q,l}$ to denote the $l$th element of $\widetilde{\boldsymbol{X}}_{ti}$. To use Lemma F.1, we define a filtration $\mathcal{F}_q$ as

$$\mathcal{F}_q = \sigma(\widetilde{X}_{1,l}\eta_1, \ldots, \widetilde{X}_{q,l}\eta_q),$$

which satisfies $\eta_q \perp \mathcal{F}_{q-1}$ for any $q \in \{1, \ldots, \bar{N}_T\}$. Let $Z_q = \widetilde{X}_{q,j}$. Then by Assumption A.1.b-c,

$$|\widetilde{X}_{q,l}|^2 \leq \|\widetilde{\boldsymbol{X}}_q\|_2^2 \leq K\left\|\sum_{j=1}^{N_t} W_{t,ij}\boldsymbol{X}_{tj}\right\|_2^2 \leq KL_x^2 d \cdot \left(\sum_j |W_{t,ij}|\right)^2 \leq dKL_w^2 L_x^2.$$

Define $L := \sqrt{dK}L_w L_x$, so that $|\widetilde{X}_{q,l}| \leq L$. According to the conclusion of Lemma F.1, we have

$$\mathbb{P}\left(\left|\sum_{s=1}^{t}\sum_{i=1}^{N_s}\widetilde{X}_{si,l}\eta_{si}\right| \geq \frac{h\bar{N}_t p_t \lambda}{\sqrt{dK}}\right) \leq \exp\left\{-\frac{h^2\bar{N}_t p_t^2 \lambda^2}{2d^2 K^2 \sigma^2 L_w^2 L_x^2}\right\}. \tag{24}$$

Combining the result of Equation (22) and (24), we have

$$\begin{aligned}
\mathbb{P}\left(\|\widehat{\boldsymbol{\beta}}_t - \boldsymbol{\beta}\|_2 \leq h\right) &\geq 1 - \sum_{l=1}^{dK}\mathbb{P}\left(\left|\sum_{s=1}^{t}\sum_{i=1}^{N_s}\widetilde{X}_{si,l}\eta_{si}\right| > \frac{h\bar{N}_t p_t \lambda}{\sqrt{dK}}\right) \geq 1 - \sum_{l=1}^{dK}\exp\left\{-\frac{h^2\bar{N}_t p_t^2 \lambda^2}{2d^2 K^2 \sigma^2 L_w^2 L_x^2}\right\} \\
&= 1 - dK\exp\left\{-\frac{h^2\bar{N}_t p_t^2 \lambda^2}{2d^2 K^2 \sigma^2 L_w^2 L_x^2}\right\}.
\end{aligned}$$

Lastly, since $\|\boldsymbol{v}\|_1 \leq \sqrt{dK}\|\boldsymbol{v}\|_2$ for any $\boldsymbol{v} \in \mathbb{R}^{dK}$, we have

$$\mathbb{P}\left(\|\widehat{\boldsymbol{\beta}}_t - \boldsymbol{\beta}\|_1 \leq h\right) \geq \mathbb{P}\left(\|\widehat{\boldsymbol{\beta}}_t - \boldsymbol{\beta}\|_2 \leq \frac{h}{\sqrt{dK}}\right) \geq 1 - dK \exp\left\{-\frac{h^2 \bar{N}_t p_t^2 \lambda^2}{2d^3 K^3 \sigma^2 L_w^2 L_x^2}\right\}.$$

The proof of Theorem 4.1 is thus complete.

## G. Proof of Theorem 4.2: the Upper Bound of Exploration

### G.1. Probability of exploration for UCB

In this subsection, we aim to prove that the upper bound of $\kappa_{ti}$ for UCB algorithm satisfies

$$\kappa_{ti}(\omega_{ti}, \boldsymbol{X}_{ti}) \leq CK(K-1)\left(\frac{2\alpha L_w L_x}{\sqrt{\bar{N}_{t-1}p_{t-1}\lambda}} + \xi\right)^\gamma + dK^2(K-1)\exp\left\{-\frac{\xi^2 \bar{N}_{t-1}p_{t-1}^2\lambda^2}{8d^3 K^3 \sigma^2 L_w^4 L_x^4}\right\}. \tag{25}$$

We split the proof into three steps.

**Step 1:** Rewrite $\kappa_{ti}(\omega_{ti}, \boldsymbol{X}_{ti}) = \sum_{1 \leq a,a' \leq K, a \neq a'} \frac{1}{2}\mathbb{P}\left(|\omega_{ti}\boldsymbol{X}_{ti}^\top(\widehat{\boldsymbol{\beta}}_{t-1,a} - \widehat{\boldsymbol{\beta}}_{t-1,a'})| < \alpha\{\widehat{\sigma}_{ta'}(\boldsymbol{X}_{ti}) - \widehat{\sigma}_{ta}(\boldsymbol{X}_{ti})\}\right)$.

For clarity, we first consider the case where $K = 2$ and then extend the formula to the general case of $K$. When $K = 2$,

$$\begin{aligned}
\kappa_{ti}(\omega_{ti}, \boldsymbol{X}_{ti}) &= \mathbb{P}(A_{ti} \neq \widehat{\pi}_{t-1}(\boldsymbol{X}_{ti})) \\
&= \underbrace{\mathbb{P}(A_{ti} \neq \widehat{\pi}_{t-1}(\boldsymbol{X}_{ti}), \widehat{\pi}_{t-1}(\boldsymbol{X}_{ti}) = 1)}_{\delta_1} + \underbrace{\mathbb{P}(A_{ti} \neq \widehat{\pi}_{t-1}(\boldsymbol{X}_{ti}), \widehat{\pi}_{t-1}(\boldsymbol{X}_{ti}) = 0)}_{\delta_0}.
\end{aligned}$$

We first consider $\delta_1$. Based on the exploration criteria of UCB algorithm, we have

$$A_{ti} = \mathbf{1}\left\{\omega_{ti}\boldsymbol{X}_{ti}^\top\widehat{\boldsymbol{\beta}}_{t-1,1} + \alpha\widehat{\sigma}_{t-1,1}(\boldsymbol{X}_{ti}) > \omega_{ti}\boldsymbol{X}_{ti}^\top\widehat{\boldsymbol{\beta}}_{t-1,0} + \alpha\widehat{\sigma}_{t-1,0}(\boldsymbol{X}_{ti})\right\},$$

where $\widehat{\sigma}_{t-1,a}(\boldsymbol{X}_{ti}) = |\omega_{ti}|\sqrt{\boldsymbol{X}_{ti}^\top\left(\widetilde{\boldsymbol{X}}_{1:(t-1)}^\top\widetilde{\boldsymbol{X}}_{1:(t-1)}\right)_{aa}^{-1}\boldsymbol{X}_{ti}}$, with $\left(\widetilde{\boldsymbol{X}}_{1:(t-1)}^\top\widetilde{\boldsymbol{X}}_{1:(t-1)}\right)_{aa}^{-1}$ denoting the $a$-th block-diagonal matrix of dimension $d \times d$ that quantifies the inverse covariance associated with arm $a$. Thus, given that $\widehat{\pi}_{t-1}(\boldsymbol{X}_{ti}) = 1$, i.e., $\omega_{ti}\boldsymbol{X}_{ti}^\top(\widehat{\boldsymbol{\beta}}_{t-1,1} - \widehat{\boldsymbol{\beta}}_{t-1,0}) > 0$, we have

$$\begin{aligned}
\delta_1 &= \mathbb{P}(A_{ti} \neq \widehat{\pi}_{t-1}(\boldsymbol{X}_{ti}), \widehat{\pi}_{t-1}(\boldsymbol{X}_{ti}) = 1) \\
&= \mathbb{P}(\omega_{ti}\boldsymbol{X}_{ti}^\top\widehat{\boldsymbol{\beta}}_{t-1,1} + \alpha\widehat{\sigma}_{t-1,1}(\boldsymbol{X}_{ti}) < \omega_{ti}\boldsymbol{X}_{ti}^\top\widehat{\boldsymbol{\beta}}_{t-1,0} + \alpha\widehat{\sigma}_{t-1,0}(\boldsymbol{X}_{ti}), \omega_{ti}\boldsymbol{X}_{ti}^\top\widehat{\boldsymbol{\beta}}_{t-1,1} > \omega_{ti}\boldsymbol{X}_{ti}^\top\widehat{\boldsymbol{\beta}}_{t-1,0}) \\
&= \mathbb{P}\left(0 < \omega_{ti}\boldsymbol{X}_{ti}^\top(\widehat{\boldsymbol{\beta}}_{t-1,1} - \widehat{\boldsymbol{\beta}}_{t-1,0}) < \alpha\{\widehat{\sigma}_{t-1,0}(\boldsymbol{X}_{ti}) - \widehat{\sigma}_{t-1,1}(\boldsymbol{X}_{ti})\}\right).
\end{aligned} \tag{26}$$

Similarly, we have

$$\begin{aligned}
\delta_0 &= \mathbb{P}(A_{ti} \neq \widehat{\pi}_{t-1}(\boldsymbol{X}_{ti}), \widehat{\pi}_{t-1}(\boldsymbol{X}_{ti}) = 0) \\
&= \mathbb{P}(\omega_{ti}\boldsymbol{X}_{ti}^\top\widehat{\boldsymbol{\beta}}_{t-1,1} + \alpha\widehat{\sigma}_{t-1,1}(\boldsymbol{X}_{ti}) > \omega_{ti}\boldsymbol{X}_{ti}^\top\widehat{\boldsymbol{\beta}}_{t-1,0} + \alpha\widehat{\sigma}_{t-1,0}(\boldsymbol{X}_{ti}), \omega_{ti}\boldsymbol{X}_{ti}^\top\widehat{\boldsymbol{\beta}}_{t-1,1} < \omega_{ti}\boldsymbol{X}_{ti}^\top\widehat{\boldsymbol{\beta}}_{t-1,0}) \\
&= \mathbb{P}\left(\alpha\{\widehat{\sigma}_{t-1,0}(\boldsymbol{X}_{ti}) - \widehat{\sigma}_{t-1,1}(\boldsymbol{X}_{ti})\} < \omega_{ti}\boldsymbol{X}_{ti}^\top(\widehat{\boldsymbol{\beta}}_{t-1,1} - \widehat{\boldsymbol{\beta}}_{t-1,0}) < 0\right).
\end{aligned} \tag{27}$$

Combining the result of Equation (26) and (27), we have that when $K = 2$,

$$\kappa_{ti}(\omega_{ti}, \boldsymbol{X}_{ti}) = \delta_1 + \delta_0 = \mathbb{P}\left(|\omega_{ti}\boldsymbol{X}_{ti}^\top(\widehat{\boldsymbol{\beta}}_{t-1,1} - \widehat{\boldsymbol{\beta}}_{t-1,0})| < \alpha\{\widehat{\sigma}_{t-1,0}(\boldsymbol{X}_{ti}) - \widehat{\sigma}_{t-1,1}(\boldsymbol{X}_{ti})\}\right). \tag{28}$$

Now let's consider a general $K$. Similarly,

$$\kappa_{ti}(\omega_{ti}, \boldsymbol{X}_{ti}) = \mathbb{P}(A_{ti} \neq \widehat{\pi}_{t-1}(\boldsymbol{X}_{ti})) = \sum_{a=1}^K \sum_{a'=1}^K \mathbb{P}(A_{ti} \neq \widehat{\pi}_{t-1}(\boldsymbol{X}_{ti}), A_{ti} = a, \widehat{\pi}_{t-1}(\boldsymbol{X}_{ti}) = a').$$

Notice that when $a = a'$, the probability $\mathbb{P}(A_{ti} \neq \widehat{\pi}_{t-1}(\boldsymbol{X}_{ti}), A_{ti} = a, \widehat{\pi}_{t-1}(\boldsymbol{X}_{ti}) = a')$ equals 0. For any $a \neq a'$, we can treat them as arms 1 and 0 and repeat the proof for $K = 2$, as shown above. Consequently, for a general $K$, Equation (28) can be extended to

$$\kappa_{ti}(\omega_{ti}, \boldsymbol{X}_{ti}) = \sum_{\substack{1 \leq a, a' \leq K \\ a \neq a'}} \frac{1}{2} \mathbb{P}\left(|\omega_{ti}\boldsymbol{X}_{ti}^{\top}(\widehat{\boldsymbol{\beta}}_{t-1,a} - \widehat{\boldsymbol{\beta}}_{t-1,a'})| < \alpha\{\widehat{\sigma}_{t-1,a'}(\boldsymbol{X}_{ti}) - \widehat{\sigma}_{t-1,a}(\boldsymbol{X}_{ti})\}\right). \tag{29}$$

**Step 2:** Bound the variance term $\widehat{\sigma}_{t-1,a}(\boldsymbol{X}_{ti})$.

For any $a \in [K]$, notice that

$$\widehat{\sigma}_{t-1,a}(\boldsymbol{X}_{ti})^2 = \omega_{ti}^2 \boldsymbol{X}_{ti}^{\top}\left(\widetilde{\boldsymbol{X}}_{1:(t-1)}^{\top}\widetilde{\boldsymbol{X}}_{1:(t-1)}\right)_{aa}^{-1}\boldsymbol{X}_{ti} \leq \omega_{ti}^2\|\boldsymbol{X}_{ti}\|_2^2 \cdot \max_{\|\boldsymbol{v}\|_2=1} \boldsymbol{v}^T\left(\widetilde{\boldsymbol{X}}_{1:(t-1)}^{\top}\widetilde{\boldsymbol{X}}_{1:(t-1)}\right)_{aa}^{-1}\boldsymbol{v}$$

$$\leq \omega_{ti}^2\|\boldsymbol{X}_{ti}\|_2^2 \cdot \lambda_{\max}\left\{\left(\widetilde{\boldsymbol{X}}_{1:(t-1)}^{\top}\widetilde{\boldsymbol{X}}_{1:(t-1)}\right)_{aa}^{-1}\right\} \leq L_w^2 L_x^2 \cdot \lambda_{\max}\left\{\left(\widetilde{\boldsymbol{X}}_{1:(t-1)}^{\top}\widetilde{\boldsymbol{X}}_{1:(t-1)}\right)^{-1}\right\}$$

$$= \frac{L_w^2 L_x^2}{\lambda_{\min}\left\{\left(\widetilde{\boldsymbol{X}}_{1:(t-1)}^{\top}\widetilde{\boldsymbol{X}}_{1:(t-1)}\right)\right\}},$$

where the first inequality holds by the definition of eigenvalues, and the last inequality holds by Assumption A.1.b-c.

Furthermore, by Assumption A.1 and A.2,

$$\lambda_{\min}\left\{\left(\widetilde{\boldsymbol{X}}_{1:(t-1)}^{\top}\widetilde{\boldsymbol{X}}_{1:(t-1)}\right)\right\} = \bar{N}_{t-1} \cdot \lambda_{\min}\left\{\frac{1}{\bar{N}_{t-1}}\sum_{s=1}^{t-1}\sum_{i=1}^{N_t}\widetilde{\boldsymbol{X}}_{si}\widetilde{\boldsymbol{X}}_{si}^{\top}\right\} > \bar{N}_{t-1} \cdot p_{t-1} \cdot \lambda_{\min}(\Sigma) \geq \bar{N}_{t-1} \cdot p_{t-1}\lambda.$$

Combining the result above to the expression of $\widehat{\sigma}_{t-1,a}(\boldsymbol{X}_{ti})$, we can further derive

$$\widehat{\sigma}_{t-1,a}(\boldsymbol{X}_{ti}) \leq \frac{L_w L_x}{\sqrt{\lambda_{\min}\left\{\left(\widetilde{\boldsymbol{X}}_{1:(t-1)}^{\top}\widetilde{\boldsymbol{X}}_{1:(t-1)}\right)\right\}}} \leq \frac{L_w L_x}{\sqrt{\bar{N}_{t-1}p_{t-1}\lambda}}. \tag{30}$$

Therefore, for any $1 \leq a, a' \leq K$ and $a \neq a'$,

$$\alpha|\widehat{\sigma}_{t-1,a}(\boldsymbol{X}_{ti}) - \widehat{\sigma}_{t-1,a'}(\boldsymbol{X}_{ti})| \leq \alpha\left\{|\widehat{\sigma}_{t-1,a}(\boldsymbol{X}_{ti})| + |\widehat{\sigma}_{t-1,a'}(\boldsymbol{X}_{ti})|\right\} \leq \frac{2\alpha L_w L_x}{\sqrt{\bar{N}_{t-1}p_{t-1}\lambda}}.$$

Combining the result above and Equation (29), we have

$$\kappa_{ti}(\omega_{ti}, \boldsymbol{X}_{ti}) \leq \frac{K(K-1)}{2} \cdot \mathbb{P}\left(|\omega_{ti}\boldsymbol{X}_{ti}^{\top}(\widehat{\boldsymbol{\beta}}_{t-1,a} - \widehat{\boldsymbol{\beta}}_{t-1,a'})| < \alpha|\widehat{\sigma}_{t-1,a'}(\boldsymbol{X}_{ti}) - \widehat{\sigma}_{t-1,a}(\boldsymbol{X}_{ti})|\right)$$

$$\leq \frac{K(K-1)}{2} \cdot \mathbb{P}\left(|\omega_{ti}\boldsymbol{X}_{ti}^{\top}(\widehat{\boldsymbol{\beta}}_{t-1,a} - \widehat{\boldsymbol{\beta}}_{t-1,a'})| < \frac{2\alpha L_w L_x}{\sqrt{\bar{N}_{t-1}p_{t-1}\lambda}}\right) \tag{31}$$

**Step 3:** Further bound the RHS of Equation (31).

For brevity, we denote $\widehat{\zeta}_{ti} = \omega_{ti}\boldsymbol{X}_{ti}^{\top}(\widehat{\boldsymbol{\beta}}_{t-1,a} - \widehat{\boldsymbol{\beta}}_{t-1,a'})$ and $\zeta_{ti} = \omega_{ti}\boldsymbol{X}_{ti}^{\top}(\boldsymbol{\beta}_a - \boldsymbol{\beta}_{a'})$. Note that in $\widehat{\zeta}_{ti}$ and $\zeta_{ti}$, we omit the dependence on the arms $(a, a')$ when there is no ambiguity, as they simply represent any two different arms in $[K]$.

For any $\xi > 0$, define a event $E := \{|\widehat{\zeta}_{ti} - \zeta_{ti}| \leq \xi\}$. By Holder's inequality and $\|\boldsymbol{v}\|_{\infty} \leq \|\boldsymbol{v}\|_2$, we have

$$|\omega_{ti}\boldsymbol{X}_{ti}^{\top}\widehat{\boldsymbol{\beta}}_{t-1,a} - \omega_{ti}\boldsymbol{X}_{ti}^{\top}\boldsymbol{\beta}_a| \leq \left\|\omega_{ti}\boldsymbol{X}_{ti}\right\|_{\infty}\left\|\widehat{\boldsymbol{\beta}}_{t-1,a} - \boldsymbol{\beta}_a\right\|_1 \leq L_w\|\boldsymbol{X}_{ti}\|_2\left\|\widehat{\boldsymbol{\beta}}_{t-1,a} - \boldsymbol{\beta}_a\right\|_1 \leq L_w L_x\left\|\widehat{\boldsymbol{\beta}}_{t-1,a} - \boldsymbol{\beta}_a\right\|_1.$$

According to Theorem 4.1,

$$\mathbb{P}\{|\omega_{ti}\boldsymbol{X}_{ti}^{\top}\widehat{\boldsymbol{\beta}}_{t-1,a} - \omega_{ti}\boldsymbol{X}_{ti}^{\top}\boldsymbol{\beta}_a| > \xi\} \leq \mathbb{P}\{L_w L_x \|\widehat{\boldsymbol{\beta}}_{t-1,a} - \boldsymbol{\beta}_a\|_1 > \xi\} = \mathbb{P}\{\|\widehat{\boldsymbol{\beta}}_{t-1,a} - \boldsymbol{\beta}_a\|_1 > \frac{\xi}{L_w L_x}\}$$

$$\leq \mathbb{P}\left(\|\widehat{\boldsymbol{\beta}}_{t-1} - \boldsymbol{\beta}\|_1 > \frac{\xi}{L_w L_x}\right) \leq dK \exp\left\{-\frac{\xi^2 \bar{N}_{t-1} p_{t-1}^2 \lambda^2}{2d^3 K^3 \sigma^2 L_w^4 L_x^4}\right\}.$$

By the triangle inequality,

$$|\widehat{\zeta}_{ti} - \zeta_{ti}| = \left|\{\omega_{ti}\boldsymbol{X}_{ti}^{\top}\widehat{\boldsymbol{\beta}}_{t-1,a} - \omega_{ti}\boldsymbol{X}_{ti}^{\top}\boldsymbol{\beta}_a\} - \{\omega_{ti}\boldsymbol{X}_{ti}^{\top}\widehat{\boldsymbol{\beta}}_{t-1,a'} - \omega_{ti}\boldsymbol{X}_{ti}^{\top}\boldsymbol{\beta}_{a'}\}\right|$$

$$\leq \left|\omega_{ti}\boldsymbol{X}_{ti}^{\top}\widehat{\boldsymbol{\beta}}_{t-1,a} - \omega_{ti}\boldsymbol{X}_{ti}^{\top}\boldsymbol{\beta}_a\right| + \left|\omega_{ti}\boldsymbol{X}_{ti}^{\top}\widehat{\boldsymbol{\beta}}_{t-1,a'} - \omega_{ti}\boldsymbol{X}_{ti}^{\top}\boldsymbol{\beta}_{a'}\right|.$$

Thus, for $|\widehat{\zeta}_{ti} - \zeta_{ti}|$, we have

$$\mathbb{P}(|\widehat{\zeta}_{ti} - \zeta_{ti}| > \xi) \leq \mathbb{P}\left(\left|\omega_{ti}\boldsymbol{X}_{ti}^{\top}\widehat{\boldsymbol{\beta}}_{t-1,a} - \omega_{ti}\boldsymbol{X}_{ti}^{\top}\boldsymbol{\beta}_a\right| + \left|\omega_{ti}\boldsymbol{X}_{ti}^{\top}\widehat{\boldsymbol{\beta}}_{t-1,a'} - \omega_{ti}\boldsymbol{X}_{ti}^{\top}\boldsymbol{\beta}_{a'}\right| > \xi\right)$$

$$\leq \mathbb{P}\left(\left|\omega_{ti}\boldsymbol{X}_{ti}^{\top}\widehat{\boldsymbol{\beta}}_{t-1,a} - \omega_{ti}\boldsymbol{X}_{ti}^{\top}\boldsymbol{\beta}_a\right| > \xi/2\right) + \mathbb{P}\left(\left|\omega_{ti}\boldsymbol{X}_{ti}^{\top}\widehat{\boldsymbol{\beta}}_{t-1,a'} - \omega_{ti}\boldsymbol{X}_{ti}^{\top}\boldsymbol{\beta}_{a'}\right| > \xi/2\right)$$

$$\leq dK \exp\left\{-\frac{\xi^2 \bar{N}_{t-1} p_{t-1}^2 \lambda^2}{8d^3 K^3 \sigma^2 L_w^4 L_x^4}\right\} + dK \exp\left\{-\frac{\xi^2 \bar{N}_{t-1} p_{t-1}^2 \lambda^2}{8d^3 K^3 \sigma^2 L_w^4 L_x^4}\right\}$$

$$= 2dK \exp\left\{-\frac{\xi^2 \bar{N}_{t-1} p_{t-1}^2 \lambda^2}{8d^3 K^3 \sigma^2 L_w^4 L_x^4}\right\}.$$

Therefore, event $E$ satisfies

$$\mathbb{P}(E) \geq 1 - 2dK \exp\left\{-\frac{\xi^2 \bar{N}_{t-1} p_{t-1}^2 \lambda^2}{8d^3 K^3 \sigma^2 L_w^4 L_x^4}\right\}. \tag{32}$$

On event $E$, we have $|\widehat{\zeta}_{ti}| \geq |\zeta_{ti}| - |\widehat{\zeta}_{ti} - \zeta_{ti}| \geq |\zeta_{ti}| - \xi$. Then going back to Equation (31), we further have

$$\kappa_{ti}(\omega_{ti}, \boldsymbol{X}_{ti}) \leq \frac{K(K-1)}{2}\mathbb{P}\left(|\widehat{\zeta}_{ti}| < \frac{2\alpha L_w L_x}{\sqrt{\bar{N}_{t-1} p_{t-1}}\lambda}\right) \leq \frac{K(K-1)}{2}\left[\mathbb{P}\left\{|\widehat{\zeta}_{ti}| < \frac{2\alpha L_w L_x}{\sqrt{\bar{N}_{t-1} p_{t-1}}\lambda} \mid E\right\} + \mathbb{P}(E^c)\right]$$

$$\leq \frac{K(K-1)}{2}\mathbb{P}\left\{|\zeta_{ti}| - \xi < \frac{2\alpha L_w L_x}{\sqrt{\bar{N}_{t-1} p_{t-1}}\lambda}\right\} + dK^2(K-1)\exp\left\{-\frac{\xi^2 \bar{N}_{t-1} p_{t-1}^2 \lambda^2}{8d^3 K^3 \sigma^2 L_w^4 L_x^4}\right\}$$

$$\leq \frac{K(K-1)}{2}\mathbb{P}\left\{|\zeta_{ti}| < \frac{2\alpha L_w L_x}{\sqrt{\bar{N}_{t-1} p_{t-1}}\lambda} + \xi\right\} + dK^2(K-1)\exp\left\{-\frac{\xi^2 \bar{N}_{t-1} p_{t-1}^2 \lambda^2}{8d^3 K^3 \sigma^2 L_w^4 L_x^4}\right\}. \tag{33}$$

By definition, $|\zeta_{ti}| = |\omega_{ti}\boldsymbol{X}_{ti}^{\top}(\boldsymbol{\beta}_a - \boldsymbol{\beta}_{a'})| = |\omega_{ti}| \cdot |f(\boldsymbol{X}_{ti}, a) - f(\boldsymbol{X}_{ti}, a')|$. Since $W_{t,ii} = 1$, we always have $|\omega_{ti}| \geq 1$ for any round-unit pair $(t, i)$. Therefore, $|\zeta_{ti}| \geq |f(\boldsymbol{X}_{ti}, a) - f(\boldsymbol{X}_{ti}, a')|$. According to Assumption A.3, there exists some constant $\gamma$ such that $\mathbb{P}\left\{|\zeta_{ti}| < \frac{2\alpha L_w L_x}{\sqrt{\bar{N}_{t-1} p_{t-1}}\lambda} + \xi\right\} \leq \mathbb{P}\left\{|f(\boldsymbol{X}_{ti}, a) - f(\boldsymbol{X}_{ti}, a')| < \frac{2\alpha L_w L_x}{\sqrt{\bar{N}_{t-1} p_{t-1}}\lambda} + \xi\right\} \leq O\left\{\left(\frac{2\alpha L_w L_x}{\sqrt{\bar{N}_{t-1} p_{t-1}}\lambda} + \xi\right)^{\gamma}\right\}$.

By taking this result back to Equation (33), we are able to show that there exists a constant $C$, such that

$$\kappa_{ti}(\omega_{ti}, \boldsymbol{X}_{ti}) \leq \frac{K(K-1)}{2}\mathbb{P}\left\{|\zeta_{ti}| < \frac{2\alpha L_w L_x}{\sqrt{\bar{N}_{t-1} p_{t-1}}\lambda} + \xi\right\} + dK^2(K-1)\exp\left\{-\frac{\xi^2 \bar{N}_{t-1} p_{t-1}^2 \lambda^2}{8d^3 K^3 \sigma^2 L_w^4 L_x^4}\right\}$$

$$\leq CK(K-1)\left(\frac{2\alpha L_w L_x}{\sqrt{\bar{N}_{t-1} p_{t-1}}\lambda} + \xi\right)^{\gamma} + dK^2(K-1)\exp\left\{-\frac{\xi^2 \bar{N}_{t-1} p_{t-1}^2 \lambda^2}{8d^3 K^3 \sigma^2 L_w^4 L_x^4}\right\}, \tag{34}$$

where $C$ is a large positive constant. The proof is thus complete.

## G.2. Probability of exploration for TS

In this subsection, we aim to prove that the upper bound of $\kappa_{ti}$ for TS algorithm satisfies

$$\kappa_{ti}(\omega_{ti}, \boldsymbol{X}_{ti}) \leq K(K-1) \exp\left(-\frac{\bar{N}_{t-1}p_{t-1}\lambda(|\zeta_{ti}| - \xi)^2}{8v^2 L_w^2 L_x^2}\right) + 2dK^2(K-1) \exp\left\{-\frac{\xi^2 \bar{N}_{t-1}p_{t-1}^2\lambda^2}{8d^3 K^3 \sigma^2 L_w^4 L_x^4}\right\}. \quad (35)$$

The proof can be split into three steps.

**Step 1:** Decompose $\kappa_{ti}(\omega_{ti}, \boldsymbol{X}_{ti})$ and bound it by $\mathbb{P}(E)$.

By definition,

$$\kappa_{ti}(\omega_{ti}, \boldsymbol{X}_{ti}) = \mathbb{P}(A_{ti} \neq \widehat{\pi}_{t-1}(\boldsymbol{X}_{ti})) = \sum_{a=1}^{K}\sum_{a'=1}^{K} \mathbb{P}(A_{ti} \neq \widehat{\pi}_{t-1}(\boldsymbol{X}_{ti}), A_{ti} = a, \widehat{\pi}_{t-1}(\boldsymbol{X}_{ti}) = a'),$$

where $a$ should be different from $a'$, otherwise the corresponding probability equals 0.

Similar to Step 3 of Section G.1, we define a event $E := \{|\widehat{\zeta}_{ti} - \zeta_{ti}| \leq \xi\}$ for any $\xi \in (0, |\zeta_{ti}|/2)$, where $\widehat{\zeta}_{ti} = \omega_{ti}\boldsymbol{X}_{ti}^\top(\widehat{\boldsymbol{\beta}}_{t-1,a} - \widehat{\boldsymbol{\beta}}_{t-1,a'})$, and $\zeta_{ti} = \omega_{ti}\boldsymbol{X}_{ti}^\top(\boldsymbol{\beta}_a - \boldsymbol{\beta}_{a'})$. According to the result of Equation (32), we have

$$\mathbb{P}(E) \geq 1 - 2dK \exp\left\{-\frac{\xi^2 \bar{N}_{t-1}p_{t-1}^2\lambda^2}{8d^3 K^3 \sigma^2 L_w^4 L_x^4}\right\}.$$

Taking the result above back to the definition of $\kappa_{ti}(\omega_{ti}, \boldsymbol{X}_{ti})$, we have

$$\begin{aligned}
\kappa_{ti}(\omega_{ti}, \boldsymbol{X}_{ti}) = \mathbb{P}(A_{ti} \neq \widehat{\pi}_{t-1}(\boldsymbol{X}_{ti})) &\leq \sum_{\substack{1 \leq a,a' \leq K \\ a \neq a'}} \left[\mathbb{P}(A_{ti} \neq \widehat{\pi}_{t-1}(\boldsymbol{X}_{ti}), A_{ti} = a, \widehat{\pi}_{t-1}(\boldsymbol{X}_{ti}) = a' \mid E) + \mathbb{P}(E^c)\right] \\
&\leq \sum_{\substack{1 \leq a,a' \leq K \\ a \neq a'}} \mathbb{P}(A_{ti} \neq \widehat{\pi}_{t-1}(\boldsymbol{X}_{ti}), A_{ti} = a, \widehat{\pi}_{t-1}(\boldsymbol{X}_{ti}) = a' \mid E) + 2dK^2(K-1) \exp\left\{-\frac{\xi^2 \bar{N}_{t-1}p_{t-1}^2\lambda^2}{8d^3 K^3 \sigma^2 L_w^4 L_x^4}\right\}.
\end{aligned} \quad (36)$$

Next, we focus on bounding the first term $\mathbb{P}(A_{ti} \neq \widehat{\pi}_{t-1}(\boldsymbol{X}_{ti}), A_{ti} = a, \widehat{\pi}_{t-1}(\boldsymbol{X}_{ti}) = a' \mid E)$ for any arm pair $(a, a')$, which is equivalent to

$$\mathbb{E}\left(\mathbb{E}\left[\mathbf{1}\{A_{ti} \neq \widehat{\pi}_{t-1}(\boldsymbol{X}_{ti}), A_{ti} = a, \widehat{\pi}_{t-1}(\boldsymbol{X}_{ti}) = a'\} \mid \widehat{\zeta}_{ti}\right] \mid E\right).$$

**Step 2:** Bound the probability of $\mathbb{E}\left[\mathbf{1}\{A_{ti} \neq \widehat{\pi}_{t-1}(\boldsymbol{X}_{ti}), A_{ti} = a, \widehat{\pi}_{t-1}(\boldsymbol{X}_{ti}) = a'\} \mid \widehat{\zeta}_{ti}\right]$ on event $E$.

Recall that in TS, we have $A_{ti} = \arg\max_a\{\omega_{ti}\boldsymbol{X}_{ti}^\top\widetilde{\boldsymbol{\beta}}_{t-1,a}\}$, where $\widetilde{\boldsymbol{\beta}}_{t-1} \sim \mathcal{N}(\widehat{\boldsymbol{\beta}}_{t-1}, v^2 A_{t-1}^{-1})$ with $\widehat{\boldsymbol{\beta}}_t = \left(\widetilde{\boldsymbol{X}}_{1:t}^\top\widetilde{\boldsymbol{X}}_{1:t}\right)^{-1}\widetilde{\boldsymbol{X}}_{1:t}^\top\boldsymbol{R}_{1:t}$ and $A_t = \widetilde{\boldsymbol{X}}_{1:t}^\top\widetilde{\boldsymbol{X}}_{1:t}$. After simple transformations, for any arm pair $(a, a')$, $\omega_{ti}\boldsymbol{X}_{ti}^\top\widetilde{\boldsymbol{\beta}}_{t-1,a} - \omega_{ti}\boldsymbol{X}_{ti}^\top\widetilde{\boldsymbol{\beta}}_{t-1,a'}$ also follows a normal distribution with

$$\omega_{ti}\boldsymbol{X}_{ti}^\top\widetilde{\boldsymbol{\beta}}_{t-1,a} - \omega_{ti}\boldsymbol{X}_{ti}^\top\widetilde{\boldsymbol{\beta}}_{t-1,a'} \sim \mathcal{N}\left(\omega_{ti}\boldsymbol{X}_{ti}^\top(\widehat{\boldsymbol{\beta}}_{t-1,a} - \widehat{\boldsymbol{\beta}}_{t-1,a'}), v^2\omega_{ti}^2\boldsymbol{X}_{ti}^\top\mathcal{D}_{t-1}\boldsymbol{X}_{ti}\right), \quad (37)$$

where $\mathcal{D}_{t-1} = \text{Var}(\widehat{\boldsymbol{\beta}}_{t-1,a} - \widehat{\boldsymbol{\beta}}_{t-1,a'}) = (A_{t-1}^{-1})_{aa} + (A_{t-1}^{-1})_{a'a'} - 2(A_{t-1}^{-1})_{aa'}$, with $(A_{t-1}^{-1})_{aa}$ denoting the $d \times d$ dimensional block-diagonal matrix of $A_{t-1}^{-1}$ that corresponds to the variance of $\boldsymbol{\beta}_{t-1,a}$, and $(A_{t-1}^{-1})_{aa'}$ denoting the sub-matrix in $A_{t-1}^{-1}$ representing the covariance between $\widehat{\boldsymbol{\beta}}_{t-1,a}$ and $\widehat{\boldsymbol{\beta}}_{t-1,a'}$. Since $\mathcal{D}_{t-1}$ is symmetric and positive semi-definite, $\mathcal{D}_{t-1} \preceq 2(A_{t-1}^{-1})_{aa} + 2(A_{t-1}^{-1})_{a'a'}$, with $X \preceq Y$ denoting that $(Y - X)$ is positive semi-definite. For the simplicity of implementation, we exclude the interaction term $(A_t^{-1})_{aa'}$ in Algorithm 1, i.e., assuming independence between $\widehat{\boldsymbol{\beta}}_{a,t}$ and $\widehat{\boldsymbol{\beta}}_{a',t}$ while making decisions.

According to the distribution derived in Equation (37), we have

$$\mathbb{E}\Big[\mathbf{1}\big\{A_{ti} \neq \widehat{\pi}_{t-1}(\boldsymbol{X}_{ti}), A_{ti} = a, \widehat{\pi}_{t-1}(\boldsymbol{X}_{ti}) = a'\big\} \mid \widehat{\zeta}_{ti}\Big]$$

$$\leq \mathbb{P}\big\{\omega_{ti}\boldsymbol{X}_{ti}^{\top}(\widetilde{\boldsymbol{\beta}}_{t-1,a} - \widetilde{\boldsymbol{\beta}}_{t-1,a'}) > 0, \omega_{ti}\boldsymbol{X}_{ti}^{\top}(\widehat{\boldsymbol{\beta}}_{t-1,a} - \widehat{\boldsymbol{\beta}}_{t-1,a'}) < 0 \mid \omega_{ti}\boldsymbol{X}_{ti}^{\top}(\widehat{\boldsymbol{\beta}}_{t-1,a} - \widehat{\boldsymbol{\beta}}_{t-1,a'})\big\}$$

$$= 1 - \Phi\Big[\omega_{ti}\boldsymbol{X}_{ti}^{\top}(\widehat{\boldsymbol{\beta}}_{t-1,a'} - \widehat{\boldsymbol{\beta}}_{t-1,a})/\sqrt{v^2\omega_{ti}^2\boldsymbol{X}_{ti}^{\top}\mathcal{D}_{t-1}\boldsymbol{X}_{ti}}\Big],$$

where $\Phi(\cdot)$ is the cumulative distribution function of $\mathcal{N}(0,1)$.

Denote $\widehat{z}_{ti} = \omega_{ti}\boldsymbol{X}_{ti}^{\top}(\widehat{\boldsymbol{\beta}}_{t-1,a'} - \widehat{\boldsymbol{\beta}}_{t-1,a})/\sqrt{v^2\omega_{ti}^2\boldsymbol{X}_{ti}^{\top}\mathcal{D}_{t-1}\boldsymbol{X}_{ti}}$. According to the tail bound established for standard normal distribution in Section 7.1 of Feller (1991), we have

$$\mathbb{E}\Big[\mathbf{1}\big\{A_{ti} \neq \widehat{\pi}_{t-1}(\boldsymbol{X}_{ti}), A_{ti} = a, \widehat{\pi}_{t-1}(\boldsymbol{X}_{ti}) = a'\big\} \mid \widehat{\zeta}_{ti}\Big]$$

$$\leq \mathbb{E}\big\{\exp(-\widehat{z}_{ti}^2/2)\big\} = \mathbb{E}\bigg\{\exp\bigg(-\frac{\omega_{ti}^2\big\{\boldsymbol{X}_{ti}^{\top}(\widehat{\boldsymbol{\beta}}_{t-1,a'} - \widehat{\boldsymbol{\beta}}_{t-1,a})\big\}^2}{2v^2\omega_{ti}^2\boldsymbol{X}_{ti}^{\top}\mathcal{D}_{t-1}\boldsymbol{X}_{ti}}\bigg)\bigg\} \tag{38}$$

Define $\widehat{\sigma}_{t-1,a}(\boldsymbol{X}_{ti}) = |\omega_{ti}|\sqrt{\boldsymbol{X}_{ti}^{\top}\Big(\widetilde{\boldsymbol{X}}_{1:(t-1)}^{\top}\widetilde{\boldsymbol{X}}_{1:(t-1)}\Big)_{aa}^{-1}\boldsymbol{X}_{ti}}$. According to the upper bound derived in Equation (30), we have

$$\omega_{ti}^2\boldsymbol{X}_{ti}^{\top}\mathcal{D}_{t-1}\boldsymbol{X}_{ti} \leq 2\widehat{\sigma}_{t-1,a}(\boldsymbol{X}_{ti})^2 + 2\widehat{\sigma}_{t-1,a'}(\boldsymbol{X}_{ti})^2 \leq \frac{4L_w^2L_x^2}{\bar{N}_{t-1}p_{t-1}\lambda}.$$

Combining the result above to Equation (38), we can further derive

$$\mathbb{E}\Big[\mathbf{1}\big\{A_{ti} \neq \widehat{\pi}_{t-1}(\boldsymbol{X}_{ti}), A_{ti} = a, \widehat{\pi}_{t-1}(\boldsymbol{X}_{ti}) = a'\big\} \mid \widehat{\zeta}_{ti}\Big]$$

$$\leq \mathbb{E}\bigg\{\exp\bigg(-\frac{\omega_{ti}^2\big\{\boldsymbol{X}_{ti}^{\top}(\widehat{\boldsymbol{\beta}}_{t-1,a} - \widehat{\boldsymbol{\beta}}_{t-1,a'})\big\}^2}{2v^2\omega_{ti}^2\boldsymbol{X}_{ti}^{\top}\mathcal{D}_{t-1}\boldsymbol{X}_{ti}}\bigg)\bigg\} \leq \mathbb{E}\bigg\{\exp\bigg(-\frac{\bar{N}_{t-1}p_{t-1}\lambda\omega_{ti}^2\big\{\boldsymbol{X}_{ti}^{\top}(\widehat{\boldsymbol{\beta}}_{t-1,a} - \widehat{\boldsymbol{\beta}}_{t-1,a'})\big\}^2}{8v^2L_w^2L_x^2}\bigg)\bigg\}.$$

Note that on event $E$, for any $0 < \xi < |\zeta_{ti}|/2$, $\widehat{\zeta}_{ti}^2 = \omega_{ti}^2\big\{\boldsymbol{X}_{ti}^{\top}(\widehat{\boldsymbol{\beta}}_{t-1,a} - \widehat{\boldsymbol{\beta}}_{t-1,a'})\big\}^2 \geq (|\zeta_{ti}| - \xi)^2$. Therefore,

$$\mathbb{E}\Big(\mathbb{E}\Big[\mathbf{1}\big\{A_{ti} \neq \widehat{\pi}_{t-1}(\boldsymbol{X}_{ti}), A_{ti} = a, \widehat{\pi}_{t-1}(\boldsymbol{X}_{ti}) = a'\big\} \mid \widehat{\zeta}_{ti}\Big] \mid E\Big)$$

$$\leq \mathbb{E}\bigg\{\exp\bigg(-\frac{\bar{N}_{t-1}p_{t-1}\lambda(|\zeta_{ti}| - \xi)^2}{8v^2L_w^2L_x^2}\bigg)\bigg\} \leq \exp\bigg(-\frac{\bar{N}_{t-1}p_{t-1}\lambda(|\zeta_{ti}| - \xi)^2}{8v^2L_w^2L_x^2}\bigg). \tag{39}$$

**Step 3:** Summary.

Combining the results of Equation (36) and (39), we finally have

$$\kappa_{ti}(\omega_{ti}, \boldsymbol{X}_{ti}) \leq K(K-1)\exp\bigg(-\frac{\bar{N}_{t-1}p_{t-1}\lambda(|\zeta_{ti}| - \xi)^2}{8v^2L_w^2L_x^2}\bigg) + 2dK^2(K-1)\exp\bigg\{-\frac{\xi^2\bar{N}_{t-1}p_{t-1}^2\lambda^2}{8d^3K^3\sigma^2L_w^4L_x^4}\bigg\}.$$

The proof is thus complete.

# H. Proof of Theorem 4.3: the Asymptotic Normality of $\widehat{\boldsymbol{\beta}}_t$

Recall that $\mathbb{E}[R_{ti}] = \widetilde{\boldsymbol{X}}_{ti}^{\top}\boldsymbol{\beta}$, where $\widetilde{\boldsymbol{X}}_{ti} = (\mathbf{1}_{N_t}^{\top}\boldsymbol{W}_{ti}\boldsymbol{I}_{\boldsymbol{A}_t=1}\boldsymbol{X}_t, \ldots, \mathbf{1}_{N_t}^{\top}\boldsymbol{W}_{ti}\boldsymbol{I}_{\boldsymbol{A}_t=K}\boldsymbol{X}_t)^{\top} \in \mathbb{R}^{dK}$. Define the number of samples collected till the end of round t as $\bar{N}_t = \sum_{s=1}^{t} N_t$ and further define $N := \bar{N}_T$. Therefore, we estimate $\widehat{\boldsymbol{\beta}}_t$ at round $t$ by

$$\widehat{\boldsymbol{\beta}}_t = \Big(\widetilde{\boldsymbol{X}}_{1:t}^{\top}\widetilde{\boldsymbol{X}}_{1:t}\Big)^{-1}\widetilde{\boldsymbol{X}}_{1:t}^{\top}\boldsymbol{R}_{1:t} = \bigg\{\frac{1}{\bar{N}_t}\sum_{s=1}^{t}\sum_{i=1}^{N_s}\widetilde{\boldsymbol{X}}_{si}\widetilde{\boldsymbol{X}}_{si}^{\top}\bigg\}^{-1}\bigg\{\frac{1}{\bar{N}_t}\sum_{s=1}^{t}\sum_{i=1}^{N_s}\widetilde{\boldsymbol{X}}_{si}R_{si}\bigg\}. \tag{40}$$

Since $R_{si} = \widetilde{\boldsymbol{X}}_{ti}^{\top}\boldsymbol{\beta} + \eta_{si}$, we can write

$$\sqrt{\bar{N}_t}(\widehat{\boldsymbol{\beta}}_t - \boldsymbol{\beta}) = \underbrace{\left\{\frac{1}{\bar{N}_t}\sum_{s=1}^{t}\sum_{i=1}^{N_s}\widetilde{\boldsymbol{X}}_{si}\widetilde{\boldsymbol{X}}_{si}^{\top}\right\}^{-1}}_{\eta_2}\underbrace{\left\{\frac{1}{\sqrt{\bar{N}_t}}\sum_{s=1}^{t}\sum_{i=1}^{N_s}\widetilde{\boldsymbol{X}}_{si}\eta_{si}\right\}}_{\eta_1} \tag{41}$$

Here, with a slight abuse of notation, we denote the two terms in the above equation as $\eta_1$ and $\eta_2$, which are to be estimated separately in later steps. Notice that this differs from the sub-Gaussian error of the reward-generating function in Equation (1).

**Step 1:** Show that $\eta_1 = \frac{1}{\sqrt{\bar{N}_t}}\sum_{s=1}^{t}\sum_{i=1}^{N_s}\widetilde{\boldsymbol{X}}_{si}\eta_{si} \xrightarrow{\mathcal{D}} \mathcal{N}(\boldsymbol{0}_{dK}, G)$.

According to Cramer-Wold device, it suffices to show that for any $\boldsymbol{v} \in \mathbb{R}^{dK}$,

$$\eta_1(\boldsymbol{v}) = \frac{1}{\sqrt{\bar{N}_t}}\sum_{s=1}^{t}\sum_{i=1}^{N_s}\boldsymbol{v}^{\top}\widetilde{\boldsymbol{X}}_{si}\eta_{si} \xrightarrow{\mathcal{D}} \mathcal{N}(\boldsymbol{0}_{dK}, \boldsymbol{v}^{\top}G\boldsymbol{v}). \tag{42}$$

Before proceeding, let's flatten the round-unit pairs $\{(t,i)\}_{1\leq t\leq T, 1\leq i\leq N_t}$ to an unit queue $Q(t,i) = \sum_{s=1}^{t-1}N_s + i$, such that all of the units are measured in a chronological order. Notice that the order of units in the same round does not matter, since the action decisions for all units in round $t$ are made at the end of that round. For any "flattened" unit index $q_0 = Q(i_0, t_0) \subset \{1, \ldots, N\}$, we define $\mathcal{H}_{q_0}$ as the $\sigma$-algebra containing the information up to unit $q_0$. That is,

$$\mathcal{H}_{q_0} = \sigma(\boldsymbol{v}^{\top}\widetilde{\boldsymbol{X}}_1\eta_1, \ldots, \boldsymbol{v}^{\top}\widetilde{\boldsymbol{X}}_{q_0}\eta_{q_0}).$$

For different indices $q$, there is a jump in information gathering for $\mathcal{H}_q$ whenever $q = Q(i=1, t)$ for some $t$. Since $\widetilde{\boldsymbol{X}}_q \in \mathcal{H}_q$, all of the action assignment information collected at round $t$, i.e., $\boldsymbol{A}_t$, is contained in $\mathcal{H}_q$ at the beginning of this round. With a slight abuse of notation, in the following proof, we will also use $\mathcal{H}_t$ to denote all historical data collected up to round $t$.

The tricky part of establishing asymptotic properties for $\widehat{\boldsymbol{\beta}}_t$ lies in the data dependence. Specifically, the transformed covariate vector $\widetilde{\boldsymbol{X}}_{si}$ is a function of $(\boldsymbol{W}_t, \boldsymbol{A}_s, \boldsymbol{X}_s)$, thus depending on all of the actions and original covariates information collected at round $t$. As such, $\widetilde{\boldsymbol{X}}_{si}\eta_{si} \not\perp \widetilde{\boldsymbol{X}}_{i's'}\eta_{i's'}$ for any $(s, s')$, since (1) if $s = s'$, units in the same round $s$ are correlated by $\boldsymbol{W}_s$; (2) if $s < s'$, unit are dependent since the later decisions made on $\boldsymbol{A}_{s'}$ will depend on $(\boldsymbol{W}_s, \boldsymbol{X}_s, \boldsymbol{A}_s)$.

Now let's use Martingale CLT to establish the asymptotic properties. We will prove shortly that $\{\boldsymbol{v}^{\top}\widetilde{\boldsymbol{X}}_{si}\eta_{si}\}$, or equivalently $\{\boldsymbol{v}^{\top}\widetilde{\boldsymbol{X}}_q\eta_q\}$ after flattening, is a Martingale difference sequence. That is, we would like to show

$$\mathbb{E}[\boldsymbol{v}^{\top}\widetilde{\boldsymbol{X}}_q\eta_q|\mathcal{H}_{q-1}] = 0, \quad \forall q \in \{1, \ldots, N\}. \tag{43}$$

Suppose $q = Q(t, i)$ for some $(t, i)$ pair. According to our assumption on the noise term in the main paper, $\eta_{ti} \perp (\boldsymbol{X}_t, \boldsymbol{W}_t)|\boldsymbol{A}_t \Rightarrow \eta_{ti} \perp \widetilde{\boldsymbol{X}}_t|\boldsymbol{A}_t$ as $\widetilde{\boldsymbol{X}}_t$ is a function of $(\boldsymbol{W}_t, \boldsymbol{X}_t, \boldsymbol{A}_t)$.

$$\mathbb{E}[\boldsymbol{v}^{\top}\widetilde{\boldsymbol{X}}_q\eta_q|\mathcal{H}_{q-1}] = \mathbb{E}\left[\mathbb{E}[\boldsymbol{v}^{\top}\widetilde{\boldsymbol{X}}_q\eta_q|\mathcal{H}_{q-1}, \boldsymbol{A}_t]|\mathcal{H}_{q-1}\right] \overset{(A1)}{=} \mathbb{E}\left[\mathbb{E}[\boldsymbol{v}^{\top}\widetilde{\boldsymbol{X}}_q|\mathcal{H}_{q-1}, \boldsymbol{A}_t] \cdot \underbrace{\mathbb{E}[\eta_q|\mathcal{H}_{q-1}, \boldsymbol{A}_t]}_{=0}|\mathcal{H}_{q-1}\right],$$

Now it suffice to (1) check the Lindeberg condition, and (2) derive the limit of conditional variance.

**(1) We first check the Lindeberg condition.**
For any $\delta > 0$, we define

$$\psi = \sum_{q=1}^{\bar{N}_q}\mathbb{E}\left[\frac{1}{\bar{N}_q}(\boldsymbol{v}^{\top}\widetilde{\boldsymbol{X}}_q)^2\eta_q^2 \cdot \boldsymbol{1}\left\{\left|\frac{1}{\sqrt{\bar{N}_q}}\boldsymbol{v}^{\top}\widetilde{\boldsymbol{X}}_q\eta_q\right| > \delta\right\} \mid \mathcal{H}_{q-1}\right] \tag{44}$$

According to Assumption A.1.b-c,

$$(\boldsymbol{v}^\top \widetilde{\boldsymbol{X}}_q)^2 \leq \|\boldsymbol{v}\|_2^2 \cdot \|\widetilde{\boldsymbol{X}}_q\|_2^2 \leq K\|\boldsymbol{v}\|_2^2 \cdot \Big\| \sum_{j=1}^{N_t} W_{t,ij} \boldsymbol{X}_{tj} \Big\|_2^2 \leq K\|\boldsymbol{v}\|_2^2 \cdot L_x^2 d \cdot \Big( \sum_j |W_{t,ij}| \Big)^2 \leq dK L_w^2 L_x^2 \|\boldsymbol{v}\|_2^2. \quad (45)$$

Then

$$\mathbf{1}\Big\{ \Big| \frac{1}{\sqrt{\bar{N}_q}} \boldsymbol{v}^\top \widetilde{\boldsymbol{X}}_q \eta_q \Big| > \delta \Big\} \leq \mathbf{1}\Big\{ dK L_w^2 L_x^2 \|\boldsymbol{v}\|_2^2 \eta_q^2 > \bar{N}_q \delta^2 \Big\} = \mathbf{1}\Big\{ \eta_q^2 > \frac{\bar{N}_q \delta^2}{dK L_w^2 L_x^2 \|\boldsymbol{v}\|_2^2} \Big\}.$$

Thus,

$$\psi \leq \frac{dK L_w^2 L_x^2 \|\boldsymbol{v}\|_2^2}{\bar{N}_q} \sum_{q=1}^{\bar{N}_q} \mathbb{E}\Big[ \eta_q^2 \cdot \mathbf{1}\Big\{ \eta_q^2 > \frac{\bar{N}_q \delta^2}{dK L_w^2 L_x^2 \|\boldsymbol{v}\|_2^2} \Big\} \mid \mathcal{H}_{q-1} \Big]. \quad (46)$$

Define $f_{\bar{N}_q} = \frac{dK L_w^2 L_x^2 \|\boldsymbol{v}\|_2^2}{\bar{N}_q} \sum_{q=1}^{\bar{N}_q} \eta_q^2 \cdot \mathbf{1}\Big\{ \eta_q^2 > \frac{\bar{N}_q \delta^2}{dK L_w^2 L_x^2 \|\boldsymbol{v}\|_2^2} \Big\}$, and $g_{\bar{N}_q} = \frac{dK L_w^2 L_x^2 \|\boldsymbol{v}\|_2^2}{\bar{N}_q} \sum_{q=1}^{\bar{N}_q} \eta_q^2$. It is obvious that $|f_{\bar{N}_q}| \leq g_{\bar{N}_q}$ a.s. and for all $q$. Since

$$\mathbb{E}[\eta_q^2 | \mathcal{H}_{q-1}] = \mathbb{E}\big[ \mathbb{E}[\eta_q^2 | \boldsymbol{A}_t, \mathcal{H}_{q-1}] | \mathcal{H}_{q-1} \big] = \sigma^2 < \infty,$$

we have

$$\mathbb{E}[g_{\bar{N}_q} | \mathcal{H}_{q-1}] \frac{dK L_w^2 L_x^2 \|\boldsymbol{v}\|_2^2}{\bar{N}_q} \sum_{q=1}^{\bar{N}_q} \sigma^2 \leq dK L_w^2 L_x^2 \|\boldsymbol{v}\|_2^2 \sigma^2 < \infty,$$

thus $g_{\bar{N}_q}$ is integrable for all $q$.

For each realization of random variable sequence $\{\eta_q\}_{q=1}^\infty$, $\lim_{\bar{N}_q \to \infty} f_{\bar{N}_q} = 0$ as $\mathbf{1}\Big\{ \eta_q^2 > \frac{\bar{N}_q \delta^2}{dK L_w^2 L_x^2 \|\boldsymbol{v}\|_2^2} \Big\} = 0$ when $\bar{N}_q$ is large enough.

Therefore, by Generalized Dominated Convergence Theorem (GDCT), it follows from Equation (46) that $\psi \leq \mathbb{E}[f_{\bar{N}_q} | \mathcal{H}_{q-1}] \to 0$ as $q \to \infty$. The Lindeberg condition is thus verified.

**(2) We next derive the limit of conditional variance.**

$$\frac{1}{\bar{N}_q} \sum_{s=1}^t \sum_{i=1}^{N_s} \mathbb{E}\big[ (\boldsymbol{v}^\top \widetilde{\boldsymbol{X}}_q)^2 \eta_q^2 | \mathcal{H}_{q-1} \big] = \frac{1}{\bar{N}_q} \sum_{s=1}^t \sum_{i=1}^{N_s} \mathbb{E}\big[ (\boldsymbol{v}^\top \widetilde{\boldsymbol{X}}_q)^2 \mathbb{E}[\eta_q^2 | \boldsymbol{A}_i] | \mathcal{H}_{q-1} \big]$$
$$= \frac{1}{\bar{N}_q} \sum_{s=1}^t \sum_{i=1}^{N_s} \sigma^2 \mathbb{E}\big[ (\boldsymbol{v}^\top \widetilde{\boldsymbol{X}}_q)^2 | \mathcal{H}_{q-1} \big] \quad (47)$$

where the last equality holds since $\eta_q^2 | \boldsymbol{A}_i$ i.i.d. follows $\mathcal{N}(0, \sigma^2)$.

Recall that for any unit index $q = Q(t, i)$,

$$\widetilde{\boldsymbol{X}}_q = \widetilde{\boldsymbol{X}}_{ti} = (\mathbf{1}_{N_t}^\top \boldsymbol{W}_{ti} \mathrm{diag}(\mathbf{1}\{A_{ti} = 0\}_{1 \leq i \leq N_t}) \boldsymbol{X}_t, \mathbf{1}_{N_t}^\top \boldsymbol{W}_{ti} \mathrm{diag}(\mathbf{1}\{A_{ti} = 1\}_{1 \leq i \leq N_t}) \boldsymbol{X}_t)^\top.$$

After some manipulations, we have

$$\widetilde{\boldsymbol{X}}_q \widetilde{\boldsymbol{X}}_q^\top = \widetilde{\boldsymbol{X}}_{ti} \widetilde{\boldsymbol{X}}_{ti}^\top := \begin{bmatrix} M_{1,1} & M_{1,2} & \cdots & M_{1,K} \\ M_{2,1} & M_{2,2} & \cdots & M_{2,K} \\ \vdots & \vdots & \ddots & \vdots \\ M_{K,1} & M_{K,2} & \cdots & M_{K,K} \end{bmatrix} \in \mathbb{R}^{dK \times dK}, \quad (48)$$

where $M_{a,a'} := \sum_{k=1}^{N_t} \sum_{l=1}^{N_t} W_{t,ik} W_{t,il} \boldsymbol{X}_{tk} \boldsymbol{X}_{tl}^\top \cdot \mathbf{1}\{A_{tk} = a\} \mathbf{1}\{A_{tl} = a'\}$ is a $K$ by $K$ sub-matrix.

Since we assume $\boldsymbol{X}_{ti} \sim \mathcal{X}$ and $\boldsymbol{W}_t \sim \mathcal{W}$ are known to us, the main challenge of deriving the conditional variance lies in estimating the conditional expectation of $\mathbb{E}[\mathbf{1}\{A_{tk} = a\} \mathbf{1}\{A_{tl} = a'\} | \boldsymbol{W}_t, \boldsymbol{X}_t, \mathcal{H}_{q-1}]$ for any $q \in \{1, \ldots, \bar{N}_q\}$.

Recall that $\mathcal{H}_{q-1}$ is defined as the $\sigma$-algebra containing the information up to unit $q-1$. That is,

$$\mathcal{H}_{q-1} = \sigma(\boldsymbol{v}^\top \widetilde{\boldsymbol{X}}_1 \eta_1, \ldots, \boldsymbol{v}^\top \widetilde{\boldsymbol{X}}_{q-1} \eta_{q-1}).$$

For different indices $q$, there is a jump in information gathering for $\mathcal{H}_q$ whenever $q = Q(i = 1, t)$ for some $t$. Since $\widetilde{\boldsymbol{X}}_q \in \mathcal{H}_q$, all of the action assignment information collected at round $t$, i.e., $\boldsymbol{A}_t$, is contained in $\mathcal{H}_q$ at the beginning of this round. Thanks to the property of $\eta_q$ that $\mathbb{E}[\eta_q | \boldsymbol{A}_t] = 0$, the conditional variance $\mathbb{E}\big[(\boldsymbol{v}^\top \widetilde{\boldsymbol{X}}_q)^2 \epsilon^2 | \mathcal{H}_{q-1}\big] = \mathbb{E}\big[(\boldsymbol{v}^\top \widetilde{\boldsymbol{X}}_q)^2 \epsilon^2\big] = \mathbb{E}\big[(\boldsymbol{v}^\top \widetilde{\boldsymbol{X}}_q)^2 \epsilon^2 | \mathcal{H}_{t-1}\big]$ for any $q = Q(t, i)$. Still, we take the $d \times d$ sub-matrix $M_{a,a'}$ as an example to calculate the asymptotic variance.

$$
\begin{aligned}
\mathbb{E}[M_{a,a'}] &= \mathbb{E}\bigg[ \sum_{k=1}^{N_t} \sum_{l=1}^{N_t} W_{t,ik} W_{t,il} \boldsymbol{X}_{tk} \boldsymbol{X}_{tl}^\top \cdot \mathbf{1}\{A_{tk} = a\} \mathbf{1}\{A_{tl} = a'\} \bigg] \\
&= \mathbb{E}\bigg[ \mathbb{E}\bigg[ \sum_{k=1}^{N_t} \sum_{l=1}^{N_t} W_{t,ik} W_{t,il} \boldsymbol{X}_{tk} \boldsymbol{X}_{tl}^\top \cdot \mathbf{1}\{A_{tk} = a\} \mathbf{1}\{A_{tl} = a'\} \big| \boldsymbol{W}_t, \boldsymbol{X}_t, \mathcal{H}_{t-1} \bigg] \bigg] \\
&= \mathbb{E}\bigg[ \sum_{k=1}^{N_t} \sum_{l=1}^{N_t} W_{t,ik} W_{t,il} \boldsymbol{X}_{tk} \boldsymbol{X}_{tl}^\top \cdot \mathbb{E}\big[ \mathbf{1}\{A_{tk} = a\} \mathbf{1}\{A_{tl} = a'\} \big| \boldsymbol{W}_t, \boldsymbol{X}_t, \mathcal{H}_{t-1} \big] \bigg].
\end{aligned}
\tag{49}
$$

Notice that $A_{tk} \perp A_{tl} | \boldsymbol{W}_t, \boldsymbol{X}_t, \mathcal{H}_{t-1}$ for any $(k, l) \in \{1, \ldots, N_t\}$ and $k \neq l$. This independence arises because the actions assigned to units $Q(k, t)$ and $Q(l, t)$ are determined by two factors: exploitation and exploration.

1. **Exploitation**: The action $A_{tk}$ is partially determined by $\hat{\pi}_{t-1}(X_{tk})$, where $\hat{\pi}_{t-1}$ is a function of $\mathcal{H}_{t-1}$ and is obtained by fitting a model to data from the first $t-1$ rounds. Therefore, given $(\boldsymbol{W}_t, \boldsymbol{X}_t, \mathcal{H}_{t-1})$, $\hat{\pi}_{t-1}(\boldsymbol{X}_{tk})$ and $\hat{\pi}_{t-1}(\boldsymbol{X}_{tl}) | \mathcal{H}_{t-1}$ are both constants and thus independent from each other.

2. **Exploration**: The action $A_{tk}$ is also influenced by a specific exploration method based on the "optimal" action identified during exploitation. In $\epsilon$-greedy, the level of exploration is determined by $\epsilon_{ti}$, which is independently assigned to each unit. For UCB and TS, the exploration level for each unit is a function of $\mathcal{H}_{t-1}$, making them mutually independent given $\mathcal{H}_{t-1}$.

Therefore,

$$\mathbb{E}\big[\mathbf{1}\{A_{tk} = a\} \mathbf{1}\{A_{tl} = a'\} \big| \boldsymbol{W}_t, \boldsymbol{X}_t, \mathcal{H}_{t-1}\big] = \mathbb{E}\big[\mathbf{1}\{A_{tk} = a\} \big| \boldsymbol{W}_t, \boldsymbol{X}_t, \mathcal{H}_{t-1}\big] \cdot \mathbb{E}\big[\mathbf{1}\{A_{tl} = a'\} \big| \boldsymbol{W}_t, \boldsymbol{X}_t, \mathcal{H}_{t-1}\big]. \tag{50}$$

For the simplicity of notation, we define $\nu_{ti}(\omega_{ti}, \boldsymbol{X}_{ti}, \mathcal{H}_{t-1}) = \mathbb{P}(A_{ti} \neq \pi^*(\boldsymbol{X}_{ti}) | \boldsymbol{W}_t, \boldsymbol{X}_{ti}, \mathcal{H}_{t-1}) = \mathbb{P}(A_{ti} \neq \pi^*(\boldsymbol{X}_{ti}) | \omega_{ti}, \boldsymbol{X}_{ti}, \mathcal{H}_{t-1})$. Since

$$\mathbf{1}\{A_{ti} = a\} = \mathbf{1}\{A_{ti} = \pi^*(\boldsymbol{X}_{ti})\} \cdot \mathbf{1}\{\pi^*(\boldsymbol{X}_{ti}) = a\} + \mathbf{1}\{A_{ti} \neq \pi^*(\boldsymbol{X}_{ti})\} \cdot \mathbf{1}\{\pi^*(\boldsymbol{X}_{ti}) \neq a\},$$

we have

$$
\begin{aligned}
& \mathbb{E}\big[\mathbf{1}\{A_{ti} = a\} \big| \boldsymbol{W}_t, \boldsymbol{X}_t, \mathcal{H}_{t-1}\big] \\
&= \mathbb{E}\big[\mathbf{1}\{A_{ti} = \pi^*(\boldsymbol{X}_{ti})\} \cdot \mathbf{1}\{\pi^*(\boldsymbol{X}_{ti}) = a\} + \mathbf{1}\{A_{ti} \neq \pi^*(\boldsymbol{X}_{ti})\} \cdot \mathbf{1}\{\pi^*(\boldsymbol{X}_{ti}) \neq a\} \big| \boldsymbol{W}_t, \boldsymbol{X}_t, \mathcal{H}_{t-1}\big] \\
&= \mathbb{P}(A_{ti} = \pi^*(\boldsymbol{X}_{ti}) | \boldsymbol{W}_t, \boldsymbol{X}_t, \mathcal{H}_{t-1}) \mathbf{1}\{\pi^*(\boldsymbol{X}_{ti}) = a\} + \mathbb{P}(A_{ti} \neq \pi^*(\boldsymbol{X}_{ti}) | \boldsymbol{W}_t, \boldsymbol{X}_t, \mathcal{H}_{t-1}) \mathbf{1}\{\pi^*(\boldsymbol{X}_{ti}) \neq a\} \\
&= (1 - \nu_{ti}(\omega_{ti}, \boldsymbol{X}_{ti}, \mathcal{H}_{t-1})) \mathbf{1}\{a = \arg\max_b \omega_{ti} \boldsymbol{X}_{ti} \boldsymbol{\beta}_b\} + \nu_{ti}(\omega_{ti}, \boldsymbol{X}_{ti}, \mathcal{H}_{t-1}) \mathbf{1}\{a \neq \arg\max_b \omega_{ti} \boldsymbol{X}_{ti} \boldsymbol{\beta}_b\}.
\end{aligned}
\tag{51}
$$

Plugging in the result of Equation (50), (51) to Equation (49), one can obtain

$$\mathbb{E}[M_{a,a'}] = \mathbb{E}_{\boldsymbol{W}_t,\boldsymbol{X}_t}\left[\sum_{k=1}^{N_t}\sum_{l=1}^{N_t}W_{t,ik}W_{t,il}\boldsymbol{X}_{tk}\boldsymbol{X}_{tl}^\top \cdot \mathbb{E}\big[\mathbf{1}\{A_{tk}=a\}\mathbf{1}\{A_{tl}=a'\}\big|\boldsymbol{W}_t,\boldsymbol{X}_t,\mathcal{H}_{t-1}\big]\right]$$

$$= \mathbb{E}_{\boldsymbol{W}_t,\boldsymbol{X}_t}\left[\sum_{\substack{1\leq k,l\leq N_t}}^{k\neq l}W_{t,ik}W_{t,il}\boldsymbol{X}_{tk}\boldsymbol{X}_{tl}^\top\right.$$

$$\cdot\left\{(1-\nu_{tk}(\omega_{tk},\boldsymbol{X}_{tk},\mathcal{H}_{t-1}))\mathbf{1}\{a=\arg\max_b\omega_{tk}\boldsymbol{X}_{tk}\boldsymbol{\beta}_b\}+\nu_{tk}(\omega_{tk},\boldsymbol{X}_{tk},\mathcal{H}_{t-1})\mathbf{1}\{a\neq\arg\max_b\omega_{tk}\boldsymbol{X}_{tk}\boldsymbol{\beta}_b\}\right\}$$

$$\cdot\left\{(1-\nu_{tl}(\omega_{tl},\boldsymbol{X}_{tl},\mathcal{H}_{t-1}))\mathbf{1}\{a'=\arg\max_b\omega_{tl}\boldsymbol{X}_{tl}\boldsymbol{\beta}_b\}+\nu_{tl}(\omega_{tl},\boldsymbol{X}_{tl},\mathcal{H}_{t-1})\mathbf{1}\{a'\neq\arg\max_b\omega_{tl}\boldsymbol{X}_{tl}\boldsymbol{\beta}_b\}\right\}$$

$$+\mathbf{1}\{a=a'\}\cdot\sum_{\substack{1\leq k,l\leq N_t}}^{k=l}W_{t,ik}^2\boldsymbol{X}_{tk}\boldsymbol{X}_{tk}^\top$$

$$\left.\cdot\left\{(1-\nu_{tk}(\omega_{tk},\boldsymbol{X}_{tk},\mathcal{H}_{t-1}))\mathbf{1}\{a=\arg\max_b\omega_{tk}\boldsymbol{X}_{tk}\boldsymbol{\beta}_b\}+\nu_{tk}(\omega_{tk},\boldsymbol{X}_{tk},\mathcal{H}_{t-1})\mathbf{1}\{a\neq\arg\max_b\omega_{tk}\boldsymbol{X}_{tk}\boldsymbol{\beta}_b\}\right\}\right].$$

Define $\kappa_\infty(\omega,\boldsymbol{x})=\lim_{q\to\infty}\mathbb{P}(A_{ti}\neq\pi^*(\boldsymbol{x}))$. Following similar procedure as page 19-20 and Lemma B.1 in Ye et al. (2023), we can also derive that $\nu_{tk}(\omega,\boldsymbol{x},\mathcal{H}_{t-1})\xrightarrow{P}\kappa_\infty(\omega,\boldsymbol{x})$, where the limit is free of historical data $\mathcal{H}_{t-1}$. Therefore,

$$\frac{1}{\bar{N}_q}\sum_{q=1}^{\bar{N}_q}\sigma^2\mathbb{E}[M_{a,a'}]\to\frac{1}{\bar{N}_q}\sum_{q=1}^{\bar{N}_q}\sigma^2\mathbb{E}_{\boldsymbol{W}_t,\boldsymbol{X}_t}\left[\sum_{\substack{1\leq k,l\leq N_t}}^{k\neq l}W_{t,ik}W_{t,il}\boldsymbol{X}_{tk}\boldsymbol{X}_{tl}^\top\mathcal{J}_\infty(\omega_{tk},\boldsymbol{X}_{tk},\boldsymbol{\beta},a)\mathcal{J}_\infty(\omega_{tl},\boldsymbol{X}_{tl},\boldsymbol{\beta},a')\right.$$

$$\left.+\mathbf{1}\{a=a'\}\cdot\sum_{\substack{1\leq k,l\leq N_t}}^{k=l}W_{t,ik}^2\boldsymbol{X}_{tk}\boldsymbol{X}_{tk}^\top\mathcal{J}_\infty(\omega_{tk},\boldsymbol{X}_{tk},\boldsymbol{\beta},a)\right], \tag{52}$$

where $\mathcal{J}_\infty(\omega,\boldsymbol{X},\boldsymbol{\beta},a)=\left\{(1-\kappa_\infty(\omega,\boldsymbol{X}))\mathbf{1}\{a=\arg\max_b\omega\boldsymbol{X}^\top\boldsymbol{\beta}_b\}+\kappa_\infty(\omega,\boldsymbol{X})\mathbf{1}\{a\neq\arg\max_b\omega\boldsymbol{X}^\top\boldsymbol{\beta}_b\}\right\}$.

Define $\boldsymbol{v}=(\boldsymbol{v}_1',\boldsymbol{v}_2')'$ where $\boldsymbol{v}_1$ and $\boldsymbol{v}_2$ are both $d$-dimensional vector.

Then $\eta_1(\boldsymbol{v})=\frac{1}{\sqrt{\bar{N}_q}}\sum_{q=1}^{\bar{N}_q}\boldsymbol{v}^\top\widetilde{\boldsymbol{X}}_q\eta_q\xrightarrow{\mathcal{D}}\mathcal{N}(\boldsymbol{0}_{dK},\boldsymbol{v}^\top G\boldsymbol{v})$ with

$$G=\begin{bmatrix}\frac{1}{\bar{N}_q}\sum_{q=1}^{\bar{N}_q}\sigma^2\mathbb{E}[M_{1,1}] & \cdots & \frac{1}{\bar{N}_q}\sum_{q=1}^{\bar{N}_q}\sigma^2\mathbb{E}[M_{1,K}]\\ \vdots & \ddots & \vdots\\ \frac{1}{\bar{N}_q}\sum_{q=1}^{\bar{N}_q}\sigma^2\mathbb{E}[M_{K,1}] & \cdots & \frac{1}{\bar{N}_q}\sum_{q=1}^{\bar{N}_q}\sigma^2\mathbb{E}[M_{K,K}]\end{bmatrix}, \tag{53}$$

where the detailed expression of each submatrix in $G$ is given in Equation (52).

**Step 2:** Show that $\eta_2=\left\{\frac{1}{N_t}\sum_{s=1}^t\sum_{i=1}^{N_s}\widetilde{\boldsymbol{X}}_{si}\widetilde{\boldsymbol{X}}_{si}^\top\right\}^{-1}\xrightarrow{P}\sigma^2 G^{-1}$, where $\sigma^2=\mathbb{E}[\eta_{ti}^2|\boldsymbol{A}_t]$.

Based on Lemma 6 of Chen et al. (2021), it suffice to find the limit of $\frac{1}{\bar{N}_q}\sum_{q=1}^{\bar{N}_q}\boldsymbol{v}^\top\widetilde{\boldsymbol{X}}_q\widetilde{\boldsymbol{X}}_q^\top\boldsymbol{v}$. According to Equation (45), we have

$$\mathbb{P}(|\boldsymbol{v}^\top\widetilde{\boldsymbol{X}}_q\widetilde{\boldsymbol{X}}_q^\top\boldsymbol{v}|>h)\leq\mathbb{P}(dKL_w^2L_x^2\|\boldsymbol{v}\|_2^2>h).$$

Therefore, by Theorem 2.19 in Hall & Heyde (2014), we have

$$\frac{1}{\bar{N}_q}\sum_{q=1}^{\bar{N}_q}\left[\boldsymbol{v}^\top\widetilde{\boldsymbol{X}}_q\widetilde{\boldsymbol{X}}_q^\top\boldsymbol{v}-\mathbb{E}\{(\boldsymbol{v}^\top\widetilde{\boldsymbol{X}}_q)^2|\mathcal{H}_{q-1}\}\right]\xrightarrow{P}0\quad\text{as }q\to\infty.$$

Based on the results in Step 1, one can easily derive $\mathbb{E}\{(\boldsymbol{v}^\top \widetilde{\boldsymbol{X}}_q)^2 | \mathcal{H}_{q-1}\} = \boldsymbol{v}^\top G \boldsymbol{v}/\sigma^2$. Combining the results above and by Continuous Mapping Theorem, we have

$$\eta_2 = \left\{ \frac{1}{\bar{N}_t} \sum_{s=1}^{t} \sum_{i=1}^{N_s} \widetilde{\boldsymbol{X}}_{si} \widetilde{\boldsymbol{X}}_{si}^\top \right\}^{-1} \xrightarrow{p} (G/\sigma^2)^{-1} = \sigma^2 G^{-1}, \tag{54}$$

which finishes the proof of this step.

**Step 3:** Summary.
According to the results of Step 1-2 and Slutsky's Theorem, we can conclude that

$$\sqrt{\bar{N}_t}(\widehat{\boldsymbol{\beta}}_t - \boldsymbol{\beta}) = \eta_2 \eta_1 \xrightarrow{\mathcal{D}} \mathcal{N}(\boldsymbol{0}_{dK}, \sigma^4 G^{-1}), \tag{55}$$

where $G$ is specified in Equation (53).

In the special case when $N_t = 1$ for all $t$, i.e., there is no interference, the asymptotic variance would degenerate to

$$G = \frac{1}{\bar{N}_q} \sum_{q=1}^{\bar{N}_q} \sigma^2 \cdot \begin{bmatrix} \mathcal{K}_\infty(\boldsymbol{\beta}) & 0 \\ 0 & \widetilde{\mathcal{K}}_\infty(\boldsymbol{\beta}) \end{bmatrix}. \tag{56}$$

with

$$\mathcal{K}_\infty(\boldsymbol{\beta}) = \int_{\boldsymbol{x}} \boldsymbol{x}\boldsymbol{x}^\top \cdot \left\{ (1 - \kappa_\infty(\boldsymbol{x}))\mathbf{1}\{\boldsymbol{x}^\top(\boldsymbol{\beta}_0 - \boldsymbol{\beta}_1) \geq 0\} + \kappa_\infty(\boldsymbol{x})\mathbf{1}\{\boldsymbol{x}^\top(\boldsymbol{\beta}_0 - \boldsymbol{\beta}_1) < 0\} \right\} d\mathcal{P}_{\boldsymbol{x}}$$

$$\widetilde{\mathcal{K}}_\infty(\boldsymbol{\beta}) = \int_{\boldsymbol{x}} \boldsymbol{x}\boldsymbol{x}^\top \cdot \left\{ (1 - \kappa_\infty(\boldsymbol{x}))\mathbf{1}\{\boldsymbol{x}^\top(\boldsymbol{\beta}_1 - \boldsymbol{\beta}_0) \geq 0\} + \kappa_\infty(\boldsymbol{x})\mathbf{1}\{\boldsymbol{x}^\top(\boldsymbol{\beta}_1 - \boldsymbol{\beta}_0) < 0\} \right\} d\mathcal{P}_{\boldsymbol{x}}.$$

which align perfectly with Ye et al. (2023) in the cases without interference.

The proof of this theorem is complete.

# I. Proof of Theorem 4.4: the Asymptotic Normality of $V^{\pi^*}$

Recall that the DR optimal value function estimator we derived is given by

$$\widehat{V}_T^{\text{DR}} = \frac{1}{\bar{N}_T} \sum_{t=1}^{T} \sum_{i=1}^{N_t} \left\{ \frac{\mathbf{1}\{a_{ti} = \widehat{\pi}_{t-1}(\boldsymbol{X}_{ti})\}}{1 - \widehat{\kappa}_{t-1}(\boldsymbol{X}_{ti})} \cdot \left( r_{si} - \widehat{\mu}_{t-1}^{(t,i)}(\boldsymbol{X}_t, \widehat{\pi}_{t-1}(\boldsymbol{X}_t)) \right) + \widehat{\mu}_{t-1}^{(t,i)}(\boldsymbol{X}_t, \widehat{\pi}_{t-1}(\boldsymbol{X}_t)) \right\}. \tag{57}$$

For the brevity of notation, we will omit the superscript in $\widehat{V}_t^{\text{DR}}$ in the following proof.

Now we defined two related value functions $\widetilde{V}_T$ and $\bar{V}_T$ as below:

$$\widetilde{V}_T = \frac{1}{\bar{N}_T} \sum_{t=1}^{T} \sum_{i=1}^{N_t} \left\{ \frac{\mathbf{1}\{a_{ti} = \widehat{\pi}_{t-1}(\boldsymbol{X}_{ti})\}}{1 - \kappa_{t-1}(\boldsymbol{X}_{ti})} \cdot \left( r_{ti} - \mu^{(t,i)}(\boldsymbol{X}_t, \widehat{\pi}_{t-1}(\boldsymbol{X}_t)) \right) + \mu^{(t,i)}(\boldsymbol{X}_t, \widehat{\pi}_{t-1}(\boldsymbol{X}_t)) \right\},$$

$$\bar{V}_T = \frac{1}{\bar{N}_T} \sum_{t=1}^{T} \sum_{i=1}^{N_t} \left\{ \frac{\mathbf{1}\{a_{ti} = \pi^*(\boldsymbol{X}_{ti})\}}{\mathbb{P}(a_{ti} = \pi^*(\boldsymbol{X}_{ti}))} \cdot \left( r_{ti} - \mu^{(t,i)}(\boldsymbol{X}_t, \pi^*(\boldsymbol{X}_t)) \right) + \mu^{(t,i)}(\boldsymbol{X}_t, \pi^*(\boldsymbol{X}_t)) \right\}.$$

The proof of this theorem can be decomposed into three steps. In step 1, we aim to prove $\widehat{V}_T = \widetilde{V}_T + o_p(\bar{N}_T^{-1/2})$. In step 2, we show that $\widehat{V}_T = \widetilde{V}_T + o_p(\bar{N}_T^{-1/2})$. In step 3, we show $\sqrt{\bar{N}_T}(\bar{V}_T - V^{\pi^*}) \xrightarrow{\mathcal{D}} \mathcal{N}(0, \sigma_V^2)$, where the variance term is given by

$$\sigma_V^2 = \int \frac{\sigma^2}{1 - \kappa_\infty(\boldsymbol{x})} d\mathcal{P}_{\boldsymbol{x}} + \frac{\sum_{i,t} \omega_{ti}^2}{\bar{N}_T} \cdot \text{Var}\left\{ \pi^*(\boldsymbol{x}) \cdot \boldsymbol{x}^\top \boldsymbol{\beta}_1 + \{1 - \pi^*(\boldsymbol{x})\} \cdot \boldsymbol{x}^\top \boldsymbol{\beta}_0 \right\}.$$

Combining the above three steps, the proof of theorem 4.4 is thus complete.

Now, let's detail the proof of step 1-3.

**Step 1:** Prove that $\widehat{V}_T = \widetilde{V}_T + o_p(\bar{N}_T^{-1/2})$.

Notice that the different between $\widehat{V}_T$ and $\widetilde{V}_T$ lies in the estimation accuracy of (1) the propensity score function $\hat{\kappa}_{t-1}$ and (2) outcome estimation function $\widehat{\mu}_{t-1}^{(t,i)}$. To simplify this problem, we introduce another intermediate value function $\check{V}_T$ as

$$\check{V}_T = \frac{1}{\bar{N}_T} \sum_{t=1}^{T} \sum_{i=1}^{N_t} \left\{ \frac{\mathbf{1}\{a_{ti} = \widehat{\pi}_{t-1}(\boldsymbol{X}_{ti})\}}{1 - \kappa_{t-1}(\boldsymbol{X}_{ti})} \cdot \left( r_{ti} - \widehat{\mu}_{t-1}^{(t,i)}(\boldsymbol{X}_t, \widehat{\pi}_{t-1}(\boldsymbol{X}_{ti})) \right) + \widehat{\mu}_{t-1}^{(t,i)}(\boldsymbol{X}_t, \widehat{\pi}_{t-1}(\boldsymbol{X}_{ti})) \right\}.$$

Now the problem becomes proving (1) $\widehat{V}_T = \check{V}_T + o_p(\bar{N}_T^{-1/2})$, and (2) $\check{V}_T = \widetilde{V}_T + o_p(\bar{N}_T^{-1/2})$.

First, let's prove (1) $\widehat{V}_T = \check{V}_T + o_p(\bar{N}_T^{-1/2})$. Notice that

$$\widehat{V}_T - \check{V}_T = \frac{1}{\bar{N}_T} \sum_{t=1}^{T} \sum_{i=1}^{N_t} \left[ \frac{\mathbf{1}\{a_{ti} = \widehat{\pi}_{t-1}(\boldsymbol{X}_{ti})\}}{1 - \hat{\kappa}_{t-1}(\boldsymbol{X}_{ti})} - \frac{\mathbf{1}\{a_{ti} = \widehat{\pi}_{t-1}(\boldsymbol{X}_{ti})\}}{1 - \kappa_{t-1}(\boldsymbol{X}_{ti})} \right] \cdot \left\{ r_{ti} - \widehat{\mu}_{t-1}^{(t,i)}(\boldsymbol{X}_t, \widehat{\pi}_{t-1}(\boldsymbol{X}_t)) \right\}$$

$$= \frac{1}{\bar{N}_T} \sum_{t=1}^{T} \sum_{i=1}^{N_t} \left\{ \hat{\kappa}_{t-1}(\boldsymbol{X}_{ti}) - \kappa_{t-1}(\boldsymbol{X}_{ti}) \right\} \cdot \left[ \frac{\mathbf{1}\{a_{ti} = \widehat{\pi}_{t-1}(\boldsymbol{X}_{ti})\}\left\{ r_{ti} - \widehat{\mu}_{t-1}^{(t,i)}(\boldsymbol{X}_t, \widehat{\pi}_{t-1}(\boldsymbol{X}_t)) \right\}}{\{1 - \hat{\kappa}_{t-1}(\boldsymbol{X}_{ti})\}\{1 - \kappa_{t-1}(\boldsymbol{X}_{ti})\}} \right].$$

This can be further decomposed to two parts:

$$\underbrace{\frac{1}{\bar{N}_T} \sum_{t=1}^{T} \sum_{i=1}^{N_t} \left\{ \hat{\kappa}_{t-1}(\boldsymbol{X}_{ti}) - \kappa_{t-1}(\boldsymbol{X}_{ti}) \right\} \cdot \left[ \frac{\mathbf{1}\{a_{ti} = \widehat{\pi}_{t-1}(\boldsymbol{X}_{ti})\}\left\{ r_{ti} - \mu^{(t,i)}(\boldsymbol{X}_t, \widehat{\pi}_{t-1}(\boldsymbol{X}_t)) \right\}}{\{1 - \hat{\kappa}_{t-1}(\boldsymbol{X}_{ti})\}\{1 - \kappa_{t-1}(\boldsymbol{X}_{ti})\}} \right]}_{\Delta_1}$$

$$+ \underbrace{\frac{1}{\bar{N}_T} \sum_{t=1}^{T} \sum_{i=1}^{N_t} \left\{ \hat{\kappa}_{t-1}(\boldsymbol{X}_{ti}) - \kappa_{t-1}(\boldsymbol{X}_{ti}) \right\} \cdot \left[ \frac{\mathbf{1}\{a_{ti} = \widehat{\pi}_{t-1}(\boldsymbol{X}_{ti})\}\left\{ \mu^{(t,i)}(\boldsymbol{X}_t, \widehat{\pi}_{t-1}(\boldsymbol{X}_t)) - \widehat{\mu}_{t-1}^{(t,i)}(\boldsymbol{X}_t, \widehat{\pi}_{t-1}(\boldsymbol{X}_t)) \right\}}{\{1 - \hat{\kappa}_{t-1}(\boldsymbol{X}_{ti})\}\{1 - \kappa_{t-1}(\boldsymbol{X}_{ti})\}} \right]}_{\Delta_2}.$$

We first show that $\Delta_1$ is $o_p(T^{-1/2})$. Similar to the proof of Theorem 3 in Ye et al. (2023), we define a class of measurable functions

$$\mathcal{F}(\boldsymbol{X}_t, a_{ti}, r_{ti}) = \left\{ \left\{ \hat{\kappa}_{t-1}(\boldsymbol{X}_{ti}) - \kappa_{t-1}(\boldsymbol{X}_{ti}) \right\} \cdot \left[ \frac{\mathbf{1}\{a_{ti} = \widehat{\pi}_{t-1}(\boldsymbol{X}_{ti})\}\left\{ r_{ti} - \mu^{(t,i)}(\boldsymbol{X}_t, \widehat{\pi}_{t-1}(\boldsymbol{X}_t)) \right\}}{\{1 - \hat{\kappa}_{t-1}(\boldsymbol{X}_{ti})\}\{1 - \kappa_{t-1}(\boldsymbol{X}_{ti})\}} \right] : \hat{\kappa}_t, \kappa_t \in \Lambda, \widehat{\pi}_t \in \Pi \right\},$$

where $\Lambda$ and $\Pi$ are two classes of functions mapping context $\boldsymbol{X}_{ti}$ to a probability in $[0, 1]$. Denote the empirical measure $\mathbb{G}_n = \sqrt{n}\mathbb{P}_n(f - \mathbb{P}f)$. Here, $n = Q(t, i)$ is the sample index, which is determined by reordering the units $i \in \{1, \ldots, N_t\}$ according to time $t$. Denote $\|z\|_{\mathcal{F}} = \sup_{f \in \mathcal{F}} |z(f)|$. Therefore,

$$\|\mathbb{G}_n\|_{\mathcal{F}} := \sup_{\pi \in \Pi} \left| \frac{1}{\sqrt{\bar{N}_T}} \sum_{q \in Q(t,i)} \left[ \mathcal{F}(\boldsymbol{X}_t, a_{ti}, r_{ti}) - \mathbb{E}\{\mathcal{F}(\boldsymbol{X}_t, a_{ti}, r_{ti}) \mid \mathcal{H}_{t-1}\} \right] \right|. \tag{58}$$

Since $\mu^{(t,i)}$ in $\mathcal{F}$ is correctly specified, we always have

$$\mathbb{E}\{\mathcal{F}(\boldsymbol{X}_t, a_{ti}, r_{ti}) \mid \mathcal{H}_{t-1}\}$$

$$= \mathbb{E}\left[ \left\{ \hat{\kappa}_{t-1}(\boldsymbol{X}_{ti}) - \kappa_{t-1}(\boldsymbol{X}_{ti}) \right\} \cdot \left\{ \frac{\mathbf{1}\{a_{ti} = \widehat{\pi}_{t-1}(\boldsymbol{X}_{ti})\}\left\{ r_{ti} - \mu^{(t,i)}(\boldsymbol{X}_t, \widehat{\pi}_{t-1}(\boldsymbol{X}_t)) \right\}}{\{1 - \hat{\kappa}_{t-1}(\boldsymbol{X}_{ti})\}\{1 - \kappa_{t-1}(\boldsymbol{X}_{ti})\}} \right\} \,\Big|\, \mathcal{H}_{t-1} \right].$$

$$= \mathbb{E}\left[ \left\{ \hat{\kappa}_{t-1}(\boldsymbol{X}_{ti}) - \kappa_{t-1}(\boldsymbol{X}_{ti}) \right\} \cdot \left\{ \frac{\mathbf{1}\{a_{ti} = \widehat{\pi}_{t-1}(\boldsymbol{X}_{ti})\} \cdot e_{ti}}{\{1 - \hat{\kappa}_{t-1}(\boldsymbol{X}_{ti})\}\{1 - \kappa_{t-1}(\boldsymbol{X}_{ti})\}} \right\} \,\Big|\, \mathcal{H}_{t-1} \right]$$

According to the iteration of expectation, the equality above can be further derived as

$$
\begin{aligned}
&\mathbb{E}\{\mathcal{F}(\boldsymbol{X}_t, a_{ti}, r_{ti}) \mid \mathcal{H}_{t-1}\}\\
&= \mathbb{E}\left[\{\hat{\kappa}_{t-1}(\boldsymbol{X}_{ti}) - \kappa_{t-1}(\boldsymbol{X}_{ti})\} \cdot \left\{\frac{\mathbb{E}[\mathbf{1}\{a_{ti} = \widehat{\pi}_{t-1}(\boldsymbol{X}_{ti})\}|\boldsymbol{X}_{ti}]}{\{1 - \hat{\kappa}_{t-1}(\boldsymbol{X}_{ti})\}\{1 - \kappa_{t-1}(\boldsymbol{X}_{ti})\}} \cdot \mathbb{E}[e_{ti}|\boldsymbol{X}_{ti}, \boldsymbol{A}_t]\right\} \Big| \mathcal{H}_{t-1}\right] = 0,
\end{aligned}
$$

where the last equality holds by $\mathbb{E}[\eta_{ti}|\boldsymbol{X}_{ti}, \boldsymbol{A}_t] = 0$ according to the definition of noise $\eta_{ti}$.

Therefore, Equation (58) can be simplified as

$$
\|\mathbb{G}_n\|_{\mathcal{F}} = \sup_{\pi \in \Pi}\left|\frac{1}{\sqrt{\bar{N}_T}} \sum_{q \in Q(t,i)} \mathcal{F}(\boldsymbol{X}_t, a_{ti}, r_{ti})\right|.
$$

Following Section 4.2 of Dedecker & Louhichi (2002), we define

$$
d_1(f) := \left\|\mathbb{E}\{|f(\boldsymbol{X}_1, a_{11}, r_{11})| \big| \mathcal{H}_0\}\right\|_{\infty}, \quad d_2(f) := \left\|\mathbb{E}\{(f(\boldsymbol{X}_1, a_{11}, r_{11}))^2|\mathcal{H}_0\}\right\|_{\infty}^{1/2}.
$$

First, we show that both $d_1(f)$ and $d_2(f)$ are finite numbers. For the brevity of content, we will take $d_2(f)$ as an example and $d_1(f) < \infty$ can be proved similarly.

In a valid bandits algorithm, the probability of exploration $\kappa_t(\boldsymbol{X}_{ti})$ is bounded away from 1. That is, there exists a constant $C_1 < 1$, such that $\kappa_t(\boldsymbol{X}_{ti}) = \mathbb{P}(a_{ti} \neq \widehat{\pi}_{t-1}(\boldsymbol{X}_{ti})) \leq C_1 < 1$, and $\hat{\kappa}_t(\boldsymbol{X}_{ti}) \leq C_1 < 1$. Therefore, for any $t \in \{1, \ldots, T\}$,

$$
\begin{aligned}
&\mathbb{E}\{(f(\boldsymbol{X}_t, a_{ti}, r_{ti}))^2|\mathcal{H}_{t-1}\}\\
&= \mathbb{E}\left[\left\{\{\hat{\kappa}_{t-1}(\boldsymbol{X}_{ti}) - \kappa_{t-1}(\boldsymbol{X}_{ti})\} \cdot \frac{\mathbf{1}\{a_{ti} = \widehat{\pi}_{t-1}(\boldsymbol{X}_{ti})\}\{r_{ti} - \mu^{(t,i)}(\boldsymbol{X}_t, \widehat{\pi}_{t-1}(\boldsymbol{X}_t))\}}{\{1 - \hat{\kappa}_{t-1}(\boldsymbol{X}_{ti})\}\{1 - \kappa_{t-1}(\boldsymbol{X}_{ti})\}}\right\}^2 \Big| \mathcal{H}_{t-1}\right]\\
&= \mathbb{E}\left[\left\{\frac{\hat{\kappa}_{t-1}(\boldsymbol{X}_{ti}) - \kappa_{t-1}(\boldsymbol{X}_{ti})}{\{1 - \hat{\kappa}_{t-1}(\boldsymbol{X}_{ti})\}\{1 - \kappa_{t-1}(\boldsymbol{X}_{ti})\}}\right\}^2 \cdot \mathbf{1}\{a_{ti} = \widehat{\pi}_{t-1}(\boldsymbol{X}_{ti})\} \cdot \{r_{ti} - \mu^{(t,i)}(\boldsymbol{X}_t, \widehat{\pi}_{t-1}(\boldsymbol{X}_t))\}^2 \Big| \mathcal{H}_{t-1}\right]\\
&= \mathbb{E}\left[\left\{\frac{\hat{\kappa}_{t-1}(\boldsymbol{X}_{ti}) - \kappa_{t-1}(\boldsymbol{X}_{ti})}{\{1 - \hat{\kappa}_{t-1}(\boldsymbol{X}_{ti})\}\{1 - \kappa_{t-1}(\boldsymbol{X}_{ti})\}}\right\}^2 \cdot \mathbf{1}\{a_{ti} = \widehat{\pi}_{t-1}(\boldsymbol{X}_{ti})\} \cdot \mathbb{E}[\eta_{ti}^2|\boldsymbol{X_t}, \boldsymbol{A}_t] \Big| \mathcal{H}_{t-1}\right]\\
&\leq \left(\frac{2}{1 - C_1}\right)^2 \cdot 1 \cdot \sigma^2 < \infty.
\end{aligned}
$$

Therefore, by Rosenthal's inequality derived for Martingale [see Dedecker & Louhichi (2002) for details], we have

$$
\mathbb{E}\left[\|\mathbb{G}_n\|_{\mathcal{F}}\right] \leq K\left(d_2(f) + \frac{1}{\sqrt{\bar{N}_T}}\left\|\max_{q \in Q(t,i)}\left|\mathcal{F}(\boldsymbol{X}_t, a_{ti}, r_{ti}) - \mathbb{E}\{\mathcal{F}(\boldsymbol{X}_t, a_{ti}, r_{ti}) \mid \mathcal{H}_{t-1}\}\right|\right\|_1\right) \tag{59}
$$

Since the right hand side is $O_p(T^{-1/2})$, we have

$$
\Delta_1 = \frac{1}{\sqrt{\bar{N}_T}}\sum_q \mathcal{F}(\boldsymbol{X}_t, a_{ti}, r_{ti}) \leq \frac{1}{\sqrt{\bar{N}_T}}\mathbb{E}\left[\|\mathbb{G}_n\|_{\mathcal{F}}\right] = O_p(T^{-1}) = o_p(T^{-1/2}). \tag{60}
$$

Now let's derive the order for $\Delta_2$.

$$
\begin{aligned}
\Delta_2 &= \frac{1}{\bar{N}_T} \sum_{t=1}^{T} \sum_{i=1}^{N_t} \left\{ \hat{\kappa}_{t-1}(\boldsymbol{X}_{ti}) - \kappa_{t-1}(\boldsymbol{X}_{ti}) \right\} \cdot \left[ \frac{\mathbf{1}\{a_{ti} = \hat{\pi}_{t-1}(\boldsymbol{X}_{ti})\} \left\{ \mu^{(t,i)}(\boldsymbol{X}_t, \hat{\pi}_{t-1}(\boldsymbol{X}_t)) - \hat{\mu}_{t-1}^{(t,i)}(\boldsymbol{X}_t, \hat{\pi}_{t-1}(\boldsymbol{X}_t)) \right\}}{\{1 - \hat{\kappa}_{t-1}(\boldsymbol{X}_{ti})\}\{1 - \kappa_{t-1}(\boldsymbol{X}_{ti})\}} \right] \\
&\leq \frac{1}{\bar{N}_T} \sum_{t=1}^{T} \sum_{i=1}^{N_t} C_2 \left| \hat{\kappa}_{t-1}(\boldsymbol{X}_{ti}) - \kappa_{t-1}(\boldsymbol{X}_{ti}) \right| \cdot \left| \hat{\mu}_{t-1}^{(t,i)}(\boldsymbol{X}_t, \hat{\pi}_{t-1}(\boldsymbol{X}_t)) - \mu^{(t,i)}(\boldsymbol{X}_t, \hat{\pi}_{t-1}(\boldsymbol{X}_t)) \right| \\
&\leq C_2 \sqrt{\frac{1}{\bar{N}_T} \sum_{t=1}^{T} \sum_{i=1}^{N_t} \left| \hat{\kappa}_{t-1}(\boldsymbol{X}_{ti}) - \kappa_{t-1}(\boldsymbol{X}_{ti}) \right|^2 \cdot \frac{1}{\bar{N}_T} \sum_{t=1}^{T} \sum_{i=1}^{N_t} \left| \hat{\mu}_{t-1}^{(t,i)}(\boldsymbol{X}_t, \hat{\pi}_{t-1}(\boldsymbol{X}_t)) - \mu^{(t,i)}(\boldsymbol{X}_t, \hat{\pi}_{t-1}(\boldsymbol{X}_t)) \right|^2} \\
&= o_p(\bar{N}_T^{-1/2}),
\end{aligned}
\tag{61}
$$

where the last line holds by Cauchy-Schwartz inequality, and the last line holds by Assumption A.4.

Combining the results of Equation (60) and (61), we have

$$
\widehat{V}_T - \check{V}_T = \Delta_1 + \Delta_2 = o_p(\bar{N}_T^{-1/2}) + o_p(\bar{N}_T^{-1/2}) = o_p(\bar{N}_T^{-1/2}).
\tag{62}
$$

Now the question becomes proving (2) $\check{V}_T = \widetilde{V}_T + o_p(\bar{N}_T^{-1/2})$.

$$
\check{V}_T - \widetilde{V}_T = \frac{1}{\bar{N}_T} \sum_{t=1}^{T} \sum_{i=1}^{N_t} \left[ 1 - \frac{\mathbf{1}\{a_{ti} = \hat{\pi}_{t-1}(\boldsymbol{X}_{ti})\}}{1 - \kappa_{t-1}(\boldsymbol{X}_{ti})} \right] \cdot \left\{ \hat{\mu}_{t-1}^{(t,i)}(\boldsymbol{X}_t, \hat{\pi}_{t-1}(\boldsymbol{X}_t)) - \mu^{(t,i)}(\boldsymbol{X}_t, \hat{\pi}_{t-1}(\boldsymbol{X}_t)) \right\}.
$$

Following similar structure as we prove $\Delta_1 = o_p(\bar{N}_T^{-1/2})$, one can define a new class of functions

$$
\mathcal{F}'(\boldsymbol{X}_t, a_{ti}, r_{ti}) = \left\{ \left[ 1 - \frac{\mathbf{1}\{a_{ti} = \hat{\pi}_{t-1}(\boldsymbol{X}_{ti})\}}{1 - \kappa_{t-1}(\boldsymbol{X}_{ti})} \right] \cdot \left\{ \hat{\mu}_{t-1}^{(t,i)}(\boldsymbol{X}_t, \hat{\pi}_{t-1}(\boldsymbol{X}_t)) - \mu^{(t,i)}(\boldsymbol{X}_t, \hat{\pi}_{t-1}(\boldsymbol{X}_t)) \right\} : \hat{\mu}_{t-1}^{(t,i)}, \mu \in \Lambda, \hat{\pi}_t \in \Pi \right\},
$$

and using Rosenthal's inequality for Martingale to prove that $\check{V}_T - \widetilde{V}_T = o_p(\bar{N}_T^{-1/2})$.

**Step 2:** Prove that $\widetilde{V}_T = \bar{V}_T + o_p(\bar{N}_T^{-1/2})$.

By definition of $\widetilde{V}_T$ and $\bar{V}_T$, we have

$$
\begin{aligned}
\sqrt{\bar{N}_T}(\widetilde{V}_T - \bar{V}_T) &= \underbrace{\frac{1}{\sqrt{\bar{N}_T}} \sum_{t=1}^{T} \sum_{i=1}^{N_t} \left[ \frac{\mathbf{1}\{a_{ti} = \hat{\pi}_{t-1}(\boldsymbol{X}_{ti})\}}{1 - \kappa_{t-1}(\boldsymbol{X}_{ti})} - 1 \right] \cdot \left\{ \mu^{(t,i)}(\boldsymbol{X}_t, \pi^*(\boldsymbol{X}_t)) - \mu^{(t,i)}(\boldsymbol{X}_t, \hat{\pi}_{t-1}(\boldsymbol{X}_t)) \right\}}_{\Delta_3} \\
&+ \underbrace{\frac{1}{\sqrt{\bar{N}_T}} \sum_{t=1}^{T} \sum_{i=1}^{N_t} \left[ \frac{\mathbf{1}\{a_{ti} = \hat{\pi}_{t-1}(\boldsymbol{X}_{ti})\}}{1 - \kappa_{t-1}(\boldsymbol{X}_{ti})} - \frac{\mathbf{1}\{a_{ti} = \pi^*(\boldsymbol{X}_{ti})\}}{\mathbb{P}(a_{ti} = \pi^*(\boldsymbol{X}_{ti}))} \right] \cdot \left\{ r_{ti} - \mu^{(t,i)}(\boldsymbol{X}_t, \pi^*(\boldsymbol{X}_t)) \right\}}_{\Delta_4}.
\end{aligned}
\tag{63}
$$

**Step 2.1:** We start from proving $\Delta_3 = o_p(1)$. Since $\kappa_t(\boldsymbol{X}_{ti}) \leq C_1 < 1$, $\left| \frac{\mathbf{1}\{a_{ti} = \hat{\pi}_{t-1}(\boldsymbol{X}_{ti})\}}{1 - \kappa_{t-1}(\boldsymbol{X}_{ti})} - 1 \right|$ is upper bounded by a constant. Therefore, to prove $\Delta_3 = o_p(1)$, it suffice to show that

$$
\frac{1}{\sqrt{\bar{N}_T}} \sum_{t=1}^{T} \sum_{i=1}^{N_t} \left[ \mu^{(t,i)}(\boldsymbol{X}_t, \pi^*(\boldsymbol{X}_t)) - \mu^{(t,i)}(\boldsymbol{X}_t, \hat{\pi}_{t-1}(\boldsymbol{X}_t)) \right] = o_p(1).
\tag{64}
$$

Before proceeding, let's break down this term to do some transformation. Notice that

$$
\begin{aligned}
\sum_{i=1}^{N_t} \mu^{(t,i)}(\boldsymbol{X}_t, \widehat{\pi}_{t-1}(\boldsymbol{X}_t)) &= \sum_{i=1}^{N_t} \sum_{j=1}^{N_t} \sum_{a \in \mathcal{A}} W_{t,ij} \boldsymbol{X}_{tj}^\top \boldsymbol{\beta}_a \cdot \mathbf{1}\{\widehat{\pi}_{t-1}(\boldsymbol{X}_{tj}) = a\} \\
&= \sum_{j=1}^{N_t} \sum_{i=1}^{N_t} \sum_{a \in \mathcal{A}} W_{t,ji} \boldsymbol{X}_{ti}^\top \boldsymbol{\beta}_a \cdot \mathbf{1}\{\widehat{\pi}_{t-1}(\boldsymbol{X}_{ti}) = a\} \\
&= \sum_{i=1}^{N_t} \sum_{a \in \mathcal{A}} \Big\{ \sum_{j=1}^{N_t} W_{t,ji} \Big\} \cdot \boldsymbol{X}_{ti}^\top \boldsymbol{\beta}_a \cdot \mathbf{1}\{\widehat{\pi}_{t-1}(\boldsymbol{X}_{ti}) = a\} \\
&= \sum_{i=1}^{N_t} \sum_{a \in \mathcal{A}} \omega_{ti} \boldsymbol{X}_{ti}^\top \boldsymbol{\beta}_a \cdot \mathbf{1}\{\widehat{\pi}_{t-1}(\boldsymbol{X}_{ti}) = a\}
\end{aligned}
\tag{65}
$$

where the second equality holds by switching the index of $(i,j)$ to $(j,i)$, and the third equality holds by Fubini's theorem. Going back to the previous equation, we have

$$
\begin{aligned}
&\frac{1}{\sqrt{\bar{N}_T}} \sum_{t=1}^{T} \sum_{i=1}^{N_t} \Big[ \mu^{(t,i)}(\boldsymbol{X}_t, \pi^*(\boldsymbol{X}_t)) - \mu^{(t,i)}(\boldsymbol{X}_t, \widehat{\pi}_{t-1}(\boldsymbol{X}_t)) \Big] \\
&= \frac{1}{\sqrt{\bar{N}_T}} \sum_{t=1}^{T} \sum_{i=1}^{N_t} \sum_{a \in \mathcal{A}} \Big[ \omega_{ti} \boldsymbol{X}_{ti}^\top \boldsymbol{\beta}_a \cdot \mathbf{1}\{\pi^*(\boldsymbol{X}_{ti}) = a\} - \omega_{ti} \boldsymbol{X}_{ti}^\top \boldsymbol{\beta}_a \cdot \mathbf{1}\{\widehat{\pi}_{t-1}(\boldsymbol{X}_{ti}) = a\} \Big] \\
&= \frac{1}{\sqrt{\bar{N}_T}} \sum_{t=1}^{T} \sum_{i=1}^{N_t} \sum_{1 \leq a,a' \leq K} \omega_{ti} \boldsymbol{X}_{ti}^\top (\boldsymbol{\beta}_a - \boldsymbol{\beta}_{a'}) \cdot \mathbf{1}\{\pi^*(\boldsymbol{X}_{ti}) = a, \widehat{\pi}_{t-1}(\boldsymbol{X}_{ti}) = a', a \neq a'\} \\
&\leq \frac{1}{\sqrt{\bar{N}_T}} \sum_{t=1}^{T} \sum_{i=1}^{N_t} \sum_{\substack{1 \leq a,a' \leq K \\ a \neq a'}} \Big| \omega_{ti} \boldsymbol{X}_{ti}^\top (\boldsymbol{\beta}_a - \boldsymbol{\beta}_{a'}) \cdot \mathbf{1}\{\omega_{ti} \boldsymbol{X}_{ti}^\top (\widehat{\boldsymbol{\beta}}_{t-1,a} - \widehat{\boldsymbol{\beta}}_{t-1,a'}) \leq 0\} \Big|
\end{aligned}
$$

Again, for the brevity of notation, we denote $\widehat{\zeta}_{ti} = \omega_{ti} \boldsymbol{X}_{ti}^\top (\widehat{\boldsymbol{\beta}}_{t-1,a} - \widehat{\boldsymbol{\beta}}_{t-1,a'})$, and $\zeta_{ti} = \omega_{ti} \boldsymbol{X}_{ti}^\top (\boldsymbol{\beta}_a - \boldsymbol{\beta}_{a'})$.

Let's first consider the case where $\zeta_{ti} > 0$. The opposite scenario can be derived in a similar manner. When $\zeta_{ti} > 0$, the RHS of the above equation is thus equivalent to

$$
\frac{1}{\sqrt{\bar{N}_T}} \sum_{t=1}^{T} \sum_{i=1}^{N_t} \sum_{\substack{1 \leq a,a' \leq K \\ a \neq a'}} \Big| \mathbf{1}\{\widehat{\zeta}_{ti} \leq 0\} \cdot \zeta_{ti} \Big| = \frac{1}{\sqrt{\bar{N}_T}} \sum_{t=1}^{T} \sum_{i=1}^{N_t} \sum_{\substack{1 \leq a,a' \leq K \\ a \neq a'}} \mathbf{1}\{\widehat{\zeta}_{ti} \leq 0\} \cdot \zeta_{ti}.
$$

Since $\mathbf{1}\{\widehat{\zeta}_{ti} \leq 0\} \cdot \widehat{\zeta}_{ti} \leq 0$, we have

$$
\Big| \mu^{(t,i)}(\boldsymbol{X}_t, \pi^*(\boldsymbol{X}_t)) - \mu^{(t,i)}(\boldsymbol{X}_t, \widehat{\pi}_{t-1}(\boldsymbol{X}_t)) \Big| \leq \mathbf{1}\{\widehat{\zeta}_{ti} \leq 0\} \cdot \zeta_{ti} \leq \mathbf{1}\{\widehat{\zeta}_{ti} \leq 0\} \cdot (\zeta_{ti} - \widehat{\zeta}_{ti}).
$$

To show that $\bar{N}_T^{-1/2} \sum_{t=1}^{T} \sum_{i=1}^{N_t} \big| \mu^{(t,i)}(\boldsymbol{X}_t, \pi^*(\boldsymbol{X}_t)) - \mu^{(t,i)}(\boldsymbol{X}_t, \widehat{\pi}_{t-1}(\boldsymbol{X}_t)) \big| = o_p(1)$, it suffice to prove

$$
\zeta := \bar{N}_T^{-1/2} \sum_{t=1}^{T} \sum_{i=1}^{N_t} \mathbf{1}\{\widehat{\zeta}_{ti} \leq 0\} \cdot (\zeta_{ti} - \widehat{\zeta}_{ti}) = o_p(1).
$$

For any $\alpha \in (0, 1/2)$, we can further decompose

$$\boldsymbol{\zeta} = \underbrace{\mathbb{P}(0 < \zeta_{ti} < \bar{N}_T^{-\alpha}) \cdot \frac{1}{\sqrt{\bar{N}_T}} \sum_{t=1}^{T} \sum_{i=1}^{N_t} \mathbf{1}\{\widehat{\zeta}_{ti} \leq 0\} \cdot (\zeta_{ti} - \widehat{\zeta}_{ti}) \cdot \mathbf{1}\{0 < \zeta_{ti} < \bar{N}_T^{-\alpha}\}}_{\boldsymbol{\zeta}_1}$$

$$+ \underbrace{\mathbb{P}(\zeta_{ti} \geq \bar{N}_T^{-\alpha}) \cdot \frac{1}{\sqrt{\bar{N}_T}} \sum_{t=1}^{T} \sum_{i=1}^{N_t} \mathbf{1}\{\widehat{\zeta}_{ti} \leq 0\} \cdot (\zeta_{ti} - \widehat{\zeta}_{ti}) \cdot \mathbf{1}\{\zeta_{ti} \geq \bar{N}_T^{-\alpha}\}}_{\boldsymbol{\zeta}_2}.$$

First, we show $\boldsymbol{\zeta}_1 = o_p(1)$. According to Theorem 4.3, $\widehat{\zeta}_{ti} - \zeta_{ti} = O_p(\bar{N}_t^{-1/2}) = o_p(\bar{N}_t^{-(1/2-\alpha\gamma)})$ for any $\alpha\gamma > 0$. Therefore,

$$\frac{1}{\sqrt{\bar{N}_T}} \sum_{t=1}^{T} \sum_{i=1}^{N_t} \left| \mathbf{1}\{\widehat{\zeta}_{ti} \leq 0\} \cdot (\zeta_{ti} - \widehat{\zeta}_{ti}) \cdot \mathbf{1}\{0 < \zeta_{ti} < \bar{N}_T^{-\alpha}\} \right| \leq \frac{1}{\sqrt{\bar{N}_T}} \sum_{t=1}^{T} \sum_{i=1}^{N_t} \left| \zeta_{ti} - \widehat{\zeta}_{ti} \right|$$

$$\leq \sqrt{\bar{N}_T} \cdot \frac{1}{\bar{N}_T} \sum_{t=1}^{T} \sum_{i=1}^{N_t} \left| \zeta_{ti} - \widehat{\zeta}_{ti} \right| = \sqrt{\bar{N}_T} \cdot o_p(\bar{N}_T^{-(1/2-\alpha\gamma)}) = o_p(\bar{N}_T^{\alpha\gamma}),$$

where the second last equality holds by Lemma 6 in Luedtke & Van Der Laan (2016).

Since $|\omega| \geq 1$, by setting $\epsilon = \bar{N}_T^{-\alpha}$ in Assumption A.3, we have $\mathbb{P}\left(0 < |\omega f(\boldsymbol{X}, a) - \omega f(\boldsymbol{X}, a')| < \bar{N}_T^{-\alpha}\right) \leq \mathbb{P}\left(0 < |f(\boldsymbol{X}, a) - f(\boldsymbol{X}, a')| < \bar{N}_T^{-\alpha}\right) = O(\bar{N}_T^{-\alpha\gamma})$. Therefore,

$$\boldsymbol{\zeta}_1 = \mathbb{P}(0 < \zeta_{ti} < \bar{N}_T^{-\alpha}) \cdot \frac{1}{\sqrt{\bar{N}_T}} \sum_{t=1}^{T} \sum_{i=1}^{N_t} \mathbf{1}\{\widehat{\zeta}_{ti} \leq 0\} \cdot (\zeta_{ti} - \widehat{\zeta}_{ti}) \cdot \mathbf{1}\{0 < \zeta_{ti} < \bar{N}_T^{-\alpha}\}$$

$$\leq \left| \mathbb{P}(0 < \zeta_{ti} < \bar{N}_T^{-\alpha}) \right| \cdot \frac{1}{\sqrt{\bar{N}_T}} \sum_{t=1}^{T} \sum_{i=1}^{N_t} \left| \mathbf{1}\{\widehat{\zeta}_{ti} \leq 0\} \cdot (\zeta_{ti} - \widehat{\zeta}_{ti}) \cdot \mathbf{1}\{0 < \zeta_{ti} < \bar{N}_T^{-\alpha}\} \right| \qquad (66)$$

$$\leq O(\bar{N}_T^{-\alpha\gamma}) \cdot o_p(\bar{N}_T^{\alpha\gamma}) = o_p(1).$$

Next, we show $\boldsymbol{\zeta}_2 = o_p(1)$.

Since $\mathbf{1}\{\widehat{\zeta}_{ti} \leq 0\} = \mathbf{1}\{\widehat{\zeta}_{ti} - \zeta_{ti} \leq -\zeta_{ti}\} = \mathbf{1}\{|\widehat{\zeta}_{ti} - \zeta_{ti}| > \zeta_{ti}\}$, we have

$$\left| \mathbf{1}\{\widehat{\zeta}_{ti} \leq 0\}(\widehat{\zeta}_{ti} - \zeta_{ti}) \right| = \mathbf{1}\{|\widehat{\zeta}_{ti} - \zeta_{ti}| > \zeta_{ti}\} \cdot |\widehat{\zeta}_{ti} - \zeta_{ti}| \leq \frac{|\widehat{\zeta}_{ti} - \zeta_{ti}|}{\zeta_{ti}} \cdot |\widehat{\zeta}_{ti} - \zeta_{ti}| = \frac{|\widehat{\zeta}_{ti} - \zeta_{ti}|^2}{\zeta_{ti}}. \qquad (67)$$

Since we assumed that $\zeta_{ti} > 0$, based on the result of Equation (67), we further have

$$\mathbf{1}\{\widehat{\zeta}_{ti} \leq 0\}(\zeta_{ti} - \widehat{\zeta}_{ti}) \leq \frac{|\widehat{\zeta}_{ti} - \zeta_{ti}|^2}{\zeta_{ti}}$$

as $\zeta_{ti} - \widehat{\zeta}_{ti} \geq 0$ always holds. Additionally, notice that $\mathbf{1}\{\zeta_{ti} \geq \bar{N}_T^{-\alpha}\} \leq \zeta_{ti} \bar{N}_T^{\alpha}$. Therefore,

$$\boldsymbol{\zeta}_2 = \mathbb{P}(\zeta_{ti} \geq \bar{N}_T^{-\alpha}) \cdot \frac{1}{\sqrt{\bar{N}_T}} \sum_{t=1}^{T} \sum_{i=1}^{N_t} \mathbf{1}\{\widehat{\zeta}_{ti} \leq 0\} \cdot (\zeta_{ti} - \widehat{\zeta}_{ti}) \cdot \mathbf{1}\{\zeta_{ti} \geq \bar{N}_T^{-\alpha}\}$$

$$\leq \frac{1}{\sqrt{\bar{N}_T}} \sum_{t=1}^{T} \sum_{i=1}^{N_t} \frac{|\widehat{\zeta}_{ti} - \zeta_{ti}|^2}{\zeta_{ti}} \cdot \zeta_{ti} \bar{N}_T^{\alpha} = \bar{N}_T^{1/2+\alpha} \cdot \frac{1}{\bar{N}_T} \sum_{t=1}^{T} \sum_{i=1}^{N_t} |\widehat{\zeta}_{ti} - \zeta_{ti}|^2.$$

By Theorem 4.3, $|\widehat{\zeta}_{ti} - \zeta_{ti}| = O_p(\bar{N}_t^{-1/2})$, which implies $|\widehat{\zeta}_{ti} - \zeta_{ti}|^2 = O_p(\bar{N}_t^{-1})$. According to Lemma 6 of Luedtke & Van Der Laan (2016), $\bar{N}_T^{-1} \sum_{t=1}^{T} \sum_{i=1}^{N_t} |\widehat{\zeta}_{ti} - \zeta_{ti}|^2 = O_p(\bar{N}_T^{-1})$. Therefore,

$$\boldsymbol{\zeta}_2 \leq \bar{N}_T^{1/2+\alpha} \cdot \frac{1}{\bar{N}_T} \sum_{t=1}^{T} \sum_{i=1}^{N_t} |\widehat{\zeta}_{ti} - \zeta_{ti}|^2 \leq \bar{N}_T^{1/2+\alpha} \cdot O_p(\bar{N}_T^{-1}) = o_p(1) \qquad (68)$$

for any $\alpha < 1/2$.

Combining the result of Equation (66) and Equation (68), we have

$$\zeta = \zeta_1 + \zeta_2 = o_p(1) + o_p(1) = o_p(1). \tag{69}$$

Therefore, $\bar{N}_T^{-1/2} \sum_{t=1}^T \sum_{i=1}^{N_t} \left| \mu^{(t,i)}(\boldsymbol{X}_t, \pi^*(\boldsymbol{X}_t)) - \mu^{(t,i)}(\boldsymbol{X}_t, \widehat{\pi}_{t-1}(\boldsymbol{X}_t)) \right| = o_p(1)$, and thus $\Delta_3 = o_p(1)$. The proof of first part is done.

**Step 2.2:** Next, we show that $\Delta_4 = o_p(1)$ as well.

Recall that

$$
\begin{aligned}
\Delta_4 &= \frac{1}{\sqrt{\bar{N}_T}} \sum_{t=1}^T \sum_{i=1}^{N_t} \left[ \frac{\mathbf{1}\{a_{ti} = \widehat{\pi}_{t-1}(\boldsymbol{X}_{ti})\}}{1 - \kappa_{t-1}(\boldsymbol{X}_{ti})} - \frac{\mathbf{1}\{a_{ti} = \pi^*(\boldsymbol{X}_{ti})\}}{\mathbb{P}(a_{ti} = \pi^*(\boldsymbol{X}_{ti}))} \right] \cdot \left\{ r_{ti} - \mu^{(t,i)}(\boldsymbol{X}_t, \pi^*(\boldsymbol{X}_t)) \right\} \\
&= \frac{1}{\sqrt{\bar{N}_T}} \sum_{t=1}^T \sum_{i=1}^{N_t} \left[ \frac{\mathbf{1}\{a_{ti} = \widehat{\pi}_{t-1}(\boldsymbol{X}_{ti})\}}{1 - \kappa_{t-1}(\boldsymbol{X}_{ti})} - \frac{\mathbf{1}\{a_{ti} = \pi^*(\boldsymbol{X}_{ti})\}}{\mathbb{P}(a_{ti} = \pi^*(\boldsymbol{X}_{ti}))} \right] \cdot \left\{ r_{ti} - \mu^{(t,i)}(\boldsymbol{X}_t, \pi^*(\boldsymbol{X}_t)) \right\} \mathbf{1}\{a_{ti} = \pi^*(\boldsymbol{X}_{ti})\} \\
&\quad + \frac{1}{\sqrt{\bar{N}_T}} \sum_{t=1}^T \sum_{i=1}^{N_t} \left[ \frac{\mathbf{1}\{a_{ti} = \widehat{\pi}_{t-1}(\boldsymbol{X}_{ti})\}}{1 - \kappa_{t-1}(\boldsymbol{X}_{ti})} \right] \cdot \left\{ r_{ti} - \mu^{(t,i)}(\boldsymbol{X}_t, \pi^*(\boldsymbol{X}_t)) \right\} \cdot \mathbf{1}\{a_{ti} \neq \pi^*(\boldsymbol{X}_{ti})\} \\
&= \underbrace{\frac{1}{\sqrt{\bar{N}_T}} \sum_{t=1}^T \sum_{i=1}^{N_t} \left[ \frac{\mathbf{1}\{a_{ti} = \widehat{\pi}_{t-1}(\boldsymbol{X}_{ti})\}}{1 - \kappa_{t-1}(\boldsymbol{X}_{ti})} - \frac{\mathbf{1}\{a_{ti} = \pi^*(\boldsymbol{X}_{ti})\}}{\mathbb{P}(a_{ti} = \pi^*(\boldsymbol{X}_{ti}))} \right] \cdot \eta_{ti} \cdot \mathbf{1}\{a_{ti} = \pi^*(\boldsymbol{X}_{ti})\}}_{\zeta_3} \\
&\quad + \underbrace{\frac{1}{\sqrt{\bar{N}_T}} \sum_{t=1}^T \sum_{i=1}^{N_t} \left[ \frac{\mathbf{1}\{a_{ti} = \widehat{\pi}_{t-1}(\boldsymbol{X}_{ti})\}}{1 - \kappa_{t-1}(\boldsymbol{X}_{ti})} \right] \cdot \left\{ \mu^{(t,i)}(\boldsymbol{X}_t, \widehat{\pi}_{t-1}(\boldsymbol{X}_t)) - \mu^{(t,i)}(\boldsymbol{X}_t, \pi^*(\boldsymbol{X}_t)) \right\} \cdot \mathbf{1}\{a_{ti} \neq \pi^*(\boldsymbol{X}_{ti})\}}_{\zeta_4} \\
&\quad + \underbrace{\frac{1}{\sqrt{\bar{N}_T}} \sum_{t=1}^T \sum_{i=1}^{N_t} \left[ \frac{\mathbf{1}\{a_{ti} = \widehat{\pi}_{t-1}(\boldsymbol{X}_{ti})\}}{1 - \kappa_{t-1}(\boldsymbol{X}_{ti})} \right] \cdot \eta_{ti} \cdot \mathbf{1}\{a_{ti} \neq \pi^*(\boldsymbol{X}_{ti})\}}_{\zeta_5} .
\end{aligned}
\tag{70}
$$

We only need to show $\zeta_3$, $\zeta_4$ and $\zeta_5$ are all $o_p(1)$. The proof for $\zeta_5$ is similar to that for $\zeta_3$ using Rosenthal's inequality for Martingales. Therefore, we will focus on proving $\zeta_3$ and $\zeta_4$, and omit the details for $\zeta_5$ for brevity.

To prove $\zeta_3 = o_p(1)$, we define a function class

$$
\mathcal{F}(\boldsymbol{X}_t, a_{ti}, \eta_{ti}) = \left\{ \left[ \frac{\mathbf{1}\{a_{ti} = \widehat{\pi}_{t-1}(\boldsymbol{X}_{ti})\}}{1 - \kappa_{t-1}(\boldsymbol{X}_{ti})} - \frac{\mathbf{1}\{a_{ti} = \pi^*(\boldsymbol{X}_{ti})\}}{\mathbb{P}(a_{ti} = \pi^*(\boldsymbol{X}_{ti}))} \right] \cdot \eta_{ti} \cdot \mathbf{1}\{a_{ti} = \pi^*(\boldsymbol{X}_{ti})\} : \kappa_t \in \Lambda, \widehat{\pi}_t \in \Pi \right\},
$$

where $\Lambda$ and $\Pi$ are two classes of functions mapping context $\boldsymbol{X}_{ti}$ to a probability in $[0,1]$. Define the supremum of the empirical process indexed by $\mathcal{F}$ as

$$
\|\mathbb{G}_n\|_{\mathcal{F}} := \sup_{\pi \in \Pi} \left| \frac{1}{\sqrt{\bar{N}_T}} \sum_{q \in Q(t,i)} \left[ \mathcal{F}(\boldsymbol{X}_t, a_{ti}, \eta_{ti}) - \mathbb{E}\{ \mathcal{F}(\boldsymbol{X}_t, a_{ti}, \eta_{ti}) \mid \mathcal{H}_{t-1} \} \right] \right|. \tag{71}
$$

Since $\mathbb{E}[\eta_{ti} | \boldsymbol{X}_{ti}, \boldsymbol{A}_t] = 0$, according to the iteration of expectation, the second term in the above equation can be derived as

$$
\begin{aligned}
&\mathbb{E}\{ \mathcal{F}(\boldsymbol{X}_t, a_{ti}, \eta_{ti}) \mid \mathcal{H}_{t-1} \} \\
&= \mathbb{E}\left[ \left[ \frac{\mathbf{1}\{a_{ti} = \widehat{\pi}_{t-1}(\boldsymbol{X}_{ti})\}}{1 - \kappa_{t-1}(\boldsymbol{X}_{ti})} - \frac{\mathbf{1}\{a_{ti} = \pi^*(\boldsymbol{X}_{ti})\}}{\mathbb{P}(a_{ti} = \pi^*(\boldsymbol{X}_{ti}))} \right] \cdot \mathbf{1}\{a_{ti} = \pi^*(\boldsymbol{X}_{ti})\} \cdot \mathbb{E}[\eta_{ti} | \boldsymbol{X}_{ti}, \boldsymbol{A}_t] \, \Big| \, \mathcal{H}_{t-1} \right] = 0,
\end{aligned}
$$

Therefore, Equation (71) can be simplified as

$$\|\mathbb{G}_n\|_{\mathcal{F}} = \sup_{\pi \in \Pi} \left| \frac{1}{\sqrt{\bar{N}_T}} \sum_{q \in Q(t,i)} \mathcal{F}(\boldsymbol{X}_t, a_{ti}, \eta_{ti}) \right|.$$

Following a similar derivation structure as that used between Equation (58) and Equation (59) in Step 1, we have

$$\boldsymbol{\zeta}_3 \leq \mathbb{E}\left[\|\mathbb{G}_n\|_{\mathcal{F}}\right] \leq K \left( d_2(f) + \frac{1}{\sqrt{\bar{N}_T}} \left\| \max_{q \in Q(t,i)} \left| \mathcal{F}(\boldsymbol{X}_t, a_{ti}, \eta_{ti}) - \mathbb{E}\{\mathcal{F}(\boldsymbol{X}_t, a_{ti}, \eta_{ti}) \mid \mathcal{H}_{t-1}\} \right| \right\|_1 \right) = o_p(1). \quad (72)$$

Next, let's prove $\boldsymbol{\zeta}_4 = o_p(1)$. Since both $\frac{\mathbf{1}\{a_{ti} = \hat{\pi}_{t-1}(\boldsymbol{X}_{ti})\}}{1 - \kappa_{t-1}(\boldsymbol{X}_{ti})}$ and $\mathbf{1}\{a_{ti} \neq \pi^*(\boldsymbol{X}_{ti})\}$ can be upper bounded, it suffice to prove that

$$\frac{1}{\sqrt{\bar{N}_T}} \sum_{t=1}^{T} \sum_{i=1}^{N_t} \left\{ \mu^{(t,i)}(\boldsymbol{X}_t, \hat{\pi}_{t-1}(\boldsymbol{X}_t)) - \mu^{(t,i)}(\boldsymbol{X}_t, \pi^*(\boldsymbol{X}_t)) \right\} = o_p(1), \quad (73)$$

which has already been established in Equation (64) in Step 2.1. Therefore, $\boldsymbol{\zeta}_4 = o_p(1)$.

Combining the results above, we have

$$\Delta_4 = \boldsymbol{\zeta}_3 + \boldsymbol{\zeta}_4 + \boldsymbol{\zeta}_5 = o_p(1). \quad (74)$$

The proof of Step 2 is thus complete.

**Step 3:** Prove that $\sqrt{\bar{N}_T}(\bar{V}_T - V^{\pi^*}) \xrightarrow{\mathcal{D}} \mathcal{N}(0, \sigma_V^2)$ and derive the asymptotic variance $\sigma_V^2$.

Recall that

$$\bar{V}_T = \frac{1}{\bar{N}_T} \sum_{t=1}^{T} \sum_{i=1}^{N_t} \left\{ \frac{\mathbf{1}\{a_{ti} = \pi^*(\boldsymbol{X}_{ti})\}}{\mathbb{P}(a_{ti} = \pi^*(\boldsymbol{X}_{ti}))} \cdot \left( r_{ti} - \mu^{(t,i)}(\boldsymbol{X}_t, \pi^*(\boldsymbol{X}_t)) \right) + \mu^{(t,i)}(\boldsymbol{X}_t, \pi^*(\boldsymbol{X}_t)) \right\}$$

$$= \frac{1}{\bar{N}_T} \sum_{t=1}^{T} \sum_{i=1}^{N_t} \left\{ \frac{\mathbf{1}\{a_{ti} = \pi^*(\boldsymbol{X}_{ti})\}}{\mathbb{P}(a_{ti} = \pi^*(\boldsymbol{X}_{ti}))} \cdot \eta_{ti} + \mu^{(t,i)}(\boldsymbol{X}_t, \pi^*(\boldsymbol{X}_t)) \right\}.$$

Given the derivation of Equation (65), we have

$$\sum_{i=1}^{N_t} \mu^{(t,i)}(\boldsymbol{X}_t, \pi^*(\boldsymbol{X}_t)) = \sum_{i=1}^{N_t} \sum_{a \in \mathcal{A}} \omega_{ti} \boldsymbol{X}_{ti}^{\top} \boldsymbol{\beta}_a \cdot \mathbf{1}\{\pi^*(\boldsymbol{X}_{ti}) = a\}. \quad (75)$$

Combining the above term with the expression of $\bar{V}_T$, we have

$$\bar{V}_T = \frac{1}{\bar{N}_T} \sum_{t=1}^{T} \sum_{i=1}^{N_t} \left\{ \frac{\mathbf{1}\{a_{ti} = \pi^*(\boldsymbol{X}_{ti})\}}{\mathbb{P}(a_{ti} = \pi^*(\boldsymbol{X}_{ti}))} \cdot \eta_{ti} + \sum_{a \in \mathcal{A}} \omega_{ti} \boldsymbol{X}_{ti}^{\top} \boldsymbol{\beta}_a \cdot \mathbf{1}\{\pi^*(\boldsymbol{X}_{ti}) = a\} \right\}.$$

To decompose, we define

$$\xi_q := \underbrace{\frac{\mathbf{1}\{a_q = \pi^*(\boldsymbol{X}_q)\}}{\mathbb{P}(a_q = \pi^*(\boldsymbol{X}_q))} \cdot \eta_q}_{\xi_{1q}} + \underbrace{\left[ \sum_{a \in \mathcal{A}} \omega_{ti} \boldsymbol{X}_{ti}^{\top} \boldsymbol{\beta}_a \cdot \mathbf{1}\{\pi^*(\boldsymbol{X}_{ti}) = a\} - V^{\pi^*} \right]}_{\xi_{2q}}, \quad (76)$$

where $q$ denotes an unit in a flattened unit queue $Q(t,i) = \sum_{s=1}^{t-1} N_s + i$. Similar to the the proof of Theorem 4.3, we define $\mathcal{H}_q$ as the $\sigma$−algebra containing the information up to unit $q$ where $\mathcal{H}_{q_0} = \sigma(\boldsymbol{v}^{\top} \widetilde{\boldsymbol{X}}_1 \eta_1, \ldots, \boldsymbol{v}^{\top} \widetilde{\boldsymbol{X}}_{q_0} \eta_{q_0})$.

Since

$$\mathbb{E}[\xi_{2q}] = \mathbb{E}\left[ \sum_{a \in \mathcal{A}} \omega_{ti} \boldsymbol{X}_{ti}^{\top} \boldsymbol{\beta}_a \cdot \mathbf{1}\{\pi^*(\boldsymbol{X}_{ti}) = a\} \right] - V^{\pi^*} = 0,$$

it holds that $\mathbb{E}[\xi_{2q}|\mathcal{H}_{q-1}] = 0$. Additionally, notice that

$$\mathbb{E}[\xi_{1q}|\mathcal{H}_{q-1}] = \mathbb{E}\left[\frac{\mathbf{1}\{a_q = \pi^*(\boldsymbol{X}_q)\}}{\mathbb{P}(a_q = \pi^*(\boldsymbol{X}_q))} \cdot \eta_q \Big| \mathcal{H}_{q-1}\right] = \mathbb{E}\left[\frac{\mathbf{1}\{a_q = \pi^*(\boldsymbol{X}_q)\}}{\mathbb{P}(a_q = \pi^*(\boldsymbol{X}_q))} \cdot \mathbb{E}[\eta_q|\mathcal{H}_{q-1}, \boldsymbol{A}_t]\Big| \mathcal{H}_{q-1}\right] = 0.$$

Thus, $\mathbb{E}[\xi_q|\mathcal{H}_{q-1}] = 0$, and $\{\xi_q\}_{1 \le q \le \bar{N}_T}$ is a Martingale difference sequence. To show that $\sqrt{\bar{N}_T}(\bar{V}_T - V^{\pi^*}) \xrightarrow{\mathcal{D}} \mathcal{N}(0, \sigma_V^2)$ and derive the asymptotic variance $\sigma_V^2$, it suffice to check the Lindeberg condition and use Martingale CLT to establish asymptotic normality.

**(1) First, let's check the Lindeberg condition.**

$$\sum_{q=1}^{\bar{N}_T} \mathbb{E}\left[\frac{1}{\bar{N}_T}\xi_q^2 \cdot \mathbf{1}\left\{\left|\frac{1}{\sqrt{\bar{N}_T}}\xi_q\right| \ge \delta\right\}\Big|\mathcal{H}_{q-1}\right] = \frac{1}{\bar{N}_T}\sum_{q=1}^{\bar{N}_T}\mathbb{E}\left[\xi_q^2\mathbf{1}\{\xi_q^2 \ge \bar{N}_T\delta^2\}\big|\mathcal{H}_{q-1}\right].$$

Notice that $\xi_q^2\mathbf{1}\{\xi_q^2 \ge \bar{N}_T\delta^2\}$ converges to 0 as $\bar{N}_T$ goes to infinity and is bounded by $\xi_q^2$ given $\mathcal{H}_{q-1}$. Therefore, we only need to check the integrability of $\xi_q^2$ given $\mathcal{H}_{q-1}$, then by Dominated Convergence Theorem (DCT), the Lindeberg condition is checked.

Since the derivation of $\mathbb{E}[\xi_q^2|\mathcal{H}_{q-1}]$ is exactly the asymptotic variance $\sigma_V^2$, we will leave the details to part (2).

**(2) Next, we derive the limit of conditional variance $\sigma_V^2 = \frac{1}{\bar{N}_T}\sum_{q=1}^{\bar{N}_T}\mathbb{E}[\xi_q^2|\mathcal{H}_{q-1}]$.**

$$\mathbb{E}[\xi_{1q}^2|\mathcal{H}_{q-1}] = \mathbb{E}\left[\frac{\mathbf{1}\{a_q = \pi^*(\boldsymbol{X}_q)\}}{[\mathbb{P}\{a_q = \pi^*(\boldsymbol{X}_q)\}]^2} \cdot \eta_q^2\Big|\mathcal{H}_{q-1}\right] = \mathbb{E}\left[\frac{\mathbf{1}\{a_q = \pi^*(\boldsymbol{X}_q)\}}{[\mathbb{P}\{a_q = \pi^*(\boldsymbol{X}_q)\}]^2} \cdot \mathbb{E}[\eta_q^2|\mathcal{H}_{q-1}, \boldsymbol{A}_t]\Big|\mathcal{H}_{q-1}\right]$$

$$= \mathbb{E}\left[\frac{\mathbf{1}\{a_q = \pi^*(\boldsymbol{X}_q)\}}{[\mathbb{P}\{a_q = \pi^*(\boldsymbol{X}_q)\}]^2} \cdot \sigma^2\Big|\mathcal{H}_{q-1}\right].$$

Therefore,

$$\frac{1}{\bar{N}_T}\sum_{q=1}^{\bar{N}_T}\mathbb{E}[\xi_{1q}^2|\mathcal{H}_{q-1}] = \sigma^2 \cdot \mathbb{E}\left[\frac{1}{\bar{N}_T}\sum_{q=1}^{\bar{N}_T}\frac{1 - \nu_q(\boldsymbol{X}_q, \mathcal{H}_{t-1})}{[\mathbb{P}\{a_q = \pi^*(\boldsymbol{X}_q)\}]^2}\right],$$

where $\nu_q(\boldsymbol{X}_q, \mathcal{H}_{t-1}) = \nu_{ti}(\boldsymbol{X}_{ti}, \mathcal{H}_{t-1}) = \mathbb{P}(A_{ti} \ne \pi^*(\boldsymbol{X}_{ti})|\boldsymbol{X}_{ti}, \mathcal{H}_{t-1})$.

Following similar proof structure of Ye et al. (2023) in Appendix page 34-35, we are able to establish that

$$\frac{1}{\bar{N}_T}\sum_{q=1}^{\bar{N}_T}\mathbb{E}[\xi_{1q}^2|\mathcal{H}_{q-1}] = \sigma^2 \cdot \mathbb{E}\left[\frac{1}{\bar{N}_T}\sum_{q=1}^{\bar{N}_T}\frac{1 - \nu_q(\boldsymbol{X}_q, \mathcal{H}_{t-1})}{[\mathbb{P}\{a_q = \pi^*(\boldsymbol{X}_q)\}]^2}\right] \to \int \frac{\sigma^2}{1 - \kappa_\infty(\boldsymbol{x})}d\mathcal{P}_{\boldsymbol{x}},$$

where $\kappa_\infty(\boldsymbol{x}) = \lim_{q\to\infty}\mathbb{P}(A_{ti} \ne \pi^*(\boldsymbol{x}))$.

Since $\mathbb{E}[\xi_{2q}] = 0$ and the randomness in $\xi_{2q}$ only comes from $(\boldsymbol{X}, \omega)$, we have

$$\mathbb{E}[\xi_{2q}^2|\mathcal{H}_{q-1}] = \mathrm{Var}(\xi_{2q}) = \mathrm{Var}\left[\sum_{a\in\mathcal{A}}\omega_{ti}\boldsymbol{X}_{ti}^\top\boldsymbol{\beta}_a \cdot \mathbf{1}\{\pi^*(\boldsymbol{X}_{ti}) = a\}\right].$$

Furthermore,

$$\mathbb{E}[\xi_{1q}\xi_{2q}|\mathcal{H}_{q-1}] = \mathbb{E}\left[\xi_{2q} \cdot \frac{\mathbf{1}\{a_q = \pi^*(\boldsymbol{X}_q)\}}{\mathbb{P}(a_q = \pi^*(\boldsymbol{X}_q))} \cdot \eta_q\Big|\mathcal{H}_{q-1}\right] = \mathbb{E}\left[\xi_{2q} \cdot \frac{\mathbf{1}\{a_q = \pi^*(\boldsymbol{X}_q)\}}{\mathbb{P}(a_q = \pi^*(\boldsymbol{X}_q))} \cdot \mathbb{E}[\eta_q|\mathcal{H}_{q-1}, \boldsymbol{A}_t]\Big|\mathcal{H}_{q-1}\right] = 0.$$

Thus,

$$\frac{1}{\bar{N}_T}\sum_{q=1}^{\bar{N}_T}\mathbb{E}[\xi_q^2|\mathcal{H}_{q-1}] = \frac{1}{\bar{N}_T}\sum_{q=1}^{\bar{N}_T}\mathbb{E}[(\xi_{1q} + \xi_{2q})^2|\mathcal{H}_{q-1}]$$

$$= \frac{1}{\bar{N}_T}\sum_{q=1}^{\bar{N}_T}\mathbb{E}[\xi_{1q}^2|\mathcal{H}_{q-1}] + \frac{1}{\bar{N}_T}\sum_{q=1}^{\bar{N}_T}\mathbb{E}[\xi_{2q}^2|\mathcal{H}_{q-1}] + \frac{2}{\bar{N}_T}\sum_{q=1}^{\bar{N}_T}\mathbb{E}[\xi_{1q}\xi_{2q}|\mathcal{H}_{q-1}] \tag{77}$$

$$\to \int \frac{\sigma^2}{1 - \kappa_\infty(\boldsymbol{x})}d\mathcal{P}_{\boldsymbol{x}} + \mathrm{Var}\left[\sum_{a\in\mathcal{A}}\omega_{ti}\boldsymbol{x}^\top\boldsymbol{\beta}_a \cdot \mathbf{1}\{\pi^*(\boldsymbol{x}) = a\}\right].$$

Therefore,

$$\sigma_V^2 = \int \frac{\sigma^2}{1 - \kappa_\infty(\boldsymbol{x})} d\mathcal{P}_{\boldsymbol{x}} + \mathrm{Var}\Big[\sum_{a \in \mathcal{A}} \omega \boldsymbol{x}^\top \boldsymbol{\beta}_a \cdot \mathbf{1}\{\pi^*(\boldsymbol{x}) = a\}\Big]. \tag{78}$$

**Finally**, by combining the results of Step 1-3, we are able to show that $\sqrt{\bar{N}_T}(\widehat{V}_T^{\mathrm{DR}} - V^{\pi^*}) \xrightarrow{\mathcal{D}} \mathcal{N}(0, \sigma_V^2)$, which concludes the proof of this theorem.

## J. Proof of Regret Bound

**Step 1:** Decompose $R_T = R_T^{(1)} + R_T^{(2)}$, which accounts for the regret of exploitation and exploration.

Recall that the regret $R_T$ is defined as

$$R_T = \sum_{t=1}^{T} \sum_{i=1}^{N_t} \mathbb{E}\big[\mu^{(t,i)}(\boldsymbol{X}_t, \pi^*(\boldsymbol{X}_t)) - \mu^{(t,i)}(\boldsymbol{X}_t, \boldsymbol{A}_t)\big]$$

$$= \sum_{t=1}^{T} \sum_{i=1}^{N_t} \mathbb{E}\big[\mu^{(t,i)}(\boldsymbol{X}_t, \pi^*(\boldsymbol{X}_t)) - \mu^{(t,i)}(\boldsymbol{X}_t, \widehat{\pi}_{t-1}(\boldsymbol{X}_t))\big] + \sum_{t=1}^{T} \sum_{i=1}^{N_t} \mathbb{E}\big[\mu^{(t,i)}(\boldsymbol{X}_t, \widehat{\pi}_{t-1}(\boldsymbol{X}_t)) - \mu^{(t,i)}(\boldsymbol{X}_t, \boldsymbol{A}_t)\big],$$

which can be decomposed into

$$R_T = \underbrace{\sum_{t=1}^{T} \sum_{i=1}^{N_t} \mathbb{E}\Big[\mu^{(t,i)}(\boldsymbol{X}_t, \pi^*(\boldsymbol{X}_t)) - \mu^{(t,i)}(\boldsymbol{X}_t, \widehat{\pi}_{t-1}(\boldsymbol{X}_t))\Big]}_{R_T^{(1)}}$$

$$+ \underbrace{\sum_{t=1}^{T} \sum_{i=1}^{N_t} \mathbb{E}\Big[\big\{\mu^{(t,i)}(\boldsymbol{X}_t, \widehat{\pi}_{t-1}(\boldsymbol{X}_t)) - \mu^{(t,i)}(\boldsymbol{X}_t, \boldsymbol{A}_t)\big\} \cdot \mathbf{1}\{A_{ti} \neq \widehat{\pi}_{t-1}(\boldsymbol{X}_{ti})\}\Big]}_{R_T^{(2)}}.$$

By definition, $R_T^{(1)}$ is nonzero only when $\pi^*(\boldsymbol{X}_{ti}) \neq \widehat{\pi}_{t-1}(\boldsymbol{X}_{ti})$, which accounts for the regret caused by estimation accuracy, i.e., exploitation. $R_T^{(2)}$ is nonzero only $A_{ti} \neq \widehat{\pi}_{t-1}(\boldsymbol{X}_{ti})$, which accounts for the regret caused by exploration. In Step 2-3, we will derive the regret bound of $R_T^{(1)}$ and $R_T^{(2)}$ separately to prove the sublinearity of $R_T$.

**Step 2:** Prove that $R_T^{(1)} = o(\bar{N}_T^{1/2})$.

Notice that in the proof of Theorem 4.4, step 2.1, we've proved in Equation (64) that

$$\frac{1}{\sqrt{\bar{N}_T}} \sum_{t=1}^{T} \sum_{i=1}^{N_t} \Big[\mu^{(t,i)}(\boldsymbol{X}_t, \pi^*(\boldsymbol{X}_t)) - \mu^{(t,i)}(\boldsymbol{X}_t, \widehat{\pi}_{t-1}(\boldsymbol{X}_t))\Big] = o_p(1).$$

Therefore,

$$R_T^{(1)} = \sum_{t=1}^{T} \sum_{i=1}^{N_t} \mathbb{E}\Big[\mu^{(t,i)}(\boldsymbol{X}_t, \pi^*(\boldsymbol{X}_t)) - \mu^{(t,i)}(\boldsymbol{X}_t, \widehat{\pi}_{t-1}(\boldsymbol{X}_t))\Big] = o(\bar{N}_T^{1/2}).$$

**Step 3:** Prove that $R_T^{(2)} = O(\bar{N}_T^{1/2} \log \bar{N}_T)$.

According to the upper bound derived in Theorem 4.2,

$$R_T^{(2)} \leq \sum_{t=1}^{T} \sum_{i=1}^{N_t} \mathbb{E}\Big[\big|\mu^{(t,i)}(\boldsymbol{X}_t, \widehat{\pi}_{t-1}(\boldsymbol{X}_t)) - \mu^{(t,i)}(\boldsymbol{X}_t, \boldsymbol{A}_t)\big| \cdot \mathbf{1}\{A_{ti} \neq \widehat{\pi}_{t-1}(\boldsymbol{X}_{ti})\}\Big]$$

$$\leq 2U \cdot \sum_{t=1}^{T} \sum_{i=1}^{N_t} \mathbb{E}\big[\mathbf{1}\{A_{ti} \neq \widehat{\pi}_{t-1}(\boldsymbol{X}_{ti})\}\big] = 2U \cdot \sum_{t=1}^{T} \sum_{i=1}^{N_t} \kappa_{ti}(\omega_{ti}, \boldsymbol{X}_{ti}).$$

Now let's decompose according to different exploration algorithms. For simplicity of notations, we continue with the flattened unit queue $q = Q(t, i) \sum_{s=1}^{t-1} N_s + t$ as shown in the proof of Theorem 4.1. As such, we can extend the definition of $p_t$ to $p_q$ by simply setting $p_q = p_t$ for any unit $q$ in round $t$. As such, $p_q$ is still a non-increasing sequence w.r.t. $q$. By Theorem 4.2, we have

1. In UCB, $\kappa_{ti}(\omega_{ti}, \boldsymbol{X}_{ti})$ is upper bounded by $O(K^2 L_w^\gamma \cdot (\bar{N}_{q-1} p_{q-1})^{-\gamma/2})$. Let $p_q = \bar{N}_q^{u/\gamma - 1}$ with $u > 0$. For $\gamma > u$, $p_q$ is decreasing. Then

$$R_T^{(2)} \lesssim \sum_{t=1}^{T} \sum_{i=1}^{N_t} \kappa_{ti}(\omega_{ti}, \boldsymbol{X}_{ti}) \lesssim L_w^\gamma K^2 \cdot \sum_{q=1}^{\bar{N}_T} q^{-u/2} = O(L_w^\gamma K^2 \cdot \bar{N}_T^{1-u/2}).$$

Taking $u = 1$ gives us $R_T^{(2)} = O(L_w^\gamma K^2 \cdot \bar{N}_T^{1/2} = O(\bar{N}_T^{1/2})$. note that when the interference constraint $L_w$ or the number of arms $K$ is large, the regret bound $R_T^{(2)}$ tends to be larger.

2. In TS, $\kappa_{ti}(\omega_{ti}, \boldsymbol{X}_{ti})$ is upper bounded by $O(dK^3 \exp\{-\bar{N}_{q-1} p_{q-1}^2 / L_w^4\})$. Let $p_q = \sqrt{\alpha \log q / \bar{N}_q}$. Then $\kappa_{ti}(\omega_{ti}, \boldsymbol{X}_{ti}) \leq O(dK^3 \exp\{-\alpha \log(q-1)/L_w^4\}) = O(dK^3 q^{-L_w^{-4}\alpha})$. Thus,

$$R_T^{(2)} \lesssim \sum_{t=1}^{T} \sum_{i=1}^{N_t} \kappa_{ti}(\omega_{ti}, \boldsymbol{X}_{ti}) \lesssim dK^3 \sum_{q=1}^{\bar{N}_T} q^{-L_w^{-4}\alpha} = O(dK^3 \bar{N}_T^{1-L_w^{-4}\alpha}).$$

When the interference constraint $L_w$ is large, the regret bound $R_T^{(2)}$ tends to be larger. By taking $\alpha = L_w^4/2$, we have $R_T^{(2)} = O(dK^3 \bar{N}_T^{1/2}) = O(\bar{N}_T^{1/2})$.

3. In EG, $\kappa_{ti}(\omega_{ti}, \boldsymbol{X}_{ti}) = \epsilon_q/2$. If we set $\epsilon_q = O(q^{-m})$ with any $m < 1/2$, then

$$R_T^{(2)} \lesssim \sum_{t=1}^{T} \sum_{i=1}^{N_t} \kappa_{ti}(\omega_{ti}, \boldsymbol{X}_{ti}) \lesssim \sum_{q=1}^{\bar{N}_T} q^{-m} = O(\bar{N}_T^{1-m}).$$

By setting $\epsilon_q = O(\log q / \sqrt{q})$, we have $R_T^{(2)} = O(\bar{N}_T^{1/2} \log \bar{N}_T)$.

Thus, in UCB, TS, and EG, the regret caused by exploration can be controlled by $R_T^{(2)} = O(\bar{N}_T^{1/2} \log \bar{N}_T)$.

Therefore, by combining the results of Step 1-3, we are able to show that

$$R_T = \sum_{t=1}^{T} \sum_{i=1}^{N_t} \mathbb{E}[R_{ti}^* - R_{ti}] = O(\bar{N}_T^{1/2} \log \bar{N}_T),$$

which is sublinear in $\bar{N}_T$.

