# OpenReview forum: "Linear Contextual Bandits With Interference"
_ICML.cc/2025/Conference — ICML 2025 poster_

### Official Review · Reviewer_SPrL · 2025-02-27

**Overall Recommendation:** 3

**Summary:**

This paper is the first to study contextual bandits with interference. The authors propose a linear contextual bandit framework and introduce an algorithm called LinCB to address the regret minimization/estimation problem under this framework. Experimental results based on MovienLens dataset further demonstrates the effectiveness of the proposed results.

**Claims And Evidence:**

Yes

**Essential References Not Discussed:**

Some references might need to be discussed. For example, in Shiliang Zuo's Federated Multi-Armed Bandits, they consider a setting where the number of agents varies over time, which is similar to the setting considered by the authors.

**Experimental Designs Or Analyses:**

Yes

**Methods And Evaluation Criteria:**

Yes

**Other Comments Or Suggestions:**

I find this to be an interesting paper and am inclined to accept it. I look forward to the authors' responses to my questions.

## update after rebuttal: The authors have addressed my concerns, so I recommend acceptance.

**Other Strengths And Weaknesses:**

Strengths:
1. This paper proposed the first study on contextual bandit with interference.
2. The proofs I have checked do not have any obvious flaws.
3.  The authors consider both the regret minimization and estimation problems.
4. The experimental results are based on a real world dataset.

Weaknesses:
1. I believe there are issues with the discussion of references in this paper. In line 86, the cited work Dubey et al., 2020 does not belong to the MAMAB problem; rather, it falls under a multi-agent (kernelized) contextual bandits problem. If, as the authors claim, Dubey 2020 studies interference-related problems, then that paper might be the first to investigate contextual bandits with interference. This contradicts the authors' assertion that their work is the first interference-related paper, which leaves me somewhat confused. How does the learning framework in that paper differ from the one in this work?
2. The authors have not clearly discussed the distinction between multi-agent bandits and bandits with interference. IMHO, most federated/distributed/MA bandit studies focus on leveraging the communication topology to design communication algorithms that accelerate the learning process. In contrast, interference bandits (e.g., [Jia et al., 2024; Agarwal et al., 2024]) focus more on learning the strength of interference between different nodes. The authors could add a discussion on this aspect.
3. The authors assume that the interference strength matrix $W_t$ is known. Based on my understanding of the previous interference paper, such as Jia et al 2024., Agarwal et al 2024., and Leung et al 2024., they do not assume that the interference strength between agents is known; instead, their task is more focused on indirectly learning this interference strength. Could you discuss references that consider a similar setting? I also look forward to the authors adding more discussion on the scenario where $W_t$ is unknown.
4. Theorem 4.2 refers to Assumptions A.1 - A.3, which the authors have placed in the appendix. I suggest that the authors include these assumptions (and the related discussion) in the main text.
5. A typical regret upper bound often includes the time horizon $T$, whereas this paper only provides an upper bound in terms of $\bar{N}_t$. The authors could add a remark discussing the relationship between the given upper bound and a more conventional upper bound that explicitly includes $T$.

**Questions For Authors:**

See the S/W section

**Relation To Broader Scientific Literature:**

N/A

**Theoretical Claims:**

Yes

---

> ### Author Rebuttal · Authors · 2025-04-01
>
> Thanks for your thoughtful questions and the time you spent reviewing our paper. We really appreciate your insights and are happy to discuss any further ideas or questions you may have.
>
> **Answer to W1**:
>
> Thank you for your insightful perspective on the literature of Dubey et al. (2020). We would like to clarify a minor error in the related work section: Dubey et al. (2020) did not model interference. By definition, interference occurs when an agent’s action influences the rewards of others. However, in their framework, the reward of agent $v$ is defined as $f_v = F(a_v, z_v)$, which depends solely on the agent’s own action $a_v$ and contextual information $z_v$, with no influence from other agents’ actions. We appreciate your attention to this detail and will remove this paper from the interference-related literature in the main paper. Consequently, we reaffirm that our work is the first to address interference in contextual bandits.
>
> **Answer to W2**:
>
> Multi-agent bandits and bandits with interference are related but focus on different aspects of the problem, with some overlap but distinct emphases:
> * **Multi-agent bandits** primarily deal with the interaction between multiple agents (typically fixed in number) and how they share information to enhance bandit learning. The central focus is on fostering collaboration or competition among agents.
> * **Bandits with interference** stem from the concept of "interference" in causal inference. The key consideration is whether one agent's action impacts the reward of others.
>
> Not all multi-agent bandit works address interference. In Sec 2, under *Cooperative Multi-Agent Bandits*, we classify prior literature:
> Paragraph 1 lists works that do not address interference, and Paragraph 2 includes studies that do, even if not explicitly framed as interference problems. For example, Bargiacchi et al. 2018 and Verstraeten et al. 2020 implicitly handle interference, similar to Jia et al. (2024) and Agarwal et al. (2024).
>
> In summary, the defining criterion for bandits with interference is straightforward: *Does one agent's action affect another’s reward?* If yes, interference exists and must be accounted for.
>
> **Answer to W3**:
>
> * First, we would like to clarify that our work, along with (Jia et al., 2024; Agarwal et al., 2024; Leung, 2022), all assume different known structures on interference, albeit in different ways. Without any assumptions on the nature of the interference pattern, both estimation and bandit learning would be infeasible due to the high-dimensional action space, $K^{N}$. Specifically, (Jia et al., 2024) and (Leung, 2022) assume that the strength of interference decays according to a specific notion ($\psi$) of distance between units, while (Agarwal et al., 2024) imposes a sparsity assumption on network interference, restricting interference to only a small neighborhood of size $s$. In contrast, we introduce an assumption in a more intuitive manner by modeling pairwise interference through a matrix $W$, which simplifies both modeling and interpretation. While this assumption may appear strong, it provides flexibility, as the entries of $W$ can take any value within $[-1,1]$. From this perspective, our formulation is more general.
> * Additionally, we would like to emphasize that (Jia et al., 2024) and (Leung, 2022) do not ”indirectly learn” the interference
> strength. For instance, in (Jia et al., 2024), all regret bounds are derived under the assumption of a known $\psi$, which prescribes
> a specific functional form for interference decay.
> * In Sec 2, the second paragraph highlights works that quantify interference in a similar setting to ours, such as
> (Getis, 2009; Valcu & Kempenaers, 2010; Su et al., 2019). This structure is widely used in network and interference-related literature.
> * Lastly, due to space constraints, regarding the case where $W_t$ is unknown: aside from the second paragraph of Sec 7, we kindly refer the reviewer to our detailed discussion in response to **Reviewer f3ef, Q1**. We hope that this plausible extension, along with the tolerance of LinCBWI w.r.t. the misspecification of $W_t$ in the added simulation, will help alleviate your concern.
>
> **Answer to W4**:
>
> Thanks for your suggestion regarding assumptions placement. We will move them to the main paper and we believe this would improve the readability.
>
> **Answer to W5**:
>
> The regret bound actually depends on $T$ only implicitly through $\bar{N}_T$. In the special case where each round involves a fixed number of units, $N_t\equiv n$, then $\bar{N}_T=nT$, leading to a regret bound of approximately $O(n^{1/2}T^{1/2})$, up to logarithmic terms.
>
> **Others**:
>
> We noticed that the reviewer referenced Shiliang Zuo's *Federated Multi-Armed Bandits*, but we were only able to find a paper with the same title by a different author. Could you kindly provide more details or clarify the reference? We would be happy to include and carefully discuss this work in the final version of our paper.

---

> > ### Comment · Reviewer_SPrL · 2025-04-05
> >
> > Thank you for your answer — it resolved my concern. Yes, I made a mistake earlier; the first author of the paper titled Federated Multi-Armed Bandits is Chengshuai Shi.
> >
> > Additionally, I look forward to seeing a more detailed discussion of the case where the interference strength is unknown in future versions of the paper — this is certainly an exciting direction for future work. I will keep my current score and am inclined to recommend acceptance.

---

> > > ### Author Response · Authors · 2025-04-08
> > >
> > > Dear Reviewer SPrL,
> > >
> > > Thank you for your clarification regarding the related work *Federated Multi-Armed Bandits* by Chengshuai Shi. We agree that this is a highly relevant and interesting paper, and we will incorporate it into the related work section of our final version. This work is among the first to connect federated learning with multi-armed bandits in dynamic agent settings. However, it focuses on standard MAB rather than contextual bandits, which is the setting considered in our work. Moreover, the issue of interference is not addressed in their framework, as each action $k$ taken by a client (or agent) $m$ only influences its own local reward $\mu_{k,m}$. The shared information among clients is used to collectively improve estimates of the global arms, rather than to model inter-agent dependencies.
> > >
> > > Regarding the case where the interference structure is unknown, we will include the discussion provided in the rebuttal in the final version of our paper. We would also like to note that this direction raises several important and technically rich questions, such as identifiability, convergence guarantees, and statistical inference under uncertainty, which we believe warrant a dedicated follow-up study. Within the scope of the current paper, our primary aim has been to establish a principled framework that bridges interference and contextual bandits, supported by rigorous theoretical analysis and extensive simulations. We believe our current contributions represent a well-substantiated and meaningful step toward addressing this important gap in the existing literature.
> > >
> > > Once again, we sincerely appreciate the time and effort you devoted to reviewing our work, as well as your thoughtful feedback and constructive suggestions throughout the process.
> > >
> > > Best,
> > >
> > > Authors from submission 4971

---

### Official Review · Reviewer_2fJW · 2025-03-12

**Overall Recommendation:** 3

**Summary:**

This paper investigates the problem of interference in linear contextual bandits, where the actions taken for one unit influence the rewards of others. This paper leverages an adjacency matrix to model the interference structure and proposes three online algorithms LinEGWI, LinUCBWI, and LinTSWI. The authors establish several theoretical guarantees, including regret bounds, asymptotic properties of the OLS estimator, and statistical inference for the optimal policy value. The proposed methods demonstrate superior empirical performance over classical linear contextual bandit approaches in both synthetic experiments and a real-world MovieLens-based dataset.

## update after rebuttal
Overall, I appreciate the technical novelty of this paper, so I will maintain my current positive score.

**Claims And Evidence:**

All claims are well-supported by clear and convincing evidence.

**Essential References Not Discussed:**

N/A.

**Experimental Designs Or Analyses:**

The experimental designs or analyses are sound and valid.

**Methods And Evaluation Criteria:**

The proposed methods and evaluation criteria make sense for the problem.

**Other Comments Or Suggestions:**

N/A.

**Other Strengths And Weaknesses:**

Strengths
1. To my knowledge, the use of an adjacency matrix to quantify interference in linear contextual bandits is novel.
2. This paper derives several theoretical guarantees, such as sublinear regret bounds that match the minimax optimal rate and asymptotic normality of the OLS estimator.
3. The paper includes extensive simulations demonstrating the advantages of the proposed algorithms. The paper also shows that without interference, the proposed algorithms reduce to classical algorithms with comparable performance.

Weaknesses
1. This paper assumes that the interference matrix is known, which may be impractical in real-world settings.
2. The algorithm design seems to be heavily influenced by Shen et al. 2024 (Doubly Robust Interval Estimation for Optimal Policy Evaluation in Online Learning).
3. The theoretical results rely on several assumptions that are non-standard in the bandit literature, especially the clipping assumption (Assumption A.2), which appears to be quite strong.

**Questions For Authors:**

1. How does the computational overhead of the proposed algorithms compare to that of classical linear bandit algorithms?

**Relation To Broader Scientific Literature:**

This work advances the field by proposing a framework for linear contextual bandits under interference. In contrast, prior studies only consider interference in multi-armed bandits or adversarial contextual bandit settings. The proposed algorithms (Algorithm 1) appear to be adapted from Shen et al., 2024.

Shen et al. Doubly Robust Interval Estimation for Optimal Policy Evaluation in Online Learning. 2024.

**Theoretical Claims:**

I have not checked all the proofs in detail. I did not identify any obvious errors.

---

> ### Author Rebuttal · Authors · 2025-03-31
>
> Thanks for your thoughtful questions and the time you spent reviewing our paper. We really appreciate your insights and are happy to discuss any further ideas or questions you may have.
>
> **Answer to W1**:
>
> Regarding the assumption that the interference matrix is known, we clarify this in three aspects:
>
> **First**, in many real-world applications, interference is either known or can be precomputed before applying bandit learning. For instance, in the context of COVID-19, geographic proximity naturally defines community connections, while in movie recommendation systems, social networks provide side information that quantifies pairwise interference. Such structural information is often available through expert knowledge or can be inferred from covariates before deploying a bandit algorithm.
>
> **Second**, the assumption of a known interference structure is widely used in both classical interference literature and its bandit extensions. In single stage, several works rely on this assumption  (Manski, 2013; Aronow & Samii, 2017;
> Su et al., 2019; Bargagli-Stoffi et al., 2020), and it has been adopted in bandit settings as well (Jia et al., 2024). While it is ideal to learn the interference structure purely from data, there is always a trade-off between what is pre-specified as an assumption and what is left for the model to infer. In other words, there is no "free lunch"--the choice of assumption depends on the specific problem setting and modeling priorities.
>
> **Lastly**, we give a practical direction to the case where the interference matrix is unknown in the second paragraph of Sec 7. Due to space constraints, we kindly refer the reviewer to our detailed discussion in response to **Reviewer f3ef, Q1**.
>
> **Answer to W2**:
>
> We clarify that our work differs substantially from Shen 2024 in several key aspects:
>
> 1.**Problem Motivation**: Our work addresses the fundamental challenge of **interference** in bandits, whereas Shen 2024 focuses on statistical inference for the optimal value in a standard bandit setting, which is an entirely different problem.
>
> 2. **Scope of Application**: Shen 2024 considers inference in a two-arm bandit setting, limiting its applicability in real-world scenarios with multiple arms. In contrast, our work accommodates general **multi-arm** settings, making it significantly broader in scope.
> 3. **Our Contributions**: Our work is the first to explore interference in contextual bandits.
> * From a statistical inference perspective, extending Shen 2024 to incorporate interference and multi-arm settings is **nontrivial**, particularly in establishing theoretical results due to the data dependency introduced by interference. Our work provides statistical guarantees in this broader context, but its contributions extend well beyond merely building on Shen 2024. Although we generalize their results, our study is fundamentally distinct, with a broader focus that does not rely heavily on their approach.
>
> * Moreover, if statistical properties (as discussed in Sec 4.3-4.4) are not relevant to a particular application, the clipping step (Line 10 of Alg. 1) can simply be omitted without affecting our main contributions, making our work entirely independent of Shen 2024. **The core novelty of our work lies in the problem formulation, the design of bandit algorithms, and regret analysis in this novel setting, all supported by extensive simulations and quasi-real data analysis. While we establish statistical inference results, they primarily reinforce our findings rather than serving as the sole core contribution.**
>
> **Answer to W3**:
>
> Regarding Assumption **A.2**, which may be of particular concern, we emphasize that:
> * **The assumption is not restrictive** due to the small multiplication factor applied on the right-hand side, $p_t$. In Alg. 1, we specify that the clipping rate $p_t$ only needs to satisfy the condition of not decaying faster than $O(\bar{N}_t^{-1/2})$. This means that any decreasing sequence, such as $p_t = O(\bar{N}_t^{-3/7})$ or $O(\bar{N}_t^{-2/5})$, or even a small fixed value (e.g., $p_t = 10^{-3}$) suffices. Given that $p_t$ is small and decreases over time, Assumption A.2 naturally holds as the sample size grows.
>
> * **A.2 is actively enforced in our algorithm**. Specifically, Line 10 of Algorithm 1 ensures that if an arm is explored insufficiently (determined based on $p_t$ and a smallest eigenvalue comparison), the algorithm enforces additional exploration, preventing extreme imbalance and ensuring statistical consistency.
>
> * **Empirical validation**: In our simulation studies, we simply set $p_t \equiv 0.01$ and observed that Line 10 is rarely triggered. This indicates that the assumption does not pose practical concerns.
>
> **Answer to "Questions For Authors"**:
>
> Due to space constraints, we refer the reviewer to our response to **Reviewer f3ef, Q2**, which includes a detailed analysis and a comparison plot at https://anonymous.4open.science/r/LinCBWI_ICML_rebuttal-C829/.

---

### Official Review · Reviewer_JVt5 · 2025-03-13

**Overall Recommendation:** 3

**Summary:**

The paper explores the intersection of causal inference and multi-armed bandits, specifically in the setting of multi-agent bandits with interference among agents. According to the authors, this is the first work in the literature to incorporate contextual information (i.e., the covariates of units). Under certain assumptions, particularly linearity, the authors propose an algorithm with three exploration variants to address the problem. Both theoretical analysis and experimental results demonstrate the effectiveness of the proposed approach.

##  Update After Rebuttal
After reviewing the authors’ response, I have decided to maintain my current score.

**Claims And Evidence:**

Yes.

**Essential References Not Discussed:**

I am not aware of any missing prior works.

**Experimental Designs Or Analyses:**

The design of conducted experiments are seems sound.

**Methods And Evaluation Criteria:**

The work is mostly theoretical but also contain some experiments which have reasonable design.

**Other Comments Or Suggestions:**

It would be better to use $\top$ for the transpose operation of a matrix (e.g., $\beta^{\top}$
instead of  $\beta’$).


The notation and formulas in Subsection 3.1 are not very clear. I believe this section could be rewritten for better clarity.

**Other Strengths And Weaknesses:**

Strengths:
1. The paper addresses an important and novel problem at the intersection of causal inference and bandits.
2. Rigorous theoretical analysis is provided, ensuring performance guarantees for the proposed algorithm.
3. The problem is well-motivated with real-world examples, and the literature review is thorough. Additionally, the paper discusses potential directions for future work.

Weaknesses:
1. Some parts of the paper are difficult to follow due to notation and presentation of results, and these could be improved (see the suggestions section).

**Questions For Authors:**

1. I did not fully understand why you modeled the inference using Equation 1. To what extent does this formulation restrict the problem? Could you discuss alternative formulations for the reward and the challenges they pose for algorithm design?
2. In Lines 2 and 11 of Algorithm 1, why do you sample actions from Bernoulli(0.5)? Shouldn't it be sampled uniformly over $[k]$?
3. In Theorem 4.2, how did you derive the third part?
4. In subsection 3.1, is it possible to express $f(X_{tj}, a_{tj})$ in a different form? For example, by a simple inner product if each action represents a vector.

**Relation To Broader Scientific Literature:**

The key contributions of the paper are in the areas of online decision-making (bandits) and causal inference.

**Theoretical Claims:**

I checked the flow of some parts of proofs but I’m not certain all are correct.

---

> ### Author Rebuttal · Authors · 2025-03-31
>
> Thanks for your thoughtful questions and the time you spent reviewing our paper. We really appreciate your insights and are
> happy to discuss any further ideas or questions you may have.
>
> **Q1**:
>
> Eq. (1) models the reward of unit $i$ in round $t$ as $R_{ti}=\sum_{j=1}^{N_t} W_{t,ij} f_{tj}+\eta_{ti}$, where the reward is a weighted sum of sub-payoff functions $ f_{tj}=f(X_{tj},A_{tj})$, with weights $W_{t,ij}$ determined by the interference matrix $W_t$. Specifically, the $i$th row of $W_t$ captures the influence of all other units on unit $i$, where each element $W_{t,ij}$ represents the strength of unit $j$’s contribution to $i$’s reward. The term $f_{tj}$ reflects unit $j$’s individual contribution, and multiplying by $W_{t,ij}$ quantifies its effect on $i$. Summing over all units $j\in\\{1,\dots,N_t\\}$ yields the final reward $R_{ti}$. This formulation is intuitive, easily decomposable across individuals, and widely used in network and interference-related literature (Cliff & Ord, 1981; Getis, 2009; Su et al., 2019).
>
> There are several alternatives to Eq (1). One approach is to generalize it as $R_{ti} = g(\boldsymbol{W_t}, \boldsymbol{X_t},\boldsymbol{A_t}; \theta) + \eta_{ti}$, where $g$ is parameterized by $\theta$, potentially using more flexible models such as neural networks. While this model is learnable given sufficient data, it would invalidate Eq. (2), which, through the decomposition in Eq. (1), effectively consolidates all interference-related information of unit $i$ into the interference weight $\omega_{ti}$ and simplifies decision making.
>
> Other directions, instead of using a matrix $W_t$ to quantify pairwise interference levels, include assuming alternative interference-related structures such as partial interference or exposure mapping (see the first paragraph of Section 2). However, the fundamental challenge of the interference problem is that, without any structural assumption, $R_{ti}$ could be arbitrarily influenced by all units' contextual information and actions $(\boldsymbol{X}_t, \boldsymbol{A}_t)$, making the problem too general to be learnable. Some assumptions (whether in the form of Eq. (1) or other models) are necessary to impose structure on the interference, depending on what best aligns with real-world data. Given the structured formulation of $\boldsymbol{W}_t$ and its interpretability within a linear framework, we find Eq. (1) to be a natural and intuitive starting point.
>
> **Q2**
>
> Thanks for pointing that out! This is actually a typo that occurred while generalizing our setting from the $2$-arm case to the $K$-arm case. The sampling should indeed be uniform over $[K]$, as you correctly mentioned. We will revise this in the final version of our paper.
>
> **Q3**
>
> Thanks again for your careful reading and good catch. In EG, $\kappa_{ti} (\omega_{ti},X_{ti})$ should be $\epsilon_{ti}(K-1)/K$, instead of $\epsilon_{ti}/2$ (which holds only when $K=2$). This is because the probability of exploration is $\epsilon_{ti}$ in EG, and the fraction allocated to randomly exploring the optimal arm is $1/K$ of $\epsilon_{ti}$. Thus, the correct expression should be
> $ \kappa_{ti} (\omega_{ti},X_{ti}) = (1 - 1/K)\epsilon_{ti} = \frac{\epsilon_{ti} (K-1)}{K}$. We will correct this in the final version of our paper.
>
> **Q4**
>
> Section 3.1 expresses the payoff function as $f(X_{tj},a) = X_{tj} \beta_a$, which assumes linearity with respect to the contextual information $X_{tj}$ for each action $a \in \mathcal{A}$. This linear formulation is widely adopted in the linear contextual bandits literature, such as  (Chu et al., 2011; Agrawal & Goyal, 2013). If I understand your suggestion correctly (i.e., representing $f$ as a simple inner product when each action is expressed as a vector), then Section 3.1 is already aligned with this idea. If we encode the action as a dummy variable vector, the function can be rewritten as an inner product, which is equivalent to your suggestion.
>
> There are several ways to extend this linear payoff assumption. First, $X_{tj}$ can be transformed using basis functions (e.g., polynomial features) to incorporate higher-order representations as needed, which is a straightforward extension. Another direction is to integrate neural bandits into this framework by modeling $f(X_{tj}, A_{tj})$ using a neural network. While this approach allows direct adaptation of existing neural bandit algorithms to interference-aware settings, deriving theoretical guarantees (particularly asymptotic properties, as established in Section 4) becomes significantly more challenging. This is also why we begin with a linear payoff function.
>
> **Regarding your suggestions about notation and writing**: We sincerely appreciate your feedback on Section 3.1. We will refine this section by adding details, clarifying concepts like $\widetilde{X}_t$, and updating the transpose notation to $\beta^\top$ for better clarity and readability.

---

### Official Review · Reviewer_f3ef · 2025-03-14

**Overall Recommendation:** 3

**Summary:**

The paper introduces a framework to address Linear Contextual Bandits with interference, where the actions of one unit can affect the rewards of others. The authors bridge the gap between causal inference and online decision-making by explicitly modeling interference through a linear structure involving an interference matrix. They propose three online algorithms LinEGWI, LinUCBWI, and LinTSWI, which extend classical algorithms by incorporating interference-aware reward modeling. They establish regret bounds and provide numeraical results on the generated based on the MovieLens data.

**Claims And Evidence:**

The claims made in the paper are supported by theoretical derivations and empirical evidence and no problematic claims were found.

**Essential References Not Discussed:**

The authors have cited the most relevant previous works in the area.

**Ethical Review Flag:**

Flag this paper for an ethics review.

**Experimental Designs Or Analyses:**

I reviewed the experimental setup and analyses presented in Section 5 and 6. The experimental design is clear and reasonable although no baseline for LinCB under interference are available and hence the algorithms are compared to their non-interference algorithmic baselines.

**Methods And Evaluation Criteria:**

The proposed methods and evaluation criteria are appropriate for addressing interference in contextual bandit problems. The authors clearly define interference via an adjacency matrix, allowing flexible modeling of different real-world problems.

**Other Comments Or Suggestions:**

Typos:

- Lines 31-32: "a synthetic data generated" → "synthetic data generated".
- Lines 106-107: "to to" → "to".
- Line 412: "of interference of interference" → "of interference".
- Line 151-152: "i.e." → "i.e., ".

**Other Strengths And Weaknesses:**

Strengths:

- First work addressing contextual bandits under general interference.
- Novel asymptotic analysis of estimators under complex dependencies induced by interference.

Weaknesses:
- The computational complexity of the proposed algorithms is unclear, which is typically an important aspect in this subclass of bandits.
- The way the proposed "Pseudo-true-reward” generating process (as Described in App. B3) is carried out is a bit dubious (see questions below)

**Questions For Authors:**

1. The authors state that their algorithm can be adapted in case of unknown interference Matrix inspired by techniques on low-rank factorisation. Could you please discuss at a high level what you believe the impact of learning such a matrix would yield to the algorithm performance? Is there any way to quantify the impact in case of misspecified interference matrix ?

2. Can you characterize (even roughly) the computational complexity of the proposed online algorithms LinEGWI, LinUCBWI, LinTSWI?

3.  Why if there are more observations from the same user the correspojding element in the intereferene matrix is set to 1? Is there any way to evaluate the fit of your model (i.e. the reward you design in Point I,II in App. B3) to the actual data?

**Relation To Broader Scientific Literature:**

This paper extends recent works in the bandits with interference literature to the Linear Contextual setting. Related works

**Theoretical Claims:**

I skimmed through the theoretical results but did not thoroughly verify all proofs in detail.

---

> ### Author Rebuttal · Authors · 2025-03-31
>
> Thanks for your thoughtful questions and the time you spent reviewing our paper. We really appreciate your insights and are
> happy to discuss any further ideas or questions you may have.
>
> **Q1**:
>
> In the 2nd paragraph of Sec 7, we briefly mentioned how we could proceed to jointly estimate $(\Phi, \beta)$ when $W_t$ is unknown. A simple way to proceed is through **Alternating Optimization**: initializing $\Phi$, optimizing for $\beta$, then fixing $\beta$ to update $\Phi$, and repeating until convergence.
>
> Sseveral interesting questions arise. First, identifiability of $(\Phi, \beta)$ needs careful consideration, especially since $\widetilde{X}_t$ depends on $\Phi$, which may introduce issues in distinguishing their effects in $\widetilde{X}_t\beta$ without additional assumptions. Second, the convergence of the alternating updates requires further investigation. Finally, the sample size for accurate estimation of both parameters, and consequently, for effective action learning and reward accumulation, would likely be higher than in the case where $W_t$ is known. We expect that **the algorithm-wise implementation is relatively straightforward, but how to establish the theory behind would be an interesting future work**.
>
> To evaluate the impact of a misspecified interference matrix, we conducted a sensitivity analysis, summarized in Sec 1 of https://anonymous.4open.science/r/LinCBWI_ICML_rebuttal-C829/. The data follows the same simulation setup as Sec 5.2, except we manually set a misspecified matrix for LinCBWI: $\breve{W}_t = W_t + \Xi_t$, where each element of $\Xi_t$ is generated from $Unif(-b,b)$. Any values in $\breve{W}_t$ exceeding $[-1,1]$ are clipped. Notably, LinCBWI remains robust, converging to the optimal decision with negligible error even under substantial misspecification ($b \leq 0.5$). Even in the extreme case where $\breve{W}_t \sim \text{Unif}(-1,1)$ (completely unrelated to $W_t$), LinCBWI slightly outperforms classical LinCB. This suggests that **incorporating interference structures, even when poorly estimated, enhances flexibility and provides deeper insights into the bandit framework**. We hope LinCBWI’s tolerance to misspecification alleviates concerns about unknown interference.
>
> **Q2:**
>
> The computational complexity of LinCBWI is approximately $O(d^2K^2\bar{N}_T)$. Specifically, for each unit $i$ in round $t$, the primary computational costs arise from matrix inversion (Line 7 of Algorithm 1) and the smallest eigenvalue computation (Line 10), both requiring $O(d^3K^3)$ in standard implementations. However, using optimization techniques such as iterative eigenvalue decomposition and the Sherman-Morrison update can reduce this to $O(d^2K^2)$. Multiplying by the total number of units $\bar{N}_T$ gives the overall complexity.
>
> To empirically validate this, we compared the runtime of LinCBWI with LinCB in Sec 2 of https://anonymous.4open.science/r/LinCBWI_ICML_rebuttal-C829/. In classical LinCB, the complexity is $O(d^2K\bar{N}_T)$. LinCBWI introduces an additional factor of $K$ due to interference, requiring a joint update of $\beta_a$ for all $a \in \mathcal{A}$. Since our algorithm consistently runs within seconds, **computational complexity is unlikely to be a bottleneck**.
>
> **Q3:**
>
> When multiple observations are collected from the same user within a single round, each movie recommendation $A$ and its corresponding reward $R$ generate a new data tuple $(X,A,R)$, meaning a user can contribute multiple tuples per round with the same $X$ but potentially different $(A,R)$. These observations can be viewed as coming from "two persons with the same brain". As a result, these data tuples naturally exhibit interference with the highest weight (set to 1), since they originate from the same individual.
>
> Regarding the question of how to evaluate the fit of reward design in 'I' and 'II' of Appendix B.3, there is insufficient data to directly assess their closeness using the existing dataset. This limitation arises because we are working with an offline dataset to simulate an online bandit setting, where the "actual" reward is typically assumed to be observable.  In our setup, $R_{ti}$ may depend on the actions of all units in round $t$, leading to $K^{N_t}$ possible reward realizations. However, in an offline dataset, we can observe only a **single** realization among these possibilities. As a result, the "actual" reward is not directly identifiable; we can only approximate a pseudo-real data environment to capture certain aspects of real-world behavior.
>
> Notably, the only two existing works on interference in bandit settings (Jia 2024; Agarwal 2024) have only conducted synthetic experiments. By incorporating semi-real data analysis, our approach takes a step forward that better approximates real-world applications.
>
> **Finally**, thank you for pointing out the typos in our writing. We will incorporate them into the final version of
> our paper and sincerely look forward to your feedback.

---

> > ### Comment · Reviewer_f3ef · 2025-04-06
> >
> > Thanks for your answer and the additional results, it resolved my concern. I increased the score accordingly.

---

> > > ### Author Response · Authors · 2025-04-08
> > >
> > > We sincerely appreciate your time and thoughtful review. We're glad to hear that our responses addressed your concerns, and we thank you for your updated evaluation.
> > >
> > > Best,
> > >
> > > Authors from submission 4971

---

### Decision · Program_Chairs · 2025-05-01

**Decision:**

Accept (poster)

**Comment:**

After a very effective author-PC discussion phase, the authors cleared a few concerns about the paper. All the 4 reviewers ultimately are positive about the contributions of this paper, which extends classic linear bandits to situations with inner action interference. Hope the authors could take into account various useful comments from the review process to improve the paper in its next version, but overall this paper is a great addition to ICML!